
# Diurnal variation and size-dependence of the hygroscopicity of organic aerosol at a forest site in Wakayama, Japan: their relationship to CCN concentrations

Yange Deng[1], Hikari Yai[1], Hiroaki Fujinari[1], Kaori Kawana[2,3], Tomoki Nakayama[4,5], and Michihiro Mochida[1,4]

[1]Graduate School of Environmental Studies, Nagoya University, Nagoya, Japan
[2]Institute of Low Temperature Science, Hokkaido University, Hokkaido, Japan
[3]Now at School of Materials and Chemical Technology, Tokyo Institute of Technology, Tokyo, Japan
[4]Institute for Space-Earth Environmental Research, Nagoya University, Nagoya, Japan
[5]Now at Graduate School of Fisheries and Environmental Sciences, Nagasaki University, Nagasaki, Japan

*Correspondence to*: Michihiro Mochida (mochida@isee.nagoya-u.ac.jp)

**Abstract.** Formation of biogenic secondary organic aerosol (BSOA) and its subsequent evolution can modify the hygroscopicity of the organic aerosol component (OA) in the forest atmosphere, and affect the concentrations of cloud condensation nuclei (CCN) there. In this study, size-resolved aerosol hygroscopic growth at 85 % relative humidity and size-resolved aerosol composition were measured using a hygroscopic tandem differential mobility analyzer and an aerosol mass spectrometer, respectively, at a forest site in Wakayama, Japan, in August and September 2015. The hygroscopicity parameter of OA ($\kappa_{\mathrm{org}}$) presented daily minima in the afternoon hours, and it also showed increase with the increase of particle dry diameter. The magnitudes of the diurnal variations of $\kappa_{\mathrm{org}}$ for particles with dry diameters of 100 and 300 nm were on average 0.091 and 0.096, respectively, and the difference of $\kappa_{\mathrm{org}}$ between particles with dry diameters of 100 and 300 nm was on average 0.056. The relative contributions of the estimated fresh BSOA and regional OA to total OA could explain 40 % of the observed diurnal variations and size-dependence of $\kappa_{\mathrm{org}}$. The hygroscopicity parameter of fresh BSOA was estimated to range from 0.089 to 0.12 for particles with dry diameters from 100 to 300 nm. Compared with the use of time- and size-resolved $\kappa_{\mathrm{org}}$, the use of time- and size-averaged $\kappa_{\mathrm{org}}$ leads to under- and over-estimation of the fractional contribution of OA to CCN number concentrations in the range from −4.9 to 26 %. This indicates that the diurnal variations and size-dependence of $\kappa_{\mathrm{org}}$ strongly affect the overall contribution of OA to CCN concentrations. The fractional contribution of fresh BSOA to CCN number



concentrations could reach 0.28 during the period of intensive BSOA formation. The aging of the fresh BSOA, if it occurs, increases the estimated contribution of BSOA to CCN number concentrations by 50–84 %.

## 1 Introduction

The hygroscopicity (ability to absorb water) of organic aerosol (OA) components is governed by their chemical composition, and has two important roles in the atmosphere. It influences light scattering by aerosols (Titos et al., 2016) and affects the ability of aerosols to work as cloud condensation nuclei (CCN; McFiggans et al., 2006). The hygroscopicity of OA may also influence the aqueous chemistry in aerosols and cloud droplets, which provide a potentially important pathway for the formation of secondary organic aerosols (SOA; McNeill, 2015). Nevertheless, the hygroscopicity of OA is not well characterized in terms of its temporal and spatial variations, size-dependence, and its relationship with the chemical composition of OA, given that OA is a complex mixture of a number of compounds.

Studies on the hygroscopicity parameter $\kappa$ of OA ($\kappa_{org}$; Petters and Kreidenweis, 2007) in different locations have presented different characteristics of temporal variation and size-dependence. Based on a year-long observation under supersaturated water vapor conditions (SUPS) at a downwind site of Manaus in central Amazonia, Thalman et al. (2017) reported that $\kappa_{org}$ presented the lowest value of ~0.1 in September and the highest value of ~0.15 in December, and that the ranges of the diurnal variations of $\kappa_{org}$ were 0.10 to 0.16 (night to day) and 0.08 to 0.14 (night to day) under the influence of local biomass-burning air masses during the dry season and urban-pollution air masses during the wet season, respectively. Bougiatioti et al. (2016) reported diurnal variation of $\kappa_{org}$ in the range 0.09–0.18 (day to night) under SUPS for particles influenced by biomass burning in the eastern Mediterranean. Deng et al. (2018) reported diurnal variation ranges of $\kappa_{org}$ of 0.09 to 0.30 (day to night) and 0.16 to 0.24 (day to night) on days with and without evident new particle formation (NPF), respectively, under SUPS (0.23 % water vapor supersaturation (SS) condition) in a forest in Wakayama, Japan. Different from the above studies, at a rural site in the southeastern United States, a small diurnal variation of $\kappa_{org}$ (~0.13 to ~0.17, night to day) under SUPS was observed (Cerully et al., 2015). With respect to the size-dependence of $\kappa_{org}$, airborne studies over United States, Canada, Pacific Ocean, and the Gulf of Mexico for a variety of air mass types, presented a decrease of $\kappa_{org}$ (from 0.13 to 0.06) with increase of the particle modal diameter (from 130 to ~210 nm) under subsaturated water vapor conditions (SUBS) (Shingler et al., 2016). By contrast,



ground-based observations in the City of Nagoya in Japan under SUBS presented relatively low $\kappa_{org}$ in small particles (0.12–0.15; 60 and 100 nm) and high $\kappa_{org}$ in large particles (0.17–0.22; 200 and 359 nm) (Kawana et al., 2016). For aerosols in forest areas, whereas Deng et al. (2018) reported a difference of 0.03 in mean $\kappa_{org}$ between sub-100 nm particles ($\kappa_{org}$ was 0.19) and ~150 nm particles ($\kappa_{org}$ was 0.22), Thalman et al. (2017) did not identify any size-dependence of $\kappa_{org}$ for 94–171 nm particles.

The hygroscopicity of laboratory-generated model SOA was also reported to be size-dependent. Frosch et al. (2011) reported that the $\kappa$ of α–pinene SOA at SUPS at 100 nm (~0.12) was ~0.06 higher than at 200 nm. Zhao et al. (2015) reported that the $\kappa$ of model SOA at SUPS produced by different precursors at 50, 100, and 200 nm were ~0.17, ~0.11, and ~0.07, respectively. Tritscher et al. (2011) also found that small (50 nm) α-pinene SOA particles had a higher $\kappa$ than large ones (150 nm) at SUBS, although the difference was small (0.03). Frosch et al. (2013) reported that the $\kappa$ of β-caryophyllene SOA decreased with

increase of SS, which can be interpreted as the increase of $\kappa$ with the increase of particle diameter, and that the maximum of the difference was about 0.1.

The variations in $\kappa_{org}$ observed in the aforementioned studies may have great influence on the prediction of CCN. Based on global climate modelling simulations, Liu and Wang (2010) reported that CCN concentration would change within 40 % by changing the $\kappa$ of SOA by ±50 % (from 0.14 to 0.07 or 0.21). Rastak et al. (2017) reported that the difference of the aerosol

radiative effects between $\kappa_{org}$ of 0.05 and 0.15 was −1.02 W m$^{-2}$, the order of which is the same as that of the overall climate forcing effect of anthropogenic aerosol during the industrial period. Based on CCN closure studies, Wang et al. (2008) reported that, for above-cloud aerosols with high volume fractions of OA, while the CCN number concentration closure could be achieved using $\kappa_{org}$ of 0.12, the use of $\kappa_{org}$ of 0.25 led to overestimation of the CCN number concentration by 50 %. Mei et al. (2013b) reported that the increase of $\kappa_{org}$ from 0.08 to 0.13 led to 30 % increase of the calculated CCN number concentration

and that increase from 0.03 to 0.18 doubled the concentration. It is therefore important to study the temporal variation and size-dependence of $\kappa_{org}$ in more locations where OA dominates the aerosol chemical composition, to characterize the $\kappa_{org}$ values and to represent the $\kappa_{org}$ appropriately in model predictions of CCN number concentrations.

The temporal variation and size-dependence of $\kappa_{org}$ of ambient aerosol is reported to relate with variations in the chemical composition of OA, which can result from the mixing of aerosols of different origins, formation of SOA, and aging processes (Cerully et al., 2015; Bougiatioti et al., 2016; Shingler et al., 2016; Thalman et al., 2017; Deng et al., 2018). The size-dependent chemical composition of model SOA has been explained by the size-dependent contributions of different organic vapors to

particle growth (e.g., Winkler et al., 2012; Ehn et al., 2014; Zhao et al., 2015; Zhao et al., 2016). The size-dependent $\kappa$ of model SOA has also been explained from the viewpoint of size-dependent chemical composition (Zhao et al., 2015; Frosch et al., 2013) and other factors: the dependence of water activity on particle size, the dependence of the surface tension on the solution concentration, and the evaporation of semi-volatile SOA under high SS conditions (Frosch et al., 2011; Frosch et al., 2013; Zhao et al., 2015). In recent studies, the variation of $\kappa_{org}$ was explained by the variation of OA subcomponents derived

from positive matrix factorization (PMF) analysis of OA mass spectra (Cerully et al., 2015; Bougiatioti et al., 2016). From these studies it is reported that the daily variation of $\kappa_{org}$ could be well explained by the daily variation in the contributions of the retrieved PMF factors to $\kappa_{org}$.

In the forest atmosphere, the oxidation of biogenic volatile organic compounds (BVOC) emitted by vegetation can produce substantial amounts of biogenic secondary organic aerosols (BSOA; Tunved et al., 2006; Pöschl et al., 2010; Han et al., 2014).

BSOA is reported to contribute to the growth of newly formed particles in forests (e.g., Han et al., 2013; Yu et al., 2014; Zhou et al., 2015). BSOA may also condense on preexisting background particles or particles transported with inflowing air masses (e.g., Cerully et al., 2015; Thalman et al., 2017). Moreover, BSOA is subject to aging processes that include photochemical oxidation and aqueous phase reactions that must depend on ambient meteorological conditions (e.g., Han et al., 2014; Thalman et al., 2017). Such processes could result in time- and size-dependent variation in the chemical composition of OA and thus

time- and size-dependent $\kappa_{org}$ in the forest atmosphere. However, the characteristics of the temporal variations and size-dependence of $\kappa_{org}$, and their relationships to the atmospheric processes of BSOA in forest environments, are not well understood.

We performed field observation at a forest site in Wakayama, Japan in August and September, 2015, and characterized the

diurnal variations and size-dependence of $\kappa_{org}$. The variations and dependence were interpreted based upon the size-resolved

chemical composition of OA from the viewpoint of BSOA formation. Furthermore, the influence of these variations on the fractional contribution of OA and BSOA to the CCN concentration was assessed. Previous observational studies at the site indicated that BSOA formation was intensive and that aging occurred after formation (Han et al., 2014; Deng et al., 2018). It was also observed that $\kappa_{\mathrm{org}}$ was time and size dependent and that the contribution of OA and BSOA to CCN number

concentrations could be substantial (Deng et al., 2018). This work is an extension of previous studies on the hygroscopicity and CCN activity of aerosols, and the contributions of OA and BSOA to the CCN concentration, in the same forest (Kawana et al., 2017; Deng et al., 2018). It is intended to clarify the diurnal variation and size-dependence of the hygroscopicity of OA and their influence on the contributions of OA and BSOA to CCN.

## 2 Field observation

The field observation was performed at Wakayama Forest Research Station, Kyoto University (34.06° N, 135.52° E, about 500 m above sea level), located in the central part of the Kii Peninsula. The observation site is about 70 km south of Osaka (2.7 million inhabitants) and 60 km northwest of the North Pacific. Both coniferous trees (such as *Cryptomeria japonica*, *Chamaecyparis obtuse*, and *Pinus densiflora*) and broad-leaf trees (such as *Quercus serrata* and *Quercus crispula*) are distributed on the Kii Peninsula (Okumura, 2009). The study period was from 1430 Japan Standard Time (JST) on 31 August

to 0600 JST on 22 September 2015.

The hygroscopic growth at 85 % relative humidity (RH), number-size distributions, and size-resolved chemical composition of ambient aerosols were measured using a hygroscopicity tandem differential mobility analyzer (HTDMA), a scanning mobility particle sizer (SMPS), and a high-resolution time-of-flight aerosol mass spectrometer (AMS), respectively. Ambient air was aspirated from an inlet about 7.5 m above the ground. The air was transferred through a $PM_{2.5}$ cyclone (URG) installed

at the lower end of the 10.4 m stainless-steel inlet tubing (1/2-inch OD) and introduced to the instrument room at a flow rate of 16.7 L min$^{-1}$. A manifold combined with an assistant pump (ULVAC, DA30S) was used to split the air flow, and the sample flow for the instrument system composed of the HTDMA, SMPS, and AMS was 0.9 L min$^{-1}$. The sample flow upstream of the AMS was dried with two diffusion driers containing silica gel. The sample flow to the HTDMA and SMPS was dried with three diffusion driers, in series, two with silica gel and one with molecular sieves.



In the HTDMA, the dried aerosol (RH < 1.8 %) was passed through the first differential mobility analyzer (DMA1; 3081, TSI), where the aerosol was classified, and quasi-monodisperse particles of 30, 50, 70, 100, 200, 300, and 360 nm in diameter ($d_{dry}$) were obtained. The setting for the classification was fixed for 5 min at each diameter. In each hour, the sequential diameter setting of DMA1 was 30, 50, 70, 100, 200, 360, 30, 50, 100, 200, 300, and 360 nm. During 0550–0554 JST and 1750–1754 JST, the setting of DMA1 was for system performance check. The classified aerosol was passed through a Nafion humidifier (MD-110-24S-4, Perma Pure) where it was humidified to ~85% RH. The aerosol was then introduced to a second DMA (DMA2; 3081, TSI) coupled to a condensation particle counter (CPC, 3775, TSI), which was operated by scanning the voltage of DMA2. For both DMA1 and DMA2, the aerosol flow rate was 0.3 L min$^{-1}$ and the sheath-to-sample flow ratio was 10:1. The residence time of the monodisperse particles from the outlet of the humidifier to the inlet of DMA2, where the RH was considered to be ~85 %, was approximately 11 s. The sheath air flow of DMA2 was also humidified using another Nafion humidifier (PD-100T-12MSS, Perma Pure). The RH (temperature) measured (HMP237, Vaisala) at the inlets of aerosol flow and sheath flow to DMA2 were 85.0 ± 0.2 % (20.3 ± 0.5 ℃) and 85.0 ± 0.2 % (20.4 ± 0.5 ℃), respectively, and that at the outlet of the sheath flow of DMA2 was 86.0 ± 0.3 % (20.3 ± 0.6 ℃). For analysis of the particle hygroscopic growth, RH of 85 % was applied. The SMPS for the measurement of aerosol number-size distributions was composed of a third DMA (DMA3; 3080, 3081, TSI) and a CPC 3772 (TSI). The aerosol flow (RH < 1.6 %) was 0.3 L min$^{-1}$ and the sheath to aerosol flow ratio was 10:1. At the inlet of CPC 3772, the sample flow was diluted to 1 L min$^{-1}$ with purified dry air, which was generated using a compressor (RD-45-N, IAC) and an air dryer (QD 30-50, IAC). The aerosol number-size distributions were measured for a dry diameter range of 13.8–749.9 nm every 5 min. The performances of the three DMAs were assessed using standard size PSL particles before and after the observation (Text S1). Furthermore, an aqueous solution of ammonium sulfate (AS) (99.999 % purity, Sigma-Aldrich) was nebulized and the generated aerosols were dried and introduced to the HTDMA, to assess the consistency of the sizing of the two DMAs under dry condition, and to validate the RH setting before the observation (Text S2).

The setup and calibration procedures of the AMS were the same as those for the observations in 2014 (Deng et al., 2018). The V-mode (MS and PToF modes) data was analyzed using the Igor high resolution data analysis package (PIKA1.20Q, Igor 6.37) to obtain the bulk and size-resolved mass concentrations of the chemical components (sulfate ($SO_4$), ammonium ($NH_4$), nitrate



(NO$_3$), chloride (Chl), and OA), and the atomic ratios of O to C (O:C ratio) and H to C (H:C ratio) for organics. In addition, high-resolution bulk OA mass spectra observed in V-mode were subjected to PMF analysis (Paatero and Tapper, 1994; Ulbrich et al., 2009) (Sect. 3.3, Text S3). The RH of the sample flow was lower than 0.5 %.

A single wavelength particle soot absorption photometer (1λ-PSAP, 567 nm, Radiance Research Inc.) with a thermodenuder

maintained at 300 °C, was deployed to obtain the mass concentration of sub-micrometer black carbon (BC; Kondo et al. 2009; Deng et al., 2018). The mixing ratios of target gaseous species, NO-NO$_2$-NO$_x$, CO, CO$_2$, and O$_3$, were monitored using commercial instruments (APNA-370, Horiba for NO-NO$_2$-NO$_x$; model 48ij, Thermo Fisher Scientific for CO; LI-820, LI-COR for CO$_2$; model 49ij, Thermo Fisher Scientific for O$_3$). Meteorological data were collected (Kyoto University, 2017). Air temperature, RH (HMP-155, Vaisala), precipitation (RH-5E, IKEDA-KEIKI), and solar radiation (CMP3-L, Campbell) were

used in this study.

All the observation data except meteorological data were screened to eliminate data that might have been under the strong influence of local anthropogenic emissions, for example, from vehicles. This was performed by omitting data with spikes in the number concentrations of aerosols from their size distribution data, and in the mass concentration of BC (Text S4).

## 3 Data analysis

### 3.1 Hygroscopicity of ambient aerosols

The hygroscopic growth factor of aerosol particles, $g_f$, was defined as the ratio of the particle wet diameter ($d_{wet}$, 85 % RH) to the corresponding dry diameter ($d_{dry}$). The distributions of $g_f$ for specific $d_{dry}$ ($n(g_f)$, i.e., the number distribution of particles as a function of $g_f$) were retrieved using the Twomey algorithm as presented by Mochida et al. (2010) with consideration of the shape of the transfer functions of the two DMAs. The difference in the processing is that the transfer function and the $n(g_f)$ in

this study were analyzed and presented in the fine mode of 1024 diameter bins per decade while 64 bins per decade were used in Mochida et al. (2010). The $g_f$ probability distribution function, $g_f$-PDF, is the normalized $n(g_f)$. The $g_f$-PDF in this study is presented in linear scale, which was converted from the original logarithmic scale distribution. The time-resolved mean value of $g_f$ for respective $d_{dry}$, $g_{f,m}$, was calculated as follows.

$$g_{f,m} = \frac{\sum n(g_f) g_f}{\sum n(g_f)} \quad (0.8 \le g_f \le 2.2 \text{ for } 30 \le d_{dry} \le 300 \text{ nm, or } 0.8 \le g_f \le 2.0 \text{ for } d_{dry} = 360 \text{ nm}) \tag{1}$$




For ambient particles, time-resolved mean-water-volume equivalent $g_f$ ($g_{f,mw}$, i.e., the average of $g_f$ that corresponds to the mean water volume retained by particles of certain $d_{dry}$) was also calculated using Eq. (2) (Kawana et al., 2016).

$$g_{f,mw} = \left[\frac{\sum n(g_f)(g_f^3 - 1)}{\sum n(g_f)} + 1\right]^{\frac{1}{3}} \quad (0.8 \leq g_f \leq 2.2 \text{ for } 30 \leq d_{dry} \leq 300 \text{ nm, or } 0.8 \leq g_f \leq 2.0 \text{ for } d_{dry} = 360 \text{ nm}) \tag{2}$$

The hygroscopicity parameter of ambient particles at 85 % RH ($\kappa_i$) was calculated following the $\kappa$-Köhler theory (Petters and Kreidenweis, 2007).

$$\kappa_t = \left(g_{f,mw}^3 - 1\right)\left[\frac{\exp\left(\frac{4\sigma M_w}{RT\rho_w d_{wet}}\right)}{0.85} - 1\right] \tag{3}$$

where $\sigma$ is the surface tension at the solution/air interface, $M_w$ and $\rho_w$ are the molecular mass and density of pure water, respectively, $d_{wet}$ is the product of $g_{f,mw}$ and $d_{dry}$, $R$ is the universal gas constant, and $T$ is the temperature in kelvin. In this study, the mean temperature at the inlets of aerosol flow and sheath flow of DMA2, weighted by their flowrates, was applied as $T$ (294 K) and the surface tension of pure water at this temperature (Vargaftik et al., 1983) was used as $\sigma$ in Eq. (3). Because $\kappa_i$ was calculated from $g_{f,mw}$, the aerosol mixing state was not considered in the analysis of $\kappa$ in this study.

## 3.2 Hygroscopicity of OA

The hygroscopicity parameter of organics, $\kappa_{org}$, was calculated using Eq. (4) assuming the volume additivity of water retained by different aerosol components (Petters and Kreidenweis, 2007).

$$\kappa_t = \varepsilon_{org}\kappa_{org} + \varepsilon_{inorgsalt}\kappa_{inorgsalt} + \varepsilon_{BC}\kappa_{BC} = \varepsilon_{org}\kappa_{org} + \sum_{i=1}^{5}\varepsilon_i \kappa_i + \varepsilon_{BC}\kappa_{BC}$$

$$\tag{4}$$

Here, $\kappa_i$ is the hygroscopicity parameter of ambient aerosol at 85 % RH calculated using Eq. (3), while $\varepsilon_{org}$, $\varepsilon_{inorgsalt}$, and $\varepsilon_{BC}$ are the volume fractions of OA, inorganic salts, and BC, respectively. The hygroscopicity parameters of OA, inorganic salts, and BC are $\kappa_{org}$, $\kappa_{inorgsalt}$, and $\kappa_{BC}$, respectively, and $\varepsilon_i$ and $\kappa_i$ are the volume fraction and hygroscopicity parameter of the inorganic salts: ammonium nitrate (AN), sulfuric acid (SA), ammonium hydrogen sulfate (AHS), letovicite (LET), and ammonium sulfate (AS). The $\varepsilon_{org}$, $\varepsilon_i$, and $\varepsilon_{BC}$ were calculated based on the size-resolved mass concentrations of organics, sulfate, nitrate, and ammonium from the AMS, and the sub-micrometer BC mass concentrations from the PSAP. BC was





assumed to have the same mass-size distribution as OA. The aerosol particles were assumed to be spherical and without voids.

PToF mode data in vacuum aerodynamic diameter ($d_{va}$) ranges that were ~1.0 (0.98–0.99) to 2.0 times that of $d_{dry}$, corresponding to the particle density of ~1.0 (0.98–0.99) to 2.0 g cm$^{-3}$, were adopted. More details about the calculations of the size-resolved $\varepsilon_{org}$, $\varepsilon_i$, and $\varepsilon_{BC}$ are presented in Text S5. The derivation of $\kappa_i$ was based on the online Extended AIM Aerosol

Thermodynamics Model II (E-AIM II, http://www.aim.env.uea.ac.uk/aim/kohler/input_kohler.html; Clegg et al., 1998; Wexler and Clegg, 2002) as presented in Text S6 and Table S3. The $\kappa$ of BC was assumed to be zero. Because of the low signal intensity of the PToF data in the sub-100-nm $d_{va}$ range (Text S7), the $\kappa_{org}$ was only derived for particles with $d_{dry}$ of 100, 200, 300, and 360 nm. Furthermore, to assess the influence of the choice of the $d_{va}$ range on the derivation of $\kappa_{org}$, the derived $\kappa_{org}$ for particles with $d_{dry}$ of 100 nm using the chemical composition in the $d_{va}$ range 98–197 nm, was compared with that using

the chemical composition in the $d_{va}$ range 69–138 nm (Fig. S4). The result indicates that $\kappa_{org}$ was not sensitive to change in the selected $d_{va}$ range when $\varepsilon_{org}$ was greater than 40 %. Note that, although the volume additivity assumption between organics and inorganics may not necessarily hold (Vaishya et al., 2013; the $\kappa_{org}$ derived in the manner in this study represents the perturbation of $\kappa_t$ as a result of the presence of organics), the inverse linear correlation between $\kappa_t$ and $\varepsilon_{org}$ (correlation coefficients: −0.45 to −0.83; Fig. S5) suggests that the additivity holds well for the aerosols studied.

**3.3 PMF analysis of OA mass spectra**

To characterize the diurnal variations and size-dependence of $\kappa_{org}$, the high-resolution OA bulk mass spectra derived from the V-mode AMS data were subjected to PMF analysis (Text S3), followed by derivation of the size-resolved contributions of the PMF factors to the OA mass concentration (Text S8). A two-factor PMF solution was adopted, which resolved two oxygenated OA factors: one with a lower atomic O:C ratio (0.47) named less-oxygenated organic aerosol (LOOA), and the other with a

higher O:C ratio (0.95) named more-oxygenated organic aerosol (MOOA). The low relative residual (2.6 %) for the bulk mass spectra supports the use of the two PMF factors to illustrate the observed OA. Note that the two OA factors resolved here represent two different groups of OA chemical structures, not necessarily two different OA sources (Zhang et al., 2011). The use of a PMF result with more factors could make illustration of the variation of $\kappa_{org}$ complex and was not adopted. The PToF mode OA mass spectra in 2 h time resolution were attributed to the two PMF factors through multivariable linear regression





(Text S8). For particles with $d_{dry}$ equals to or larger than 100 nm, the variation of $\kappa_{org}$ was discussed with regard to the variations of the two PMF factors. Furthermore, the hygroscopicity parameters for the two OA fractions were derived, and then used to estimate the hygroscopicity of freshly formed BSOA (Sect. 4.2).

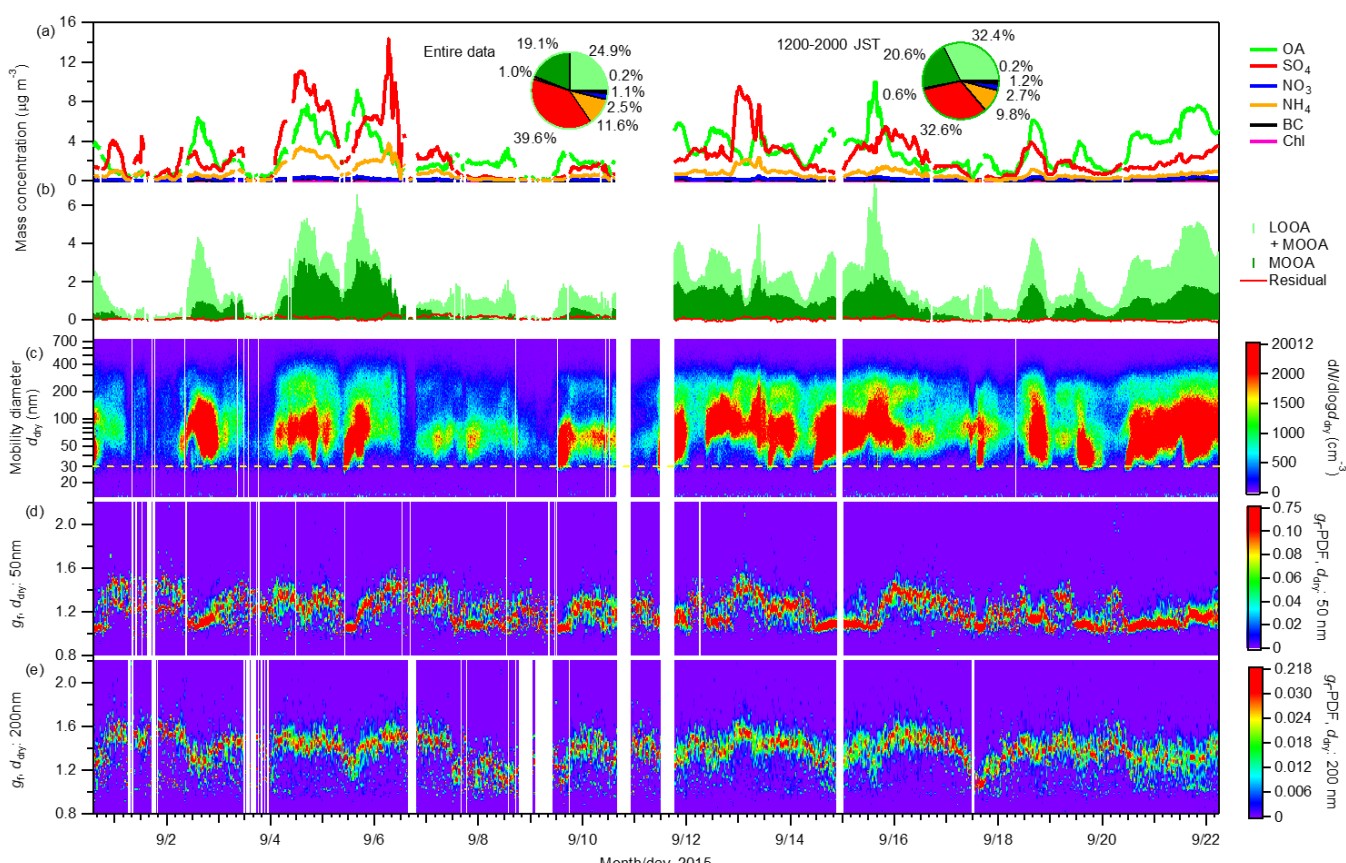

**Figure 1:** Time series of (a) sub-micrometer mass concentrations of non-refractory aerosol chemical components (OA, SO$_4$, NO$_3$, NH$_4$, and Chl) from the AMS measurement and BC from the PSAP measurement, (b) mass concentrations of LOOA and MOOA, and the residuals from the PMF analysis, (c) aerosol number-size distributions, and $g_f$-PDF of aerosol particles with $d_{dry}$ of (d) 50 nm and (e) 200 nm. The two pie charts in panel (a) present the mass fractions of chemical components for the entire study period and for the afternoon hours (1200–2000 JST) during the study period. The dashed line in panel (c) represents a diameter of 30 nm.



## 4 Results and discussions

### 4.1 Overview of the observations

#### 4.1.1 Meteorological conditions, gaseous species, and aerosol chemical composition

During the observation, the mean ± standard deviation (SD) of the temperature and RH of the ambient air were 18.2 ± 2.4 ℃

and 94.2 ± 7.6 %, respectively. Precipitation events occurred intermittently during 1–3, 6–10, and 16–17 September (Fig. S7).

Backward air mass trajectories (Fig. S9) generated using NOAA's HYSPLIT atmospheric transport and dispersion modeling

system (Draxler and Hess, 1998) indicate that, except on 1 and 17 September, most of the air masses that arrived at the

observation site had traveled from the Japan archipelago (and even from the Asian continent) within five days, and may have

transported aged anthropogenic pollutants to the observation site. The mean ± SD of the BC concentration during the entire

study period was 0.07 ± 0.06 μg m$^{-3}$ (Fig. S7). The mean ± SD of the mixing ratios of CO, NO, NO$_2$, NO$_x$, and O$_3$ during the

entire study period were 164 ± 42, 0.33 ± 0.12, 0.56 ± 0.35, 0.63 ± 0.38, and 11.5 ± 8.4 ppb, respectively (Fig. S7). The

concentration of BC was low, and the mixing ratios of CO and NO$_x$ were modest. A daily maximum of BC appeared in the

afternoon hours (Fig. 2), which however might have been caused by the charring of OA at the heating temperature of 300 ℃.

The mixing ratios of CO and NO$_x$ tended to be relatively high during 1000–2200 JST (Fig. S8), which might have been caused

by the transport of anthropogenic pollution to the surface site by enhanced vertical convection in the daytime. The

concentration of O$_3$ was substantial and presented obvious diurnal variation (Fig. S7). On average, O$_3$ peaked during noon

with the solar radiation (Fig. S8), indicating the occurrence of photochemical reactions during the daytime.

The time series of the mass concentrations of aerosol chemical components and aerosol number-size distributions are presented

in Fig. 1. Among non-refractory aerosol chemical components derived from the AMS and BC derived from the PSAP (total

concentration: 6.2 ± 4.4 μg m$^{-3}$), organics on average accounted for the largest fraction (45.0 %; of which LOOA and MOOA

accounted for 24.9 % and 19.1 %, respectively), followed by sulfate (39.6 %) and ammonium (11.6 %). The contribution of

nitrate, BC, and chloride were minor: their mass fractions were on average 2.5 %, 1.1 %, and 0.2 %, respectively. The

contribution of OA to the sub-micrometer aerosol mass increased and that of sulfate decreased in the afternoon hours (1200–

2000 JST). The mean aerosol number concentration ($N_{CN}$) was 1241 ± 1012 cm$^{-3}$. The geometric mean diameter of the aerosols



ranged from 45 to 154 nm with a mean ± SD of 88 ± 17 nm. No strong burst of small particles (i.e., $d_{dry} < 30$ nm) was identified during the observation, which is different from the two former observations in 2010 and 2014 (Han et al., 2013; Deng et al., 2018).

The diurnal variation of the number-size distributions and the mass concentrations of the chemical components of aerosols are presented in Fig. 2. The $N_{CN}$, OA, and LOOA presented similar diurnal variation patterns. Their daily minima were observed between 0600 and 0830 JST. After 0830 JST, they increased monotonically and reached their maxima during 1500–1800 JST. Then they gradually decreased until approximately 0600 JST of the next day. MOOA also increased slowly (following the trend of LOOA) in the daytime and reached its maximum around 1800 JST. The pattern of the enhancement of OA in the

daytime followed that of the solar radiation (Fig. 2b), indicating that the enhancement of OA was caused by the formation of BSOA through photochemical reactions of BVOC (Han et al., 2014; Deng et al., 2018). This is supported by an analysis indicating that anthropogenic pollution was not the main contributor to the enhancement of OA, at least during the period 1200–1600 JST (Text S9), and by the report that primary biogenic OA is mainly in the supermicrometer aerosol diameter range in a forest environment, for the Amazon at least (Pöschl et al., 2010). Furthermore, the stronger enhancement of LOOA

than of MOOA indicates that the freshly formed BSOA was mainly composed of LOOA and had a low oxygenation state. The O:C ratio of OA increased slowly from around noon to midnight, together with the appearance of MOOA, indicating the aging of freshly formed BSOA (Han et al., 2014). Although no abrupt increase of sub-30 nm particles was observed, the increase in the number concentration of 30–50 nm particles around noon indicates the formation of new particles near the observation site. These particles probably had grown by the condensation of BSOA formed from BVOC at the time they were transported to

the observation site. The concentrations of nitrate and chloride stayed low, although they also presented maxima in the afternoon. Sulfate, which may have been strongly influenced by transported anthropogenic aerosol, did not present an obvious diurnal variation. This result supports the view that the contribution of anthropogenic OA to the observed enhancement of OA was small.



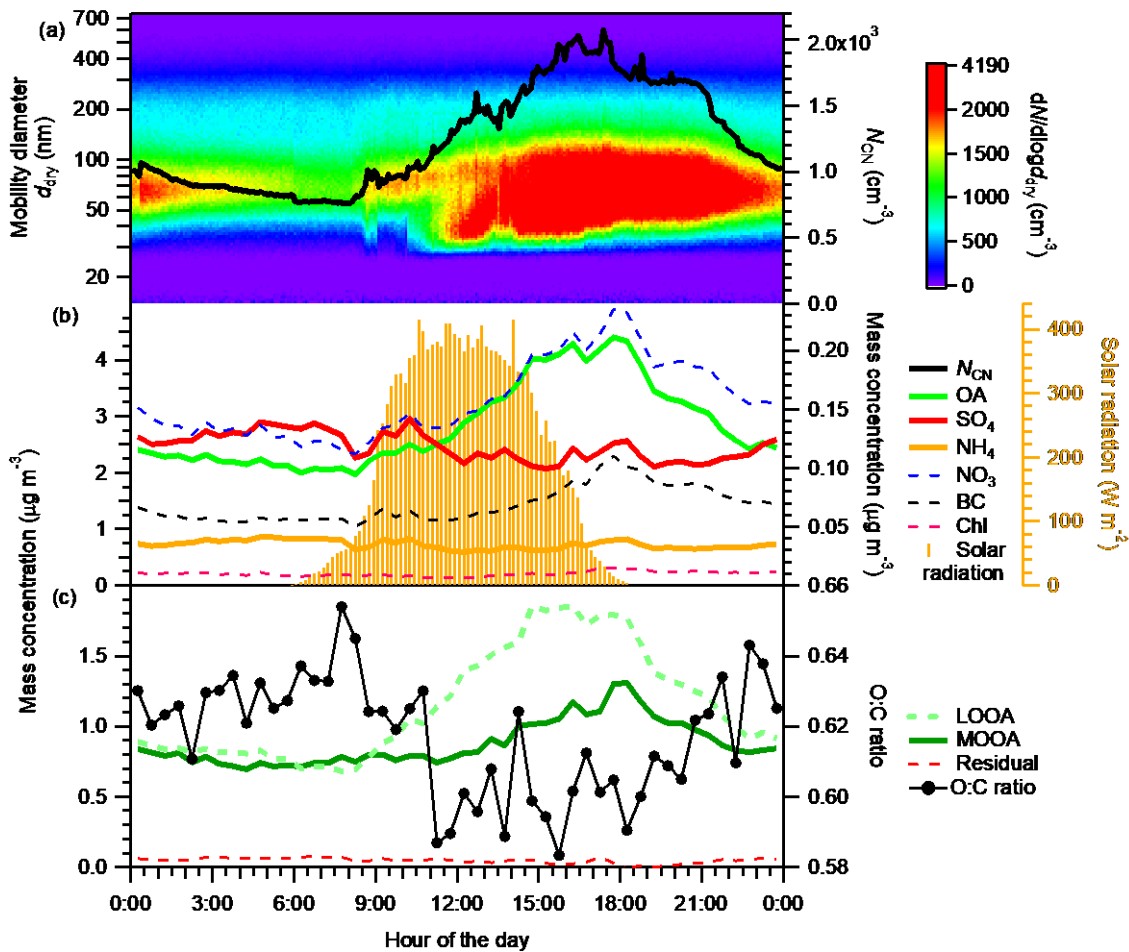

**Figure 2: Diurnal variations of (a) number-size distribution (image plot) and number concentration ($N_{CN}$, right axis) of ambient aerosols, and the mass concentrations of (b) OA, SO$_4$, and NH$_4$ (left axis), and NO$_3$, BC, and Chl (right axis in black), and (c) LOOA, MOOA, and residual, and the O:C ratio of bulk OA (only data with $m_{org} > 0.3$ µg m$^{-3}$ were included) averaged for the entire study period. The diurnal variation of solar radiation averaged for the entire period is superimposed in panel (b).**

### 4.1.2 Hygroscopicity of atmospheric aerosols

Similar to a former observation at the same site in 2010 (Kawana et al., 2017), the hygroscopic growth factor $g_f$ presented unimodal distributions at respective particle diameters (Figs. 1d, 1e, and S11), and the mean hygroscopic growth factor $g_{f,m}$ of the aerosols increased with increase of the particle diameters (Fig. S12). The mean ± SD of $g_{f,m}$ at 30, 50, 70, 100, 200, 300, and 360 nm were 1.13 ± 0.08, 1.21 ± 0.09, 1.22 ± 0.09, 1.26 ± 0.10, 1.36 ± 0.10, 1.40 ± 0.08, and 1.42 ± 0.08, respectively.





The unimodal pattern of the $g_f$-PDF indicated the internal mixing state of the observed aerosol at the respective particle diameters. Decreases of $g_{f,m}$ (Fig. S12) were observed for all particles during periods of intensive BSOA formation (i.e., episodes when the mass concentration of OA especially LOOA greatly increased; such episodes were observed on 31 August, and on 2, 5, 7, 9, 14, 15, 17, 18, 19, 20, and 21 September; Fig. 1b).

The hygroscopicity parameter of ambient aerosol particles that corresponds to $g_{f,mw}$ ($\kappa_t$) also increased with the increase of aerosol particle diameters (Fig. 3a). Similar diurnal variation patterns were observed for all the diameters studied. The $\kappa_t$ started to decrease around 0800 JST, then reached daily minima between 1300 and 1900 JST. Then it increased continually until around 0200 JST of the next day, and remained high until 0800 JST the next morning. For particles with $d_{dry} \geq 100$ nm, the diurnal variation pattern and size-dependence of $\kappa_t$ were opposite to those of the volume fraction of OA (Fig. 3b) and were

similar to those of the volume fraction of total inorganic salts (Fig. S13). The results suggest that, at least for ambient aerosol particles with $d_{dry} \geq 100$ nm, OA and inorganic salts had low and high hygroscopicity, respectively, resulted in the variations of $\kappa_t$. Although $\kappa_{inorgsalt}$ is much greater than $\kappa_{org}$ (Petters and Kreidenweis, 2007), the high $\varepsilon_{org}$ makes the influence of OA on $\kappa_t$ significant. Thus, the variation of $\kappa_{org}$ (Sect. 4.2) may also contribute to the variation of $\kappa_t$. For particles with $d_{dry} \leq 70$ nm, decrease of the particle hygroscopicity with decrease of the particle diameter is also explained by the accompanying increase

of $\varepsilon_{org}$ and the decrease of $\varepsilon_{inorgsalt}$ (Levin et al., 2014). This is indicated by the substantially lower mean mass fraction of inorganic salts than of organics in the corresponding $d_{va}$ range of less than 150 nm (Fig. S2). In particular, for particles with $d_{dry}$ of 30 nm, $\kappa_t$ remained constant in a low range (0.079–0.089) from 1300 JST to 1700 JST with a mean value of 0.082. BSOA formed during this period probably dominated the particle mass (Kawana et al., 2017; Han et al., 2014).



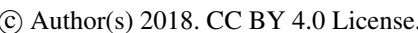

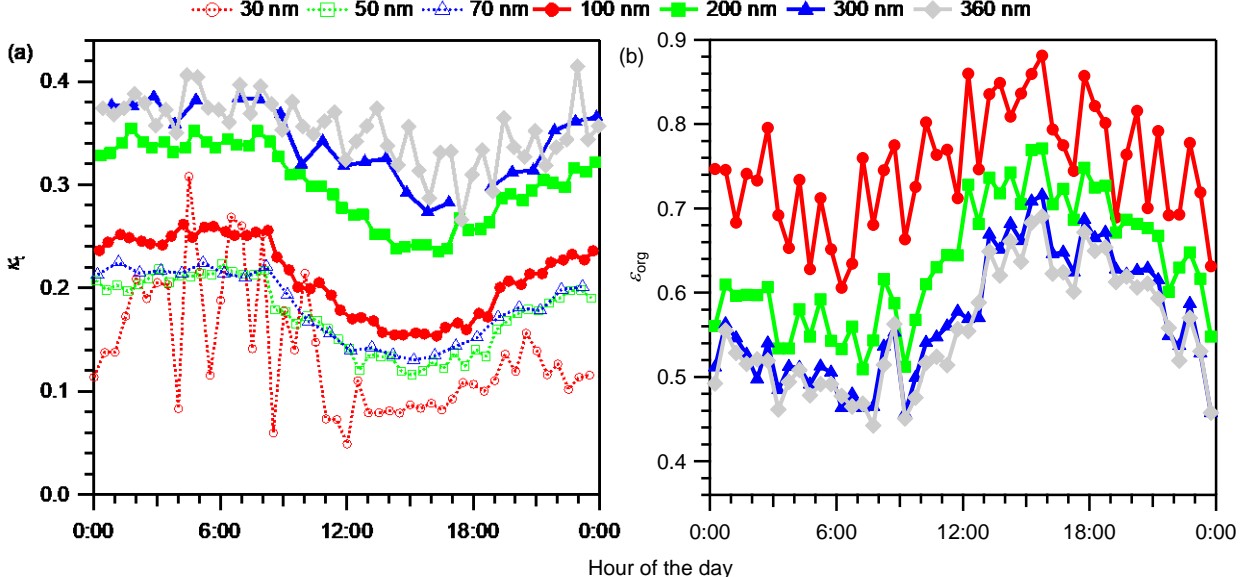

**Figure 3: Diurnal variations of (a) size-resolved hygroscopicity of aerosols ($\kappa_t$) and (b) size-resolved volume fractions of OA ($\varepsilon_{org}$) for the entire study period.**

## 4.2 Hygroscopicity of organic aerosol components

### 4.2.1 Variation of $\kappa_{org}$ and its relation to the chemical structure of OA

The diurnal variation of $\kappa_{org}$ and the volume fraction of LOOA in OA, $v_{LOOA}/(v_{LOOA}+v_{MOOA})$, where $v_{LOOA}$ and $v_{MOOA}$ refer to the volume concentrations of LOOA and MOOA, respectively, for the entire study period are presented in Fig. 4. Data with $\varepsilon_{org} < 0.40$ were excluded from the $\kappa_{org}$ values presented because the uncertainty that originated from subtraction of the contribution of inorganic components was considered large in the low $\varepsilon_{org}$ range (Mei et al., 2013a). The $\kappa_{org}$ decreased rapidly from approximately 0800 JST in the morning when the mass concentrations of OA and LOOA started to increase (Fig. 2). The $\kappa_{org}$ reached daily minima during 1000–1800 JST, and increased after the minima (Fig. 4a). The $v_{LOOA}/(v_{LOOA}+v_{MOOA})$ in Fig. 4b presents the opposite diurnal variation pattern. The characteristics of the size-dependence of $\kappa_{org}$ and $v_{LOOA}/(v_{LOOA}+v_{MOOA})$ were dependent on time periods. To characterize the size-dependence of $\kappa_{org}$ and $v_{LOOA}/(v_{LOOA}+v_{MOOA})$, the mean values of $\kappa_{org}$ and $v_{LOOA}/(v_{LOOA}+v_{MOOA})$ during 1200–2000 JST and 2000–1200 JST were plotted separately in the $\kappa_{org}$ - $v_{LOOA}/(v_{LOOA}+v_{MOOA})$

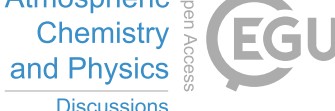

space for different $d_{dry}$ (Fig. 5), and the difference in the diurnal variation data between particles with different diameters were evaluated using a 10 % two-sided t-test (Table S7). During 1200–2000 JST, opposite size-dependences were observed between the mean $\kappa_{org}$ and $v_{LOOA}/(v_{LOOA}+v_{MOOA})$. Although the differences of $\kappa_{org}$ between 200 and 300 nm particles (p-value: 0.71) and 300 and 360 nm particles (p-value: 0.15) were not significant, the differences of $\kappa_{org}$ between 100 and 200 nm particles (p-value: 0.01) and 200 and 360 nm particles (p-value: 0.07), and the differences of $v_{LOOA}/(v_{LOOA}+v_{MOOA})$ between particles with all different diameters (p-value: <0.02), were significant during that period. During 2000–1200 JST, the size-dependences of $\kappa_{org}$ and $v_{LOOA}/(v_{LOOA}+v_{MOOA})$ were not clear. The clearer size-dependence of both $\kappa_{org}$ and $v_{LOOA}/(v_{LOOA}+v_{MOOA})$ during 1200–2000 JST than during 2000–1200 JST, was explained by the formation of BSOA during the afternoon hours. The patterns of diurnal variation and of size-dependence between $\kappa_{org}$ and $v_{LOOA}/(v_{LOOA}+v_{MOOA})$ during 1200–2000 JST indicate that the variation of $\kappa_{org}$ could be explained at least in part by the relative contributions of LOOA and MOOA to OA. That is, the presence of LOOA with low oxygenation state (O:C ratio of 0.47) lowered the observed $\kappa_{org}$, while the presence of MOOA with high oxygenation state (O:C ratio of 0.95) increased the observed $\kappa_{org}$. A similar relationship was observed in a former study at the observation site: $\kappa_{org}$ was positively correlated with the O:C ratio of the organics (Deng et al., 2018).

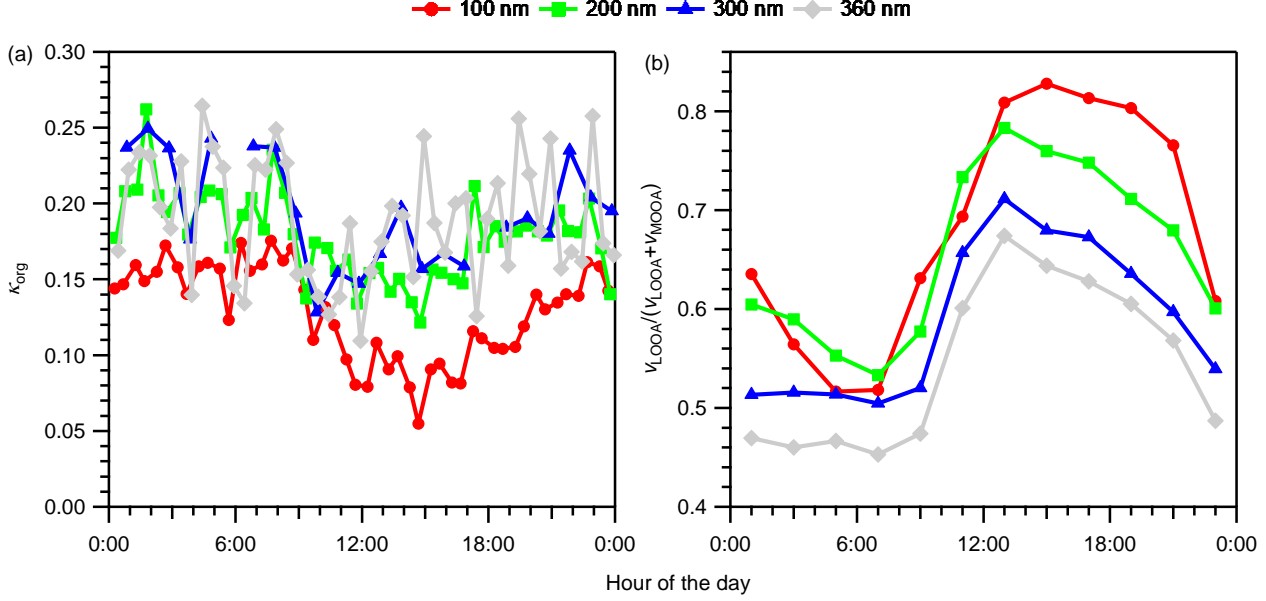



**Figure 4: Diurnal variations of (a) size-resolved hygroscopicity of OA ($\kappa_{org}$) and (b) size-resolved volume fractions of LOOA in OA ($v_{LOOA}/(v_{LOOA}+v_{MOOA})$)) for the entire study period. For $\kappa_{org}$ in panel (a), only data with $\varepsilon_{org} > 0.40$ were considered. The values in panel (b) were calculated from the diurnal variations of the average volume concentrations of LOOA and MOOA (Fig. S16), not from the averages of $v_{LOOA}/(v_{LOOA}+v_{MOOA})$.**

The hygroscopicity parameters of LOOA ($\kappa_{LOOA}$) and MOOA ($\kappa_{MOOA}$) were determined to evaluate the variations of $\kappa_{org}$ that can be explained by the relative contributions of LOOA and MOOA to OA. For the determination of $\kappa_{LOOA}$ and $\kappa_{MOOA}$, $\kappa_{org}$ was plotted against $v_{LOOA}/(v_{LOOA}+v_{MOOA})$ and their correlation was analyzed based on linear regression analysis (Figs. 5 and S14). For particles with $d_{dry}$ of 100, 200, 300, and 360 nm, the correlation coefficients between $\kappa_{org}$ and $v_{LOOA}/(v_{LOOA}+v_{MOOA})$ were −0.50, −0.58, −0.27, and −0.099, respectively. Relatively high correlations were observed for particles with $d_{dry}$ of 100

and 200 nm, probably because higher particle number concentrations (Fig. 1c) and higher OA volume fractions (Fig. 3) led to smaller uncertainties in the derived $\kappa_{org}$ in those diameter ranges, than in those of 300 and 360 nm. The regression line for particles with $d_{dry}$ of both 100 and 200 nm (Fig. 5) were used to derive $\kappa_{LOOA}$ and $\kappa_{MOOA}$ by applying $v_{LOOA}/(v_{LOOA}+v_{MOOA})$ of zero and unity to the obtained regression equation, respectively. The derived $\kappa_{LOOA}$ and $\kappa_{MOOA}$ were 0.083 and 0.28, respectively. This result is in between the results if particles with $d_{dry}$ of only 100 nm (derived $\kappa_{LOOA}$ and $\kappa_{MOOA}$ were 0.060

and 0.25, respectively) and only 200 nm (derived $\kappa_{LOOA}$ and $\kappa_{MOOA}$ were 0.095 and 0.34, respectively) were used. Compared with the $\kappa$ of PMF factors reported by Jimenez et al. (2009), the derived $\kappa_{LOOA}$ and $\kappa_{MOOA}$ are within the ranges of $\kappa$ for semi-volatile oxygenated OA (0.04–0.18) and low-volatility oxygenated OA (0.18–0.35), respectively (Fig. 5). The size-resolved $v_{LOOA}/(v_{LOOA}+v_{MOOA})$ and the above derived $\kappa_{LOOA}$ and $\kappa_{MOOA}$ were used to reconstruct the diurnal variation data and mean values of size-resolved $\kappa_{org}$ during 1200–2000 JST based on the volume additivity assumption. The reconstructed $\kappa_{org}$ and the

$\kappa_{org}$ described above (measured $\kappa_{org}$) were compared and their correlations were evaluated through linear regression analysis (Fig. S15). The slope and $r^2$ of the regression line are used to assess the ability of LOOA and MOOA to explain the variations of $\kappa_{org}$. For 100 and 200 nm particles, the relative contribution of LOOA and MOOA to OA can explain the majority of the variations of $\kappa_{org}$ (the slope and $r^2$ for 100 nm particles were 0.72 and 0.79, respectively and those for 200 nm particles were 0.63 and 0.68, respectively). The variations of $\kappa_{org}$ of 300 and 360 nm particles are explained less by the relative contributions

of LOOA and MOOA to OA (the slope and $r^2$ for 300 nm particles were 0.31 and 0.48, respectively; for 360 nm particles were 0.16 and 0.07, respectively). The slope and $r^2$ of the regression line between reconstructed and measured $\kappa_{org}$ for particles with




all four $d_{dry}$ were 0.39 and 0.44, respectively. This result indicates that the relative contribution of LOOA and MOOA can explain around 40 % of the observed diurnal variations of $\kappa_{org}$. The slope and $r^2$ of the regression line over the mean reconstructed and observed $\kappa_{org}$ of the four different sizes during afternoon hours (1200–2000 JST) were 0.39 and 0.84, respectively, which indicates that the size-dependence of $\kappa_{org}$ is explained by the relative contribution of LOOA and MOOA to OA by at least ~ 40 % during the time period.

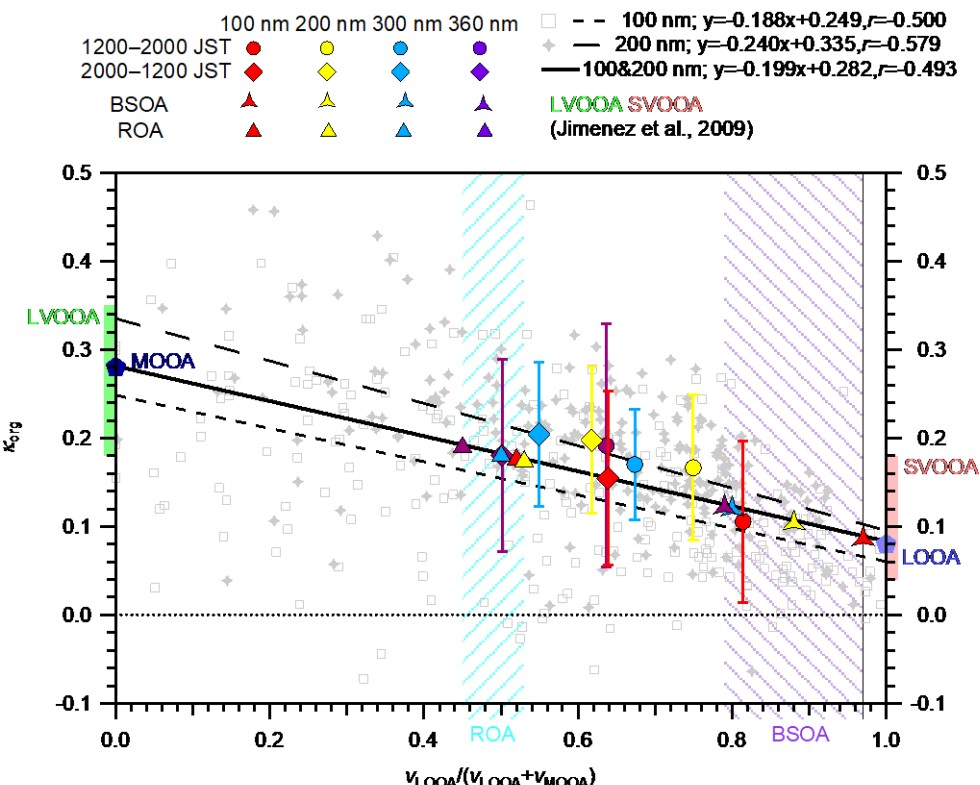

**Figure 5:** $\kappa_{org}$ versus $\nu_{LOOA}/(\nu_{LOOA}+\nu_{MOOA})$ for particles with $d_{dry}$ of 100 nm (gray open squares) and 200 nm (gray cross markers) over the entire study period. The time resolution of individual data was 2 h. Only data with $\varepsilon_{org} > 0.40$ were considered. The short-dashed, long-dashed, and solid lines are the regression lines for particles with $d_{dry}$ of 100 nm, 200 nm, and the sum of the particles with the two sizes, respectively. The $\kappa$ values of LOOA and MOOA derived from the regression lines (Sect. 4.2.1) are indicated by the light and dark blue pentagons, respectively. The size-resolved mean $\kappa_{org}$ during 1200–2000 JST and 2000–1200 JST are indicated as filled circles and diamond markers, respectively; the diameters were differentiated by colors. The whiskers represent the standard deviations of $\kappa_{org}$; the standard deviations of $\nu_{LOOA}/(\nu_{LOOA}+\nu_{MOOA})$ were presented in Table S6. The size-resolved $\kappa$ values of BSOA and ROA are indicated by the three-pointed stars and triangles, respectively; the diameters were differentiated by colors. The ranges of $\kappa$ for low-volatility oxygenated OA (LVOOA) and semi-volatile oxygenated OA (SVOOA) from Jimenez et al. (2009) are superimposed on the left and right axes, respectively. The shaded areas represent the estimated ranges of $\nu_{LOOA}/(\nu_{LOOA}+\nu_{MOOA})$ for ROA (left slash pattern) and BSOA (right slash pattern) of 100–360 nm particles.



### 4.2.2 Hygroscopicity of biogenic secondary organic aerosols

The hygroscopicity parameter of freshly formed BSOA ($\kappa_{\mathrm{BSOA}}$) was calculated as the volume weighted mean of $\kappa_{\mathrm{LOOA}}$ and $\kappa_{\mathrm{MOOA}}$. As discussed in Sect. 4.1.1, the enhanced OA mass in the daytime can be regarded as fresh BSOA. To simplify the analysis, the remaining part of the observed OA can be regarded as regionally transported OA (ROA), which may contain some aged, locally formed BSOA (Deng et al., 2018). To estimate the size-resolved contributions of LOOA and MOOA to BSOA and ROA, the size-resolved diurnal variations of sulfate was assumed as a tracer of regionally transported aerosol, and was scaled to represent the diurnal variations of LOOA and MOOA that constitute ROA (LOOA-ROA and MOOA-ROA; Fig. S16). For the scaling, the period of 0600–0800 JST, when OA and its subcomponents reached their daily minima (Figs. 2 and S16), was regarded as the background period, and all the LOOA and MOOA during the period were considered constituents of ROA. The remaining fractions of LOOA and MOOA were regarded as constituents of BSOA, referred to as LOOA-BSOA and MOOA-BSOA, respectively. The $v_{\mathrm{LOOA}}/(v_{\mathrm{LOOA}}+v_{\mathrm{MOOA}})$ of ROA was estimated to be 0.52, 0.53, 0.50, and 0.45 for particles with $d_{\mathrm{dry}}$ of 100, 200, 300, and 360 nm, respectively (the range is presented by a left-slash pattern in Fig. 5). The period when the diurnal variations of the size-resolved concentration of LOOA reached their maxima (i.e., 1400–1600 JST; Fig. S16) was chosen to estimate the size-resolved $v_{\mathrm{LOOA}}/(v_{\mathrm{LOOA}}+v_{\mathrm{MOOA}})$ of BSOA. The estimated $v_{\mathrm{LOOA}}/(v_{\mathrm{LOOA}}+v_{\mathrm{MOOA}})$ of the BSOA were 0.97, 0.88, 0.80, and 0.79 for particles with $d_{\mathrm{dry}}$ of 100, 200, 300, and 360 nm, respectively (the range is presented by a right-slash pattern in Fig. 5). Although the estimated BSOA could have aged to some extent, it was defined as fresh BSOA. The $\kappa_{\mathrm{BSOA}}$ (and $\kappa_{\mathrm{ROA}}$) were calculated using the derived $v_{\mathrm{LOOA}}/(v_{\mathrm{LOOA}}+v_{\mathrm{MOOA}})$ of BSOA (ROA) for particles with different $d_{\mathrm{dry}}$ and $\kappa_{\mathrm{LOOA}}$ and $\kappa_{\mathrm{MOOA}}$, and found to be 0.089 (0.18), 0.11 (0.18), 0.12 (0.18), and 0.12 (0.19) for particles with $d_{\mathrm{dry}}$ of 100, 200, 300, and 360 nm, respectively. The result indicates that $\kappa_{\mathrm{BSOA}}$ may increase with increase of the particle diameter as a result of the size-dependent contribution of LOOA and MOOA to BSOA (Fig. 5; colored three-pointed stars), which however needs to be confirmed by further studies. The size-dependence of the estimated $\kappa_{\mathrm{ROA}}$ (Fig. 5; colored triangles) was less obvious than that of $\kappa_{\mathrm{BSOA}}$. The $\kappa_{\mathrm{BSOA}}$ derived at 85 % RH for particles with $d_{\mathrm{dry}}$ of 100 nm in this study (0.089) was slightly smaller than that in a previous study for particles with similar diameters under SUPS condition (0.10 at 94±11 nm) at the same site (Deng

et al., 2018). The derived $\kappa_{ROA}$ is similar to the average $\kappa_{org}$ during the nighttime (Fig. 4). The size-resolved volume concentrations of BSOA ($v_{BSOA}$) and ROA ($v_{ROA}$) were also estimated using those size-resolved $v_{LOOA}/(v_{LOOA}+v_{MOOA})$ values of BSOA and ROA (Text S10). The obtained volume fraction of BSOA in aerosol particles ($\varepsilon_{BSOA}$; Fig. S18) presented diurnal variation patterns that were similar to those of $v_{LOOA}/(v_{LOOA}+v_{MOOA})$. Furthermore, the size dependence of $\varepsilon_{BSOA}$ during

afternoon hours was also similar to that of $v_{LOOA}/(v_{LOOA}+v_{MOOA})$. Because both the volume concentrations and hygroscopicity of BSOA and ROA were derived from that of LOOA and MOOA, the variation of the relative contributions of the estimated BSOA and ROA to OA can explain 40 % of the diurnal variation and size-dependence of the measured $\kappa_{org}$.

## 4.3 Ranges of the variations of $\kappa_t$, $\kappa_{org}$, and $\kappa_{BSOA}$

The ranges of the diurnal variations of $\kappa_t$ and $\kappa_{org}$, the difference between their maxima and minima, were obtained from their diurnal variation data with 2 h resolution (Table S8). The variation ranges of $\kappa_t$ were 0.14, 0.091, 0.084, 0.10, 0.11, 0.11, and 0.070 for particles with $d_{dry}$ of 30, 50, 70, 100, 200, 300, and 360 nm, respectively. The variation ranges of $\kappa_{org}$ for particles with $d_{dry}$ of 100, 200, 300, and 360 nm were 0.091, 0.079, 0.096, and 0.11, respectively. The size-dependence of $\kappa_t$ and $\kappa_{org}$ were quantified by the mean $\kappa_t$ and $\kappa_{org}$ values for the entire study period (Table S8). The difference of $\kappa_t$ between particles

with $d_{dry}$ of 100 and 300 nm was 0.13, and that of $\kappa_{org}$ was 0.056.

The ranges of both the diurnal variations and the size-dependence of $\kappa_t$ are similar to those reported from a previous study at the same site in 2010 (the mean of the differences between 0900–2100 JST and 2100–0900 JST on NPF event days and on nonevent days for the $d_{dry}$ range of 28.9–359 nm was in the range of 0.09–0.13, and the difference between Aitken mode and accumulation mode particles was 0.12) (Kawana et al., 2017), which pointed out the importance of the variation of particle

hygroscopicity with time and size to the CCN number concentration. The ranges of both the diurnal variations and the size-dependence of $\kappa_{org}$ are comparable to the range of 0.05 (from 0.08 to 0.13) that could lead to 30 % or more bias in the predicted CCN number concentration if not considered (Mei et al., 2013b). Here, only the ranges of the variation of $\kappa_t$ and $\kappa_{org}$ are discussed; other factors such as the absolute values of $\kappa_t$ or $\kappa_{org}$ should also be important to the prediction of CCN number concentrations. If SS is 0.1–1 %, typical maximum values in cloud systems (Farmer et al., 2015), the $d_{dry}$ of 100 nm is close

to the mode diameters of the CCN number-size distributions in previous studies at the same site (Kawana et al., 2017; Deng et al., 2018). Whereas the $d_{dry}$ of 300 nm is close to the mode mobility diameters of the mass-size distributions of OA and other aerosol components in this study (Fig. S2). The difference in the two types of mode diameters indicates that significant bias could be introduced if the bulk aerosol composition and/or OA composition is used for the prediction of CCN number

concentrations.

The difference of the estimated $\kappa_{BSOA}$ between particles with $d_{dry}$ of 100 nm and 300 nm was estimated to be 0.031. The difference implies the importance of the size-dependence of $\kappa_{BSOA}$ in the prediction of the contribution of BSOA to the CCN number concentration.

### 4.4 Contributions of OA and BSOA to CCN concentrations

The contributions of OA and BSOA to CCN number concentrations were assessed from the viewpoint of their contributions to the aerosol water uptake, which are size-dependent. For the estimate, the observed aerosols were assumed to be internally mixed. This is supported by a result from a previous study at the observation site: there was almost no difference in the prediction of the number fractions of CCN between the use of time- and size-resolved $g_f$ distributions and time- and size-resolved $g_{f,m}$ (Kawana et al., 2017). In two previous observations at the same site, the average CCN activation diameters of

aerosols under 0.41 and 0.42 % SS were 71 and 68 nm, respectively (Kawana et al., 2017; Deng et al., 2018). Based on these facts, all the particles with $d_{dry}$ greater than 70 nm were assumed to be CCN active. The estimated total CCN number concentration is referred to as $N_{CCN,t}$. For each size range (Text S11), the contribution of OA (BSOA) to the aerosol water uptake was represented as the product of the volume fraction of OA (BSOA) and $\kappa_{org}$ ($\kappa_{BSOA}$) divided by $\kappa_t$ [i.e., $\varepsilon_{org}\kappa_{org}/\kappa_t$ ($\varepsilon_{BSOA}\kappa_{BSOA}/\kappa_t$)], and was used to represent the fractional contribution of OA (and BSOA) to $dN_{CN}/dlogd_{dry}$ in the size range.

The fractional contribution of OA (and BSOA) to $N_{CCN,t}$, hereafter referred to as $F_{CCN,OA}$ ($F_{CCN,BSOA}$), was derived by integrating the product of $dN_{CN}/dlogd_{dry}$ and $\varepsilon_{org}\kappa_{org}/\kappa_t$ ($\varepsilon_{BSOA}\kappa_{BSOA}/\kappa_t$) above the CCN activation diameter (70 nm) and by dividing the obtained value by $N_{CCN,t}$. Details for the estimation of $F_{CCN,OA}$ and $F_{CCN,BSOA}$ are presented in Text S11.





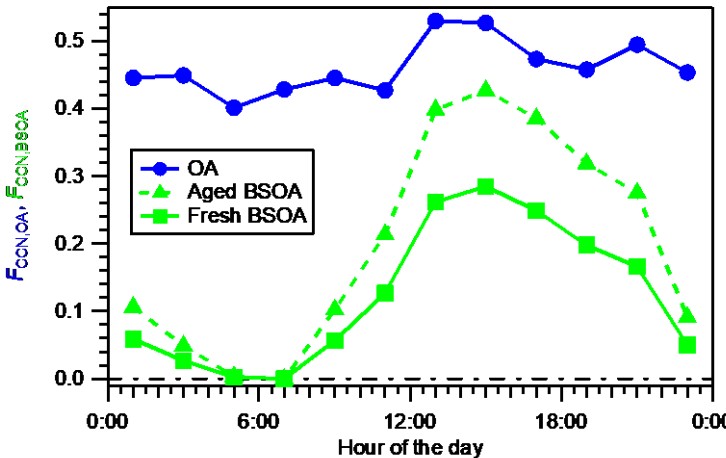

**Figure 6: Diurnal variation of $F_{CCN,OA}$ estimated using time- and size-resolved $\kappa_{org}$, and of $F_{CCN,BSOA}$ estimated using size-resolved $\kappa_{BSOA}$ and $\kappa_{ROA}$ (the assumed $\kappa$ values of aged BSOA).**

5    The diurnal variation of $F_{CCN,OA}$ estimated using time- and size-resolved $\kappa_{org}$ and that of $F_{CCN,BSOA}$ using size-resolved $k_{BSOA}$

are presented in Fig. 6. Both $F_{CCN,OA}$ and $F_{CCN,BSOA}$ reached their maxima during 1200–1600 JST, when intensive BSOA

formation was observed. The magnitude of the variation of $F_{CCN,BSOA}$ (from 0.00 to 0.28) was larger than that of $F_{CCN,OA}$ (from

0.40 to 0.53). This is explained by the larger magnitude of the diurnal variation range of $\varepsilon_{BSOA}$ (Fig. S18), compared to that of

$\varepsilon_{org}$ (Fig. 3). The $F_{CCN,BSOA}$ of 0.28 during 1200–1600 JST indicates a significant contribution of BSOA to the CCN number

10    concentration.

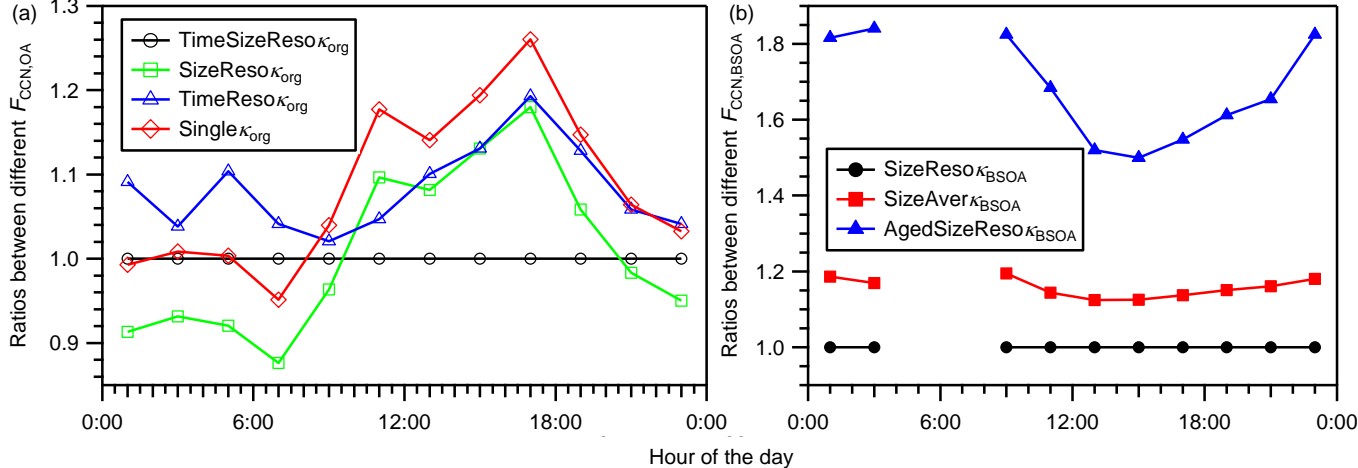





**Figure 7: (a) Diurnal variation of the ratios of the $F_{CCN,OA}$ derived using time- and size-resolved $\kappa_{org}$ (TimeSizeReso$\kappa_{org}$), time-averaged and size-resolved $\kappa_{org}$ (SizeReso$\kappa_{org}$), size-averaged and time-resolved $\kappa_{org}$ (TimeReso$\kappa_{org}$), and time- and size-averaged $\kappa_{org}$ (Single$\kappa_{org}$) to that derived using the time- and size- resolved $\kappa_{org}$ (TimeSizeReso$\kappa_{org}$). (b) Diurnal variation of the ratios of the $F_{CCN,BSOA}$ derived using size-resolved $\kappa_{BSOA}$ (SizeReso$\kappa_{BSOA}$), size-averaged $\kappa_{BSOA}$ (SizeAver$\kappa_{BSOA}$), and aged size-resolved $\kappa_{BSOA}$**
**(AgedSizeReso$\kappa_{BSOA}$) to that derived using the size-resolved $\kappa_{BSOA}$ (SizeReso$\kappa_{BSOA}$). In panel (b), the condition of aged size-resolved $\kappa_{BSOA}$ assumes that the value of $\kappa_{BSOA}$ equals that of $\kappa_{ROA}$, and the data during 0400–0800 JST, when the concentration of BSOA was low (volume concentration less than $0.01 \times 10^{-6}$ cm$^3$ m$^{-3}$; Fig. S18), are not presented (data are presented in Table S12).**

Because obvious diurnal variations and size-dependence of $\kappa_{org}$ were found and because $\kappa_{BSOA}$ was also estimated to be size-

dependent (Sect. 4.2), the sensitivities of the estimated $F_{CCN,OA}$ and $F_{CCN,BSOA}$ on the variations of $\kappa_{org}$ and $\kappa_{BSOA}$ were assessed.

To assess the influence of the variation of $\kappa_{org}$ on $F_{CCN,OA}$, the diurnal variations of $F_{CCN,OA}$ was estimated using (0a, base case)

time- and size-resolved $\kappa_{org}$, (1a) size-resolved, time-averaged (note that the average here refers to the arithmetic mean, it is

the same in other places of this paragraph) $\kappa_{org}$, (2a) time-resolved, size-averaged $\kappa_{org}$, and (3a) time- and size-averaged $\kappa_{org}$

(Table S10). Cases (1a)–(3a) were compared with the base case (0a), as presented in Fig. 7a. Using time-averaged $\kappa_{org}$ (case

1a), the $F_{CCN,OA}$ was overestimated by 18% during 1600–1800 JST and underestimated by 12 % during 0600–0800 JST. Using

size-averaged $\kappa_{org}$ (case 2a), the $F_{CCN,OA}$ was overestimated by 2–19 % on a diurnal basis. Using time- and size-averaged $\kappa_{org}$

(i.e., a single mean $\kappa_{org}$, case 3a), $F_{CCN,OA}$ was overestimated by 26 % during 1600–1800 JST and underestimated by 4.9 %

during 0600–0800 JST. The deviation of case (3a) from the base case (0a) resulted from the factors leading to the deviations

of cases (1a) and (2a). The magnitudes of the deviations, defined here as the difference between the lowest and highest values

of the ratios in Fig. 7a, for (1a), (2a), and (3a) are 30, 17, and 31 %, respectively. The substantial differences suggest that the

diurnal variations and size-dependence of $\kappa_{org}$ are important for accurate prediction of the contribution of OA to the CCN

number concentration in modelling studies. To assess the influence of the size-dependence of $\kappa_{BSOA}$ on $F_{CCN,BSOA}$, the diurnal

variations of $F_{CCN,BSOA}$ was estimated using (0b, base case) size-resolved $\kappa_{BSOA}$ and (1b) size-averaged $\kappa_{BSOA}$ (Table S11).

Case (1b) was compared with case (0b), as presented in Fig. 7b. Using size-averaged $\kappa_{BSOA}$ caused overestimation of $F_{CCN,BSOA}$

by 12–19 %, which relates to the decrease of the estimated $\kappa_{BSOA}$ and the increase of d$N_{CN}$/dlog$d_{dry}$ (Fig. S17) with decrease

in the dry particle diameter.

Furthermore, because the BSOA accumulated in the daytime probably aged as nighttime approached (Sect. 4.1.1), the influence

of the aging of the estimated fresh BSOA (assuming $\kappa_{BSOA}$ was as large as that of $\kappa_{ROA}$ (Table S11)) on $F_{CCN,BSOA}$ was also

evaluated. Here, the estimation of $F_{\mathrm{CCN,BSOA}}$ in the aged condition ignored the possible change in the CCN activation diameter accompanying the aging processes (Text S12). The possible change in the aerosol size distribution accompanying the aging process was also not considered here. Aged BSOA can contribute more to the aerosol water uptake and thus to the CCN number concentration. Assuming that the BSOA was as aged as ROA, the estimated $F_{\mathrm{CCN,BSOA}}$ was 50–84% larger than that

estimated assuming fresh BSOA (Fig. 7b), and it could have been 0.43 during 1400–1600 JST (Fig. 6). The result suggests that, whereas the contribution of BSOA to CCN was substantial at the study site, the magnitude of the contribution might be increased substantially by aging of the BSOA during transport after its formation in the forest.

## 5 Summary and conclusions

The size-resolved hygroscopicity at 85 % RH, chemical composition, and number-size distributions of atmospheric aerosols

were observed at a forest site in Wakayama, Japan in August and September, 2015. The diurnal variation and size-dependence in the hygroscopicity of the observed aerosol and organic aerosol components (OA) were discussed in view of the formation of BSOA. The fractional contributions of OA and BSOA to the total CCN number concentration were discussed in view of the variations of the hygroscopicity parameter of OA and BSOA.

Similar to two previous observations at the same site (Han et al., 2013, 2014; Kawana et al., 2017; Deng et al., 2018), OA was

the dominant sub-micrometer aerosol component, followed by sulfate. While the mass concentration of sulfate, on average, did not vary much in a day, the mass concentration of OA increased substantially in the afternoon hours, which was presumably explained by the condensation of BSOA. The hygroscopicity of ambient aerosol ($\kappa_\mathrm{t}$) and of OA ($\kappa_\mathrm{org}$) increased with increase of the dry particle diameter and presented daily minima in the afternoon hours. In this study, the ranges of the diurnal variations of $\kappa_\mathrm{org}$ of 100–360 nm particles were 0.079–0.11 and the $\kappa_\mathrm{org}$ of 300 nm particles was 0.056 larger than that of the 100 nm

particles. The diurnal variations and size-dependence of $\kappa_\mathrm{t}$ can be explained by the relative contributions of OA and inorganic salts in the observed aerosol. The relative contributions of the estimated fresh BSOA and regional OA can explain 40 % of the diurnal variation and size-dependence of $\kappa_\mathrm{org}$. The hygroscopicity of fresh BSOA ($\kappa_\mathrm{BSOA}$) was estimated to increase (0.089–0.12) with increase of the dry particle diameter (100–300 nm).



The fractional contributions of OA and fresh BSOA to CCN number concentrations estimated from the viewpoint of their contributions to the water uptake by the aerosol were in the ranges 0.40–0.53 and 0.00–0.28, respectively. Compared with the use of time- and size-resolved $\kappa_{\text{org}}$, the use of time- and size-averaged $\kappa_{\text{org}}$ overestimated the contribution of OA to the CCN number concentration by up to 26 % (1600–1800 JST) and underestimated the contribution by up to 4.9 % (0600–0800 JST).

These results indicate the importance of the diurnal variations and size-dependence of $\kappa_{\text{org}}$ in the prediction of the contribution of OA to the CCN number concentration. The use of size-averaged $\kappa_{\text{BSOA}}$ overestimated the contribution of fresh BSOA to the CCN number concentration by 12–19 % compared with the use of size-resolved $\kappa_{\text{BSOA}}$. If aging of BSOA occurs, the contribution of fresh BSOA to the CCN number concentration could be increased by 50–84 %, and could have reached a high value of 0.43 during 1400–1600 JST.

This study revealed the large magnitude of the diurnal variation and size-dependence of $\kappa_{\text{org}}$ at the observation site under the influence of the formation of BSOA. Also revealed was the importance of the variation of $\kappa_{\text{org}}$ to the estimation of the contribution of OA to the CCN number concentration from the viewpoint of the size-resolved contribution of OA to the water uptake of aerosols. Because both the diurnal variation and size-dependence of $\kappa_{\text{org}}$ in the studied forest are different from those in some other forest environments (Cerully et al., 2015; Thalman et al., 2017), further studies on the variation of the

hygroscopicity of organics and on the contributions of OA and BSOA to the CCN concentrations should be performed in other forest environments. Furthermore, the size-dependence of the hygroscopicity of fresh BSOA estimated here should be confirmed by additional studies.

*Data availability.* All of the final derived data supporting the findings of this study are available in the article or in its supporting

information file.

*Author contributions.* MM and YD designed the experiments, and YD, HY, MM, HF, and TN performed them. YD analyzed the data with contributions from MM, TN, and KK. YD prepared the manuscript with contributions from MM, KK, and TN.

*Competing interests.* The authors declare that they have no conflict of interest.

*Acknowledgements.* We thank the faculty and staff of the Wakayama Forest Research Station, Field Science Education and

Research Center of Kyoto University, Japan, for the provision of the study site and the meteorological data. We thank Qingcai



Chen, Kouji Adachi, Yuuki Kuruma, Ryuji Fujimori, and Takayuki Yamasaki for their help in the field observation. We acknowledge Kazuma Aoki for the use of the 1λ-PSAP instrument. We acknowledge the NOAA Air Resources Laboratory (ARL) for providing the HYSPLIT transport and dispersion model. This study was supported in part by JSPS KAKENHI Grant Number JP26281007.

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
