# Peer review of "Diurnal variation and size-dependence of the hygroscopicity of organic aerosol at a forest site in Wakayama, Japan: their relationship to CCN concentrations"

_Atmospheric Chemistry and Physics, 2018_

## Referee Comment (RC1) · Anonymous Referee #1 · 19 Nov 2018

Overview: Deng et al. present a detailed characterization and analysis of organic aerosol contribution to aerosol particle hygroscopicity through measurements with a Humidity Tandem differential mobility analyzer (HTDMA), Aerosol Mass Spectrometer (AMS) and complementary measurements of black carbon and trace gas species in Wakayama, Japan. The site is one that is very well characterized by previous field campaigns and well described in the literature. This study combines positive matrix factorization (PMF) analysis with aerosol hygroscopicity measurements to understand the time and size – dependent variation of organic hygroscopicity on overall aerosol

hygroscopicity. I recommend publication of the study after addressing a few minor issues.

General Comments: In general, the discussion of biogenic secondary organic aerosol (aged and fresh BSOA) and the AMS volatility factors (LOOA and MOOA) seem disconnected from each other. As the paper transitions from the PMF analysis to a hygroscopicity based derivation of BSOA (section 4.3 to section 4.4) there doesn't not appear to be a clear transition of tying together of the two concepts or how/why they should or should not be connected. A clearer distinction and transition would be helpful. Specific Comments: Page 11 line 4: "observation" conveys a short time period or single time, where the measurements happened over the course of 20 days. A different description (measurement period, campaign, etc.) might be more appropriate. Page 13 line 8: prior or previous rather than former. Page 13 Figure 2: The O:C ratio seems to vary quite a lot for a value that is an average. What do the percentiles look like (similar to a box and whiskers plot)? This would probably help since the range of change in O:C really isn't that large (0.58 – 0.64). Page 14 Figure 3: Similar to the issue with figure 2, some of the data is very noisy at the 30 min bins. Specifically, the 30 nm has as much variability point to point as the range of other lines on the graph. Looking at the times series in the Supplementary information (Figure S12), this is because 30 nm also has the lowest data coverage and the 30 min bins do not afford high enough points per average. Either consider longer time bins or remove the 30 nm line from the panel. Page 22 Figure 6: It was not initially clear looking at this figure that the aged and fresh lines were different based on different analyses of the data. It wasn't clear why the shouldn't have added up to the OA line. Consider adding to the caption to allow the figure to stand alone better. Supplement Figure S14: If only the data in the 360 nm panel <0.4 is being used to fit the line, then the other point at 0.9 zooms the graph out and makes the fit look better than it really is (a line fit through a cloud of data points similar to the 300 nm panel). Also, with this graph, the negative korg values are non-real and must be the result of issues with the combination of the AMS data and the kappa values. Consider filters for removing these in quality control, or changing the

limits on the range of volume fractions of organics required to calculate korg (as you mentioned on page 9 line 10).

---

## Referee Comment (RC2) · Anonymous Referee #3 · 6 Dec 2018

The authors presented a comprehensive study of the hygroscopic properties of organic aerosols at a forest site in Wakayama, Japan using a HTDMA and an AMS. The hygroscopicity parameter of fresh biogenic secondary organic aerosols was estimated and its relationship to CCN concentration was also evaluated. The dataset is rich with substantial amount of information, however, the discussion is over spread that the major conclusion becomes blurry. The manuscript is acceptable for publication in ACP after the following concerns are clearly addressed.

Major comments: In general, the definition or quantification of BSOA and ROA should be clearly clarified and highlighted with proper references in your manuscript, as most of your discussion is based on this assumption. I suggest to make an individual section introducing it rather a few lines, for instance Page 19, line 9-15. Similarly, in your TextS9, you said 'The diurnal variation data on the mass concentration of BC was scaled to represent the diurnal variation of non-BSOA-OA'. How did you prove your method is valid, any references? As I see, there is big uncertainty within the estimation of BSOA-OA concentration from this method, which you used further to calculate their CCN contribution. Please carefully clarify. Also, I see you occasionally have BC peaks, correlated with high CO concentration. You might have biomass burning organic aerosols, how did you deal with those or did you filter their contribution or should we neglect their contribution? Please discuss.

Your RH values are pretty high, which means supersaturation conditions might be possible reached in the real atmosphere. This indicates that your particles, especially large ones (larger than 300 nm) might already activate under supersaturation and then lose water again due to evaporation after RH decreases. This process will strongly affect your results, did you consider this into your discussion.

The correlation between $\kappa_{org}$ and $\nu LOOA/(\nu LOOA+\nu MOOA)$ is not high, and you used this relation to derive $\kappa LOOA$ and $\kappa MOOA$, which may introduce even higher uncertainties. I think your analysis should start from the closure between measured $\kappa$ and ZSR_derived $\kappa$. You can replace $\kappa_{org}$ with $\kappa LOOA$ and $\kappa MOOA$, and ask your computer to find the best solution for $\kappa LOOA$ and $\kappa MOOA$ and to see if these values are different from those of your current analysis. In addition, I don't understand those error bars in your Fig. 5.

I am not sure about your Section 4.4. You said 'your particles larger than 70 nm are assumed to be CCN active', which means you neglected the effect from the chemical composition. Then you started to consider the effect from chemical composition by dividing the spectrum with BSOA-contribution and contributions from other components, see Page 21, line 17-22. To me, this is a little bit in conflict with each other. Secondly, your Fig. S17 are actually based on external mixing state assumption. For your internal mixing aerosol population, particles are having quite similar chemical composition. I don't see the point that how could BSOA contributes to CCN concentration alone. The logic behind it as I see is the involving of BSOA into organic

aerosols will change the hygroscopicity parameter $\kappa$, then influence the critical diameter of particles that are able to activate, for instance, not 70 nm anymore, which thus change the potential CCN concentration. If this is true, then your method to derive the contribution of BSOA to CCN concentration is not sound or at least with huge uncertainties. Please carefully clarify. Mei et al., (2013b), who you cited in your introduction, gave a proper way to calculate the CCN concentration due to an elevated $\kappa_{org}$ in their section 5.2.

---

## Author Response (AR1)

We appreciate valuable comments from the reviewers. Our answers to the comments are provided below. (The reviewers' comments are written in italic. The line numbers in the response are from the revised version of the manuscript.)

**Response to Anonymous Referee #1**

*Anonymous Referee #1*

*Overview: Deng et al. present a detailed characterization and analysis of organic aerosol contribution to aerosol particle hygroscopicity through measurements with a Humidity Tandem differential mobility analyzer (HTDMA), Aerosol Mass Spectrometer (AMS) and complementary measurements of black carbon and trace gas species in Wakayama, Japan. The site is one that is very well characterized by previous field campaigns and well described in the literature. This study combines positive matrix factorization (PMF) analysis with aerosol hygroscopicity measurements to understand the time and size – dependent variation of organic hygroscopicity on overall aerosol hygroscopicity. I recommend publication of the study after addressing a few minor issues.*

*General Comments: In general, the discussion of biogenic secondary organic aerosol (aged and fresh BSOA) and the AMS volatility factors (LOOA and MOOA) seem disconnected from each other. As the paper transitions from the PMF analysis to a hygroscopicity based derivation of BSOA (section 4.3 to section 4.4) there doesn't not appear to be a clear transition of tying together of the two concepts or how/why they should or should not be connected. A clearer distinction and transition would be helpful.*

In this study, the fresh BSOA was defined as the enhanced mass of both LOOA and MOOA in the daytime (Sect. 4.2.2). The existence of aged BSOA at the studied site is only briefly explained from the diurnal variation of O:C ratio and MOOA (page 12 lines 16–17), which is, however, not the main point of this study. The discussion on the fractional contribution of aged BSOA to the CCN number concentration is based on a hypothetical condition of BSOA after its transport (the last paragraph in Sect. 4.4), and is not related to the observed relative abundances of LOOA and MOOA. To clarify this

point, the first sentence in the last paragraph of Sect. 4.4 has been modified to: "Furthermore, because fresh BSOA probably become aged after atmospheric transport, the influence of the aging of the estimated fresh BSOA (assuming $\kappa_{BSOA}$ was as large as that of $\kappa_{ROA}$ (Table S11)) on $F_{CCN,BSOA}$ was also evaluated."

Note that LOOA and MOOA in the manuscript are not volatility factors. They were defined according to the degree of oxygenation (i.e., O:C ratio), as indicated in page 9 lines 18–20: "A two-factor PMF solution was adopted, which resolved two oxygenated OA factors: one with a lower atomic O:C ratio (0.47) named less-oxygenated organic aerosol (LOOA), and the other with a higher O:C ratio (0.95) named more-oxygenated organic aerosol (MOOA)."

*Specific Comments:*
*Page 11 line 4: "observation" conveys a short time period or single time, where the measurements happened over the course of 20 days. A different description (measurement period, campaign, etc.) might be more appropriate.*

The word "observation" has been changed to "measurement period with effective data". (page 11 line 4)

*Page 13 line 8: prior or previous rather than former.*

The "former" has been changed to "prior". (page 13 line 10)

*Page 13 Figure 2: The O:C ratio seems to vary quite a lot for a value that is an average. What do the percentiles look like (similar to a box and whiskers plot)? This would probably help since the range of change in O:C really isn't that large (0.58 – 0.64).*

A box and whiskers plot of the diurnal variation of O:C ratio has been added to the supplementary pdf file as Fig. S19, and a relating explanation has been added to the caption of Fig. 2c.

Figure S19 and its caption are as follows:

[Figure]

**Figure S19:** Box and whiskers plot of the diurnal variation of the O:C ratios of bulk OA (only data with $m_{org} > 0.3$ µg m$^{-3}$ are included) for the entire study period. The horizontal lines in the boxes indicate the median values, boundaries of the boxes indicate the 25th- and 75th-percentiles, and the whiskers indicate the highest and lowest values. The cross symbols in the boxes indicate the mean values.

The caption of Fig. 2c is now as follows: "... (c) LOOA, MOOA, and residual, and the O:C ratio of bulk OA (only data with $m_{org} > 0.3$ µg m$^{-3}$ are included) averaged for the entire study period. (A box and whiskers plot of the diurnal variation of O:C ratio is presented in Fig. S19.)" (page 13 lines 4–6)

*Page 14 Figure 3: Similar to the issue with figure 2, some of the data is very noisy at the 30 min bins. Specifically, the 30 nm has as much variability point to point as the range of other lines on the graph. Looking at the times series in the Supplementary information (Figure S12), this is because 30 nm also has the lowest data coverage and the 30 min bins do not afford high enough points per average. Either consider longer time bins or remove the 30 nm line from the panel.*

The diurnal variation of $\kappa_t$ of 30 nm particles in Fig. 3a is now presented in 2-h time resolution. Further, a related explanation has been added to the end of the caption of Fig. 3: "Note that for particles with $d_{dry}$ of 30 nm, $\kappa_t$ is presented in 2-h time resolution because of the low data coverage (Fig. S12)." (page 15 lines 3–4)

*Page 22 Figure 6: It was not initially clear looking at this figure that the aged and*

*fresh lines were different based on different analyses of the data. It wasn't clear why they shouldn't have added up to the OA line. Consider adding to the caption to allow the figure to stand alone better.*

The caption of Fig. 6 has been modified to: "Diurnal variation of the fractional contribution of OA to the total CCN number concentration ($F_{\text{CCN,OA}}$) estimated using time- and size-resolved $\kappa_{\text{org}}$, and diurnal variation of the fractional contribution of BSOA to the total CCN number concentration ($F_{\text{CCN,BSOA}}$) estimated assuming fresh BSOA (using size-resolved $\kappa_{\text{BSOA}}$) and aged BSOA (using size-resolved $\kappa_{\text{ROA}}$)." (page 22 lines 2–4)

*Supplement Figure S14: If only the data in the 360 nm panel <0.4 is being used to fit the line, then the other point at 0.9 zooms the graph out and makes the fit look better than it really is (a line fit through a cloud of data points similar to the 300 nm panel). Also, with this graph, the negative korg values are non-real and must be the result of issues with the combination of the AMS data and the kappa values. Consider filters for removing these in quality control, or changing the limits on the range of volume fractions of organics required to calculate korg (as you mentioned on page 9 line 10).*

We have applied filters to the observed data (Text S4) and to the data used for the derivation of $\kappa_{\text{org}}$ (page 15 lines10–11) in the ACPD manuscript. Using stricter filters to rule out the large $\kappa_{\text{org}}$ in the 360 nm panel in Fig. S14 and/or the negative $\kappa_{\text{org}}$ values will result in loss of data that are likely real. For example, to omit all negative $\kappa_{\text{org}}$ values in Fig. S14b, data points with $\varepsilon_{\text{org}}$ smaller than 0.89 must be excluded. In addition, although the volume additivity assumption used for the derivation of $\kappa_{\text{org}}$ in general holds well (page 9 lines 11–14), we should not rule out the possibility that the approximation of the additivity assumption of $\kappa$ results in some negative $\kappa_{\text{org}}$ as "apparent" $\kappa$ values. Hence, no modification has been made to this point.

**Response to Anonymous Referee #3**

*Anonymous Referee #3*

*The authors presented a comprehensive study of the hygroscopic properties of organic aerosols at a forest site in Wakayama, Japan using a HTDMA and an AMS. The hygroscopicity parameter of fresh biogenic secondary organic aerosols was estimated and its relationship to CCN concentration was also evaluated. The dataset is rich with substantial amount of information, however, the discussion is over spread that the major conclusion becomes blurry. The manuscript is acceptable for publication in ACP after the following concerns are clearly addressed.*

*Major comments: In general, the definition or quantification of BSOA and ROA should be clearly clarified and highlighted with proper references in your manuscript, as most of your discussion is based on this assumption. I suggest to make an individual section introducing it rather a few lines, for instance Page 19, line 9-15.*

BSOA is defined and quantified in Sect. 4.2.2, in association with the derivation of its hygroscopicity. ROA is also defined and quantified in the same section. Although it is also reasonable to make an individual section to define and quantify BSOA and ROA, we have left the original structure to emphasize the characteristics of the hygroscopicity of OA and BSOA and of their contributions to CCN concentrations. Thus, no modification has been made to this point.

*Similarly, in your TextS9, you said 'The diurnal variation data on the mass concentration of BC was scaled to represent the diurnal variation of non-BSOA-OA'. How did you prove your method is valid, any references? As I see, there is big uncertainty within the estimation of BSOA-OA concentration from this method, which you used further to calculate their CCN contribution. Please carefully clarify. Also, I see you occasionally have BC peaks, correlated with high CO concentration. You might have biomass burning organic aerosols, how did you deal with those or did you filter their contribution or should we neglect their contribution? Please discuss.*

Although we have not found a reference for our method to assess non-BSOA-OA, we regard that the method is appropriate for the purpose of this study. Although a possible contribution of local biomass burning is in contradict to the assumption of the method, it must be small in the studied remote mountain area. The following Fig. R1 supports this idea. The mass fraction of fragment $C_2H_4O_2^+$, which is a tracer of biomass burning OA, was low (0.15%), as compared to that in a city site (0.62%) (Xu et al., 2015). Besides, $C_2H_4O_2^+$ correlated with OA ($R^2$: 0.95) more strongly than with BC ($R^2$: 0.72). The possible contribution of local anthropogenic pollution to BC was ruled out by using the screening explained in Text S4. No modification has been made concerning this point.

Note that BSOA used for the estimation of the fractional contribution of BSOA to CCN number concentrations was quantified using size-resolved data (Sect. 4.2.2), whereas BSOA-OA in Text S9 was derived from bulk OA mass concentrations.

[Figure]

**Figure R1: Time series of the mass concentrations of BC, OA, and fragment $C_2H_4O_2^+$.**

*Your RH values are pretty high, which means supersaturation conditions might be possible reached in the real atmosphere. This indicates that your particles, especially large ones (larger than 300 nm) might already activate under supersaturation and then lose water again due to evaporation after RH decreases. This process will strongly affect your results, did you consider this into your discussion.*

Possible in-cloud processes suggested from high ambient RH conditions could have changed the properties of the observed aerosol, for example by the aging of the freshly formed BSOA (Han et al., 2014). It is now discussed briefly as follows:

"The O:C ratio of OA increased slowly from around noon to midnight (Fig. 2c), together with the appearance of MOOA, indicating the aging of freshly formed BSOA (Han et al., 2014). Because of high RH conditions (Fig. S7), aqueous phase reactions including in-cloud processes could have played an important role in the aging of fresh BSOA (Han et al., 2014), which could have modified the hygroscopicity of ambient aerosols (Jimenez et al., 2009; Farmer et al., 2015)." (page 12 lines 15–19)

*The correlation between $\kappa_{org}$ and $v_{LOOA}/(v_{LOOA}+v_{MOOA})$ is not high, and you used this relation to derive $\kappa_{LOOA}$ and $\kappa_{MOOA}$, which may introduce even higher uncertainties. I think your analysis should start from the closure between measured $\kappa$ and ZSR_derived $\kappa$. You can replace $\kappa_{org}$ with $\kappa_{LOOA}$ and $\kappa_{MOOA}$, and ask your computer to find the best solution for $\kappa_{LOOA}$ and $\kappa_{MOOA}$ and to see if these values are different from those of your current analysis. In addition, I don't understand those error bars in your Fig. 5.*

The method recommended by the reviewer (referred to as ALT method) is in essence same as the one we used (referred to as ORIG method). Both of them are based on the volume additivity assumption (page 8 lines 13–15). With the ALT method, the derived $\kappa_{LOOA}$ and $\kappa_{MOOA}$ using data same as those in Fig. 5 are 0.090 and 0.23, which are 8.4% higher and 18% lower than the ones derived from the ORIG method. If these two values are applied for the derivation of $\kappa_{BSOA}$ and $\kappa_{ROA}$ (Table R1) and $F_{CCN,BSOA}$ (Table R2), the resulting differences are: the $\kappa_{BSOA}$ value for 100 nm particles (0.094) is 5.9% higher, whereas the changes for larger particles are negligible; $\kappa_{ROA}$ are 11–13% lower; using size-averaged $\kappa_{BSOA}$ overestimated $F_{CCN,BSOA}$ by 8–13% if compared to those using size-resolved $\kappa_{BSOA}$ (12–19% using the ORIG method (page 23 line 25)); $F_{CCN,BSOA}$ increased by 35–57% with the assumption of aged BSOA if compared to those under the condition of fresh BSOA (50–84% with ORIG method (page 24 line 4)). Because the main conclusion does not change, the derivation of $\kappa_{LOOA}$ and $\kappa_{MOOA}$ in the manuscript is not modified.

**Table R1: Size-resolved $\kappa_{BSOA}$ and $\kappa_{ROA}$ calculated from $\kappa_{LOOA}$ and $\kappa_{MOOA}$ derived using ALT method**

| $d_{dry}$ (nm) | $\kappa_{BSOA}$ | $\kappa_{ROA}$ |
|---|---|---|
| 100 | 0.094 | 0.16 |
| 200 | 0.11 | 0.16 |
| 300 | 0.12 | 0.16 |
| 360 | 0.12 | 0.17 |

**Table R2: Diurnal variation of the ratios of $F_{CCN,BSOA}$ derived using size-resolved $\kappa_{BSOA}$, size-averaged $\kappa_{BSOA}$, and size-resolved aged $\kappa_{BSOA}$ to that derived using size-resolved $\kappa_{BSOA}$[1]**

| Hour of day | Ratios of $F_{CCN,BSOA}$ from different assumptions | | |
|---|---|---|---|
| | Size-resolved | Size-averaged | Aged, Size-resolved |
| | $\kappa_{BSOA}$ | $\kappa_{BSOA}$ | $\kappa_{BSOA}$[2] |
| 0000–0200 JST | 1 | 1.13 | 1.56 |
| 0200–0400 JST | 1 | 1.12 | 1.57 |
| 0400–0600 JST[3] | 1 | 0.922 | 1.31 |
| 0600–0800 JST[3] | 1 | 1.92 | 2.76 |
| 0800–1000 JST | 1 | 1.13 | 1.56 |
| 1000–1200 JST | 1 | 1.10 | 1.48 |
| 1200–1400 JST | 1 | 1.08 | 1.36 |
| 1400–1600 JST | 1 | 1.08 | 1.35 |
| 1600–1800 JST | 1 | 1.09 | 1.38 |
| 1800–2000 JST | 1 | 1.10 | 1.43 |
| 2000–2200 JST | 1 | 1.11 | 1.45 |
| 2200–0000 JST | 1 | 1.12 | 1.56 |

[1]Both $\kappa_{BSOA}$ and $\kappa_{ROA}$ are from Table R1;

[2]The condition of size-resolved aged $\kappa_{BSOA}$ assumes that the value of $\kappa_{BSOA}$ equals that of $\kappa_{ROA}$;

[3]The concentration of BSOA was low (refer to the caption of Fig. 7).

The error bars in Fig. 5 represent the standard deviation of size-resolved $\kappa_{org}$ during 1200–2000 JST and 2000–1200 JST. This point was not clearly addressed in the original caption. It has been modified to: "… The size-resolved mean $\kappa_{org}$ during 1200–2000 JST and 2000–1200 JST are indicated as filled circles and diamond markers, respectively. The standard deviations of the mean $\kappa_{org}$ are indicated by the whiskers. The standard deviations of the mean $v_{LOOA}/(v_{LOOA}+v_{MOOA})$ are presented in Table S6. The size-resolved $\kappa$ values of BSOA and ROA are indicated by the three-pointed stars and triangles, respectively. The diameters of $\kappa_{org}$, $\kappa_{BSOA}$, and $\kappa_{ROA}$ are differentiated by colors. …" (page 18 lines 12–15)

*I am not sure about your Section 4.4. You said 'your particles larger than 70 nm are assumed to be CCN active', which means you neglected the effect from the chemical composition. Then you started to consider the effect from chemical composition by dividing the spectrum with BSOA-contribution and contributions from other components, see Page 21, line 17-22. To me, this is a little bit in conflict with each other. Secondly, your Fig. S17 are actually based on external mixing state assumption. For your internal mixing aerosol population, particles are having quite similar chemical composition. I don't see the point that how could BSOA contributes to CCN concentration alone. The logic behind it as I see is the involving of BSOA into organic aerosols will change the hygroscopicity parameter κ, then influence the critical diameter of particles that are able to activate, for instance, not 70 nm anymore, which thus change the potential CCN concentration. If this is true, then your method to derive the contribution of BSOA to CCN concentration is not sound or at least with huge uncertainties. Please carefully clarify. Mei et al., (2013b), who you cited in your introduction, gave a proper way to calculate the CCN concentration due to an elevated κorg in their section 5.2.*

The evaluation of the contribution of BSOA to the CCN number concentration was from the viewpoint of its contribution to the aerosol total water uptake (page 21 lines 10–11). For the evaluation, the aerosols were assumed to be internally-mixed in respective diameter ranges (page 14 line 2 and page 21 lines 11–14). Therefore, they have same critical activation diameters under certain water vapor supersaturation condition. Fig. S17 should be understood from the viewpoint of the water uptake fraction. To better address this point, the first sentence in the caption of Fig. S17 has been changed to: "Estimate of the contributions of BSOA to the CCN number concentration from the viewpoint of its size-resolved contribution to the aerosol water uptake."

The influence of the variation of CCN activation diameter on the prediction of $F_{CCN,OA}$ and $F_{CCN,BSOA}$ was not assessed in the original manuscript. It was now added as Text S12 (the original Text S12 is now Text S13) as follows:

**"Text S12. Assessment of the diurnal variation of the CCN activation diameter**

Although the variation of the CCN activation diameter with time influences the prediction of $F_{CCN,OA}$ and $F_{CCN,BSOA}$, the degree was found to be small. In the summertime observation in 2014 (Deng et al., 2018), the range of the diurnal variation of the CCN activation diameter was from 64 to 76 nm, whereas the CCN activation diameter assumed in this study is 70 nm. Applying 64 or 76 nm to an assumed CCN activation diameter results in the deviations of the predicted $F_{CCN,OA}$ and $F_{CCN,BSOA}$ only by −1.9–2.3 % and −3.1–3.8 %, respectively."

A corresponding explanation was also added to the end of the first paragraph of Sect. 4.4: "The diurnal variation of the CCN activation diameter was not considered for the estimate of $F_{CCN,OA}$ and $F_{CCN,BSOA}$ (Text S12)."

It is reasonable to use the method in Mei et al. (2013b) to calculate the CCN concentration contributed by BSOA, which was adopted in our previous paper (Deng et al., 2018). However, in this manuscript we instead assessed the fractional contribution of OA and BSOA to the CCN number concentration from the viewpoint of their contributions to the aerosol water uptake over the effective measurement period on a diurnal basis.

**Other minor changes:**

1) Fig. 6 and the corresponding data in Table S9, and Fig. 7 and the corresponding data in Table S12 have been corrected because the $\kappa_{ROA}$ at 100 nm was erroneously used as $\kappa_{BSOA}$ for all the four diameters with the assumption of size-resolved aged BSOA. The size-resolved $\kappa_{ROA}$ is now used instead.
2) The "0.41" in Text S12, which was a typo, has been corrected to "0.15".
3) The reference "Deng et al., 2018" in the reference list has been updated because the status of the paper has been updated.

4) JSPS KAKENHI JP18K19852 is now acknowledged.

5) Current affiliation of one of the authors has been updated.

6) Some minor changes that have no influence on the points of the manuscript are also made.

All changes can be found in the track-change version of the manuscript.

**References:**

Deng, Y. G., Kagami, S., Ogawa, S., Kawana, K., Nakayama, T., Kubodera, R., Adachi, K., Hussein, T., Miyazaki, Y., and Mochida, M.: Hygroscopicity of Organic Aerosols and Their Contributions to CCN Concentrations Over a Midlatitude Forest in Japan, Journal of Geophysical Research-Atmospheres, 123, 9703-9723, 10.1029/2017jd027292, 2018.

Han, Y. M., Iwamoto, Y., Nakayama, T., Kawamura, K., and Mochida, M.: Formation and evolution of biogenic secondary organic aerosol over a forest site in Japan, Journal of Geophysical Research-Atmospheres, 119, 259-273, 10.1002/2013jd020390, 2014.

Xu, W. Q., Sun, Y. L., Chen, C., Du, W., Han, T. T., Wang, Q. Q., Fu, P. Q., Wang, Z. F., Zhao, X. J., Zhou, L. B., Ji, D. S., Wang, P. C., and Worsnop, D. R.: Aerosol composition, oxidation properties, and sources in Beijing: results from the 2014 Asia-Pacific Economic Cooperation summit study, Atmospheric Chemistry and Physics, 15, 13681-13698, 10.5194/acp-15-13681-2015, 2015.

[revised manuscript text omitted]
 $1.13 \pm 0.08$, $1.21 \pm 0.09$, $1.22 \pm 0.09$, $1.26 \pm 0.10$, $1.36 \pm 0.10$, $1.40 \pm 0.08$, and $1.42 \pm 0.08$, respectively. The unimodal pattern of the $g_\text{f}$-PDF indicated the internal mixing state of the observed aerosol at the respective particle diameters. Decreases of $g_\text{f,m}$ (Fig. S12) were observed for all particles during periods of intensive BSOA formation (i.e., episodes when the mass concentration of OA especially LOOA greatly increased; such episodes were observed on 31 August, and on 2, 5, 7, 9, 14, 15, 17, 18, 19, 20, and 21 September; Fig. 1b).

The hygroscopicity parameter of ambient aerosol particles that corresponds to $g_\text{f,mw}$ ($\kappa_\text{t}$) also increased with the increase of aerosol particle diameters (Fig. 3a). Similar diurnal variation patterns were observed for all the diameters studied. The $\kappa_\text{t}$ started to decrease around 0800 JST, then reached daily minima between 1300 and 1900 JST. Then it increased continually until around 0200 JST of the next day, and remained high until 0800 JST the next morning. For particles with $d_\text{dry} \geq 100$ nm, the diurnal variation pattern and size-dependence of $\kappa_\text{t}$ were opposite to those of the volume fraction of OA (Fig. 3b) and were similar to those of the volume fraction of total inorganic salts (Fig. S13). The results suggest that, at least for ambient aerosol particles with $d_\text{dry} \geq 100$ nm, OA and inorganic salts had low and high hygroscopicity, respectively, resulted in the variations of $\kappa_\text{t}$. Although $\kappa_\text{inorgsalt}$ is much greater than $\kappa_\text{org}$ (Petters and Kreidenweis, 2007), the high $\varepsilon_\text{
[revised manuscript text omitted]

**Text S1. Performance check of the DMAs**

Before and after the atmospheric observation, the accuracy of the sizing by three DMAs was assessed using standard size PSL particles (JSR SIZE STANDARD PARTICLES: SC-0055-D, SC-0100-D, and SC-032-S; Thermo Scientific™: 3500A). The mode diameters from fittings for the measurement data (Kawana et al., 2014) were compared with the manufacturer warranty (Table S1), which is interpreted as prescribed ranges of mean diameter $\pm$ the expanded uncertainty ($k = 2$). The mode diameters after the atmospheric observation agreed with those before the observation within 0.84 %. The results obtained before the atmospheric observation are as follows. For DMA1, whereas the measured mode diameter of SC-0100-D was within the prescribed range, the measured mode diameters of SC-0055-D and SC-032-S were 1.0 % larger than the upper end of the prescribed range and 0.76 % lower than the lower end of the prescribed range, respectively. For DMA2, the measured mode diameter of SC-0055-D was 1.5 % larger than the upper end of the prescribed range, and the measured mode diameters of SC-0100-D, SC-032-S, and 3500A were 0.85, 3.2, and 2.2 % lower than the lower end of the prescribed ranges, respectively. For DMA3, the measured mode diameter of SC-0055-D was 6.9 % larger than the upper end of the prescribed diameter range, and the measured mode diameters for all three of the other PSL standards were within the prescribed ranges.

**Text S2. Performance check of HTDMA using ammonium sulfate particles**

Before the atmospheric observation, an aqueous solution of ammonium sulfate (AS) (99.999 % purity, Sigma-Aldrich) was nebulized and the generated aerosols were dried and introduced to the HTDMA to assess the difference in the sizing between the two DMAs under dry condition and to validate the control of RH in the HTDMA. The diameter setting for the measurements was the same as that of ambient particles (Sect. 2). The mean growth factors ($g_{f,m}$) of the AS particles under both dry and wet conditions were

retrieved using the same method as that for ambient particles (Sect. 3.1). The $g_{f,m}$ of the dry AS aerosol particles with diameters of 30–360 nm were 1.2–4.1 % deviated from unity (Table S2). The deviations were used to correct the difference of sizing between DMA1 and DMA2 for the $g_f$ of AS and ambient aerosol particles measured at 85 % RH. The respective $g_{f,m}$ of AS particles at 85 % RH with $d_{dry}$ of 30, 50, 70, 100, 200, 300, and 360 nm were 1.52, 1.54, 1.54, 1.54, 1.55, 1.57, and 1.59, which agree within 2.0 % with the calculated values ($g_{f,AS}$; Table S2) based on the Extended AIM Aerosol Thermodynamics Model II (E-AIM II, http://www.aim.env.uea.ac.uk/aim/model2/model2a.php; Clegg et al., 1998; Wexler and Clegg, 2002). The derivation of the hygroscopic growth factor of AS particles using the E-AIM II model is presented in Text S6.

**Text S3. PMF analysis of organic mass spectra**

The bulk mass spectra of organics observed in V-mode were subjected to PMF analysis (PMF Evaluation Tool v3.04A). For the analysis, high resolution fragment ions with signal to noise ratio (SNR) smaller than 0.2 were omitted, fragments with SNR in the range of 0.2–2 were down-weighted by a factor of three, and fragments related to $CO_2$ (i.e., $CO_2$, $CO$, $H_2O$, $HO$, and $O$) were down-weighted so that fragment $CO_2$ only contributed once. The obtained two-factor solution, with a more-oxygenated OA component (MOOA) and a less-oxygenated OA component (LOOA), with seed = 1 and fpeak = 0, was adopted for the explanation of OA composition. The two-factor PMF results are summarized in Fig. S1.

**Text S4. Data screening methods**

All the data except for the meteorological data obtained during the atmospheric observation were subjected to the screening to exclude data under possible influence from intermittent local anthropogenic emissions. For the SMPS data with 5 min resolution, and the BC and gaseous species data with 30 min resolution, if

the data value at one point was more than 30 % deviated from both the former data point and the next data point in the time series, the data at that time point was deleted. The chemical composition data derived from AMS measurements with 30 min resolution were deleted whenever the BC data were deleted. The hygroscopic growth data derived from the HTDMA measurements were deleted whenever the SMPS data were deleted. Furthermore, if the total count of particles measured using the CPC in the HTDMA in the diameter range of 0.80–2.2 times of $d_{dry}$ (or 0.80–2.0 times of $d_{dry}$ for particles with $d_{dry}$ of 360 nm) during a single scan was less than eight, the HTDMA data was not used, either. Here, the $d_{dry}$ were corrected for the difference of sizing between DMA1 and DMA2.

**Text S5. Derivation of size-resolved volume fractions of the chemical components**

The size-resolved volume fractions of inorganic salts ($\varepsilon_i$), organics ($\varepsilon_{org}$), and BC ($\varepsilon_{BC}$) were calculated as follows. First, BC was assumed to be internally mixed with non-refractory aerosol components and to have the same mass-size distribution as OA. The aerosol particles were assumed to be spherical and without voids. Using the PToF mode data from the AMS, the mass concentrations of aerosol components in the vacuum aerodynamic diameter ($d_{va}$) ranges that are ~1.0 (0.98–0.99) to 2.0 times of $d_{dry}$ were obtained: the ranges of $d_{va}$ for particles with $d_{dry}$ of 100, 200, 300, and 360 nm were 98–197, 197–395, 295–589, and 353–707 nm, respectively. Second, the mole amounts of sulfate, nitrate, and ammonium in 1 m$^3$ of air were derived. Third, the amount of ammonium nitrate (AN), ammonium sulfate (AS), letovicite (LET), ammonium hydrogen sulfate (AHS), and sulfuric acid (SA) per mole in 1 m$^3$ of air was determined, assuming that nitrate was fully neutralized with ammonium, and that sulfate could present in the form of AS, LET, AHS, and/or SA according to the amount of remaining ammonium. The detected non-refractory chloride was not considered because of its low concentration compared with the concentrations of the other non-refractory components. The contribution of sea salt and minerals to the sub-micrometer aerosol

particles was likely small (Han et al., 2014; Deng et al., 2018) and was also not considered. Fourth, the volumes of BC, OA, and LET were derived using their respective densities and the volumes of AN, AS, AHS, and SA were derived using their respective molar volumes. The density of BC was assumed to be 1.77 g cm$^{-3}$ (Park et al., 2004). The density of organics ($\rho_{org}$) was estimated to be $1.32 \pm 0.09$ g cm$^{-3}$ using the O:C and H:C ratios of organics derived from AMS measurements (Kuwata et al., 2012), and the mean value of 1.32 g cm$^{-3}$ was adopted for this study. The density of LET was assumed to be 1.83 g cm$^{-3}$ (Padró et al., 2010). The molar volumes of AN, AS, AHS, and pure liquid SA, which are the same as the ones used in the E-AIM II model, were adopted in this study. Finally, the volume fractions of each species were obtained.

**Text S6. Derivation of $g_{f,AS}$ and $\kappa_i$ using the E-AIM II model**

The hygroscopic growth factor of pure ammonium sulfate particles ($g_{f,AS}$) and the hygroscopicity parameter of each inorganic salt ($\kappa_i$) at 85 % RH were derived based on the output of the E-AIM II model and the $\kappa$–Köhler equation (Petters and Kreidenweis, 2007) as follows.

The water activity ($a_w$) range from 0.8000 to 0.8499 at a resolution of 0.0001 was applied to the E-AIM II model for unit mole of each inorganic salt at the temperature of 294 K. For the calculation, the partition of HNO$_3$, NH$_3$, and H$_2$SO$_4$ into the vapor phase was prevented and the formation of solid AS was also prevented. The hygroscopic growth factor ($g_f$) corresponding to each $a_w$ was derived from the ratio of total wet volume ($V_{wet}$) to the dry molar volume of the pure salt. The $\kappa_i$ corresponding to each $a_w$ was derived from Equation 2 of Petters and Kreidenweis (2007). The exponential part of the $\kappa$–Köhler equation was derived on the assumption that the partial molar volume of water equals the molar volume of pure water. Here, $a_w$ was calculated based on the relationship that the RH above the particle surface equals the product of the $a_w$ and the exponential part of the $\kappa$–Köhler equation representing the Kelvin effect. For particles

with respective $d_{dry}$, an $a_w$ value at which the RH above the particle surface was nearest to 85 % ($a_{w,85}$) was obtained by applying a wet diameter represented by the product of $d_{dry}$ and $g_f$. The hygroscopicity parameter value at $a_{w,85}$ was defined as $\kappa_i$ (Table S3). For AS, the $g_f$ at $a_{w,85}$ was defined as $g_{f,AS}$ (Table S2). For the derivation of $\kappa_i$, the surface tension of the solution was applied, whereas for the derivation of $\kappa_f$ (Sect. 3.1) the surface tension of pure water was applied. The uncertainty of the surface tension should not introduce large uncertainty in the derived $\kappa_{org}$ because the difference of $\kappa_i$ obtained using the surface tension of pure water and that obtained using the surface tension of the solution was small (within 0.38 %, Table S3).

**Text S7. Determination of the range of $d_{va}$ for the calculation of $\kappa_{org}$**

The PToF mode AMS data over a lower $d_{va}$ range was subjected to low signal intensity (Fig. S2) when it was adopted for the derivation of size-resolved $\kappa_{org}$. To determine the applicable range of $d_{va}$ for this study, the mean mass-size distribution data of OA during the entire study period was compared to that of a baseline region (Fig. S3). The baseline region here is the transition region between the regions dominated by gaseous species and particle signals of OA, and corresponds to the PToF time region of 0.00135458–0.00150458 s. To eliminate data under the strong influence of the signals from gaseous species, the $d_{va}$ of 98 nm (where the mass ratio of the mean observed OA to mean baseline OA became greater than three) was adopted as the lower limit of the $d_{va}$ range for the deviation of $\kappa_{org}$. In Fig. S3, the mean mass-size distribution of OA for filtered air (collected by connecting a HEPA filter (TSI) to the inlet tubing outside the instrument room) is also presented for comparison. Only in the $d_{va}$ range of 90–716 nm, the mean values of the observed OA mass in each $d_{va}$ bin were greater than that of OA for the filtered air. This also indicates the strong influence of the signals from gaseous species at $d_{va}$ smaller than ~90 nm. Because the OA data of filtered air was noisy, it was not used for the determination of the applicable $d_{va}$ range.

**Text S8. Derivation of size-resolved PMF factors**

To explain the diurnal variation and size-dependence of $\kappa_{org}$ from the compositional characteristics of OA, size-resolved PMF factors (LOOA and MOOA) were derived through multivariable linear regression analysis as follows.

First, the fragment profiles of LOOA and MOOA from bulk OA mass spectra were converted to the profiles in unit m/z resolution by summing up the intensity of the fragment ion signals at the same unit m/z. Then, for each time period in 2 h time resolution and for each PToF size bin $i$, the contributions from LOOA ($a_i$) and MOOA ($b_i$) were derived using the Solver function in Microsoft Excel by minimizing the value of the following formula:

$$\sum_{j=12}^{115} \left[ x_{ij} - \left( a_i f_{1j} + b_i f_{2j} \right) \right]^2$$

Where $x_{ij}$ is the measured signal intensity in size bin $i$ at m/z $= j$, and $f_{1j}$ and $f_{2j}$ are the respective normalized signal intensity of LOOA and MOOA at m/z $= j$. The unit of $x_{ij}$, $a_i$, and $b_i$ was μg m$^{-3}$, whereas $f_{1j}$ and $f_{2j}$ were dimensionless.

The size-resolved LOOA and MOOA were derived for the $d_{va}$ range from $\leq 10$ nm to around 900 nm (Fig. S6). However, only the $d_{va}$ range above 98 nm was adopted for the analysis. This is because the uncertainty of the contributions from LOOA and MOOA in the lower $d_{va}$ range was presumably relatively large, given the low organic signal intensity (Fig. S3) and high residual to measured OA mass ratio (Fig. S6).

**Text S9. Assessment of the contributions of BSOA and anthropogenic OA to the enhancement of OA in the daytime**

The contributions of BSOA and anthropogenic OA to the enhancement of OA in the daytime (in relation to the background period) were assessed using BC as a tracer of OA that did not come from BSOA

formation (non-BSOA-OA). Here, the non-BSOA-OA was considered the sum of regional OA and other anthropogenic OA. The diurnal variation data on the mass concentration of BC was scaled to represent the diurnal variation of non-BSOA-OA. For the scaling, the observed OA during the background period (i.e., 0600–0800 JST, when the daily minima of $m_{\mathrm{org}}$ appeared) was assumed to be composed only of non-BSOA-OA. The scaling factor was calculated to be 36.5. The mass concentrations of non-BSOA-OA ($m_{\mathrm{non\text{-}BSOA,bulk}}$) and BSOA ($m_{\mathrm{BSOA,bulk}}$) were then estimated using the following equations.

$$m_{\mathrm{non\text{-}BSOA,bulk}} = m_{\mathrm{BC}} \times 36.5 \tag{S1}$$

$$m_{\mathrm{BSOA,bulk}} = m_{\mathrm{org}} - m_{\mathrm{non\text{-}BSOA,bulk}} \tag{S2}$$

Note that $m_{\mathrm{BSOA,bulk}}$ and $m_{\mathrm{non\text{-}BSOA,bulk}}$ are different from the mass concentrations of the BSOA and ROA defined from the size-resolved LOOA/MOOA data (Sect. 4.2.2 and Text S10). The $m_{\mathrm{BSOA,bulk}}$ may be negatively biased because the charring of OA during the PSAP measurement may have resulted in a positive bias of $m_{\mathrm{BC}}$. The increase of the OA mass concentration in the daytime ($m_{\mathrm{org,ENH}}$) was estimated by subtracting the $m_{\mathrm{org}}$ during the background period from that during the period of interest. The ratio of $m_{\mathrm{BSOA,bulk}}$ to $m_{\mathrm{org,ENH}}$ was in the range 0.6–0.9 during 1200–1600 JST, and it was in the range 0.1–0.5 during 1600–2030 JST (Fig. S10). The result indicates that BSOA was the main contributor to the enhancement of OA at least during 1200–1600 JST. In a later period, a larger contribution from anthropogenic OA is not ruled out.

**Text S10. Derivation of $v_{\mathrm{BSOA}}$ and $v_{\mathrm{ROA}}$ from $v_{\mathrm{LOOA}}$ and $v_{\mathrm{MOOA}}$**

Because the observed OA can be assumed to be contributed either by LOOA and MOOA, or by BSOA and ROA, the sum of LOOA and MOOA should equal the sum of BSOA and ROA. The size-resolved $v_{\mathrm{BSOA}}$ and $v_{\mathrm{ROA}}$ can be derived from the size-resolved $v_{\mathrm{LOOA}}$ and $v_{\mathrm{MOOA}}$. From the analysis in Sect. 4.2.2, 0.97 (0.52), 0.88 (0.53), 0.80 (0.50), and 0.79 (0.45) of the volume of BSOA (ROA) for particles with $d_{\mathrm{dry}}$

of 100, 200, 300, and 360 nm were assigned to LOOA, and the balances were assigned to MOOA. The relations are expressed by the equations:

$$v_{BSOA} + v_{ROA} = v_{LOOA} + v_{MOOA} \qquad (S3)$$

$$a \times v_{BSOA} + b \times v_{ROA} = v_{LOOA} \qquad (S4)$$

Where $v_{BSOA}$, $v_{ROA}$, $v_{LOOA}$, and $v_{MOOA}$ are the size-resolved volume concentrations of BSOA, ROA, LOOA, and MOOA, respectively, and $a$ and $b$ represent the size-resolved volume fractions of BSOA and ROA, respectively, that were assigned to LOOA. The volume concentrations of BSOA and ROA were estimated using the equations:

$$v_{BSOA} = [(1 - b) / (a - b)] \times v_{LOOA} - [b / (a - b)] \times v_{MOOA} \qquad (S5)$$

$$v_{ROA} = [(a - 1) / (a - b)] \times v_{MOOA} + [a / (a - b)] \times v_{LOOA} \qquad (S6)$$

Equations (S5) and (S6) were used to estimate the contributions of BSOA to aerosol water uptake and to the CCN number concentration (Sect. 4.4 and Text S11).

**Text S11. Estimation of the contributions of OA and BSOA to CCN concentrations**

The contributions of OA and BSOA to the CCN number concentration were estimated from their size-resolved fractional contributions to aerosol water uptake and from the measured aerosol number-size distributions. The analysis was performed for diurnal variation data with 2 h resolution. A schematic of the estimate is presented in Fig. S17. For the estimate, the observed aerosol particles were assumed to be internally mixed, and all the particles with $d_{dry}$ larger than 70 nm were assumed to be CCN active. The contributions of OA (BSOA) to the water uptake of particles with $d_{dry}$ of 100, 200, 300, and 360 nm were applied for the diameter ranges of 70–150, 150–250, 250–330, 330–430 nm, respectively. For the diameter ranges larger than 430 nm, the CCN number concentration contributed by OA (BSOA) was not considered because of the low aerosol number concentrations. In each diameter range, the fractional contribution of

OA (BSOA) to d$N_{CN}$/dlog$d_{dry}$ equals the fractional contribution of OA (BSOA) to the total aerosol water uptake, which was represented as the product of the volume fraction of OA (BSOA) and $\kappa_{org}$ ($\kappa_{BSOA}$) divided by $\kappa_t$ [i.e., $\varepsilon_{org}\kappa_{org}/\kappa_t$ ($\varepsilon_{BSOA}\kappa_{BSOA}/\kappa_t$)]. The total fractional contribution of OA (BSOA) to the total CCN number concentration, $F_{CCN,OA}$ ($F_{CCN,BSOA}$), equals the integration of the product of the water uptake fraction and d$N_{CN}$/dlog$d_{dry}$ above the assumed CCN activation diameter (70 nm), divided by the total CCN number concentration.

For the above analysis, the water uptake of each aerosol component was represented by the product of the volume fraction of the aerosol component ($\varepsilon_i$) and its hygroscopicity ($\kappa_i$), that is, $\varepsilon_i\kappa_i$. The mean $\varepsilon_i$ in each 2 h time section of the day was derived as follows. First, the mean values of the volume concentrations of each inorganic species, organics, and organic fractions (AN, AS, LET, AHS, SA, BC, OA, LOOA, and MOOA; $\bar{v}_i$) were calculated from the 2 h resolution data. Second, the mean volume concentrations of LOOA ($\bar{v}_{LOOA}$) and MOOA ($\bar{v}_{MOOA}$) were scaled so that their sum equals the mean volume concentration of OA ($\bar{v}_{OA}$). Third, the mean volume concentrations of BSOA ($\bar{v}_{BSOA}$) and ROA ($\bar{v}_{ROA}$) were estimated using Eqs. (S5) and (S6). Then, $\varepsilon_i$ was calculated directly from those $\bar{v}_i$. The $\kappa$ of the inorganic salts under the condition of 0.42 % SS, and at the temperature of the HTDMA measurement in this study, were used to consider the difference of $\kappa$ between sub- and super-saturated water vapor conditions. Here, the $\kappa$ for AN, AS, LET, AHS, and SA were calculated to be 0.73, 0.60, 0.63, 0.62, and 0.65, respectively, following the method in Deng et al. (2018). The difference of the $\kappa$ of organics under sub- and super-saturated conditions was not considered. The $\kappa$ values of OA and BSOA used for the calculation are presented in Tables S10 and S11, respectively. The contributions of OA and BSOA to the water uptake were calculated as $\varepsilon_{org}\kappa_{org}/\kappa_{t,reconst,org}$ and $\varepsilon_{BSOA}\kappa_{BSOA}/\kappa_{t,reconst,BSOA}$, respectively, where,

$$\kappa_{t,reconst,org} = \varepsilon_{AN}\kappa_{AN} + \varepsilon_{AS}\kappa_{AS} + \varepsilon_{LET}\kappa_{LET} + \varepsilon_{AHS}\kappa_{AHS} + \varepsilon_{SA}\kappa_{SA} + \varepsilon_{org}\kappa_{org} \qquad (S7)$$

$$\kappa_{t,reconst,BSOA} = \varepsilon_{AN}\kappa_{AN} + \varepsilon_{AS}\kappa_{AS} + \varepsilon_{LET}\kappa_{LET} + \varepsilon_{AHS}\kappa_{AHS} + \varepsilon_{SA}\kappa_{SA} + \varepsilon_{BSOA}\kappa_{BSOA} + \varepsilon_{ROA}\kappa_{ROA}$$

**Text S12. Assessment of the diurnal variation of the CCN activation diameter**

Although the variation of the CCN activation diameter with time influences the prediction of $F_{CCN,OA}$ and $F_{CCN,BSOA}$, the degree was found to be small. In the summertime observation in 2014 (Deng et al., 2018), the range of the diurnal variation of the CCN activation diameter was from 64 to 76 nm, whereas the CCN activation diameter assumed in this study is 70 nm. Applying 64 or 76 nm to an assumed CCN activation diameter results in the deviations of the predicted $F_{CCN,OA}$ and $F_{CCN,BSOA}$ only by −1.9–2.3 % and −3.1–3.8 %, respectively.

**Text S13. Assessment of the change of CCN activation diameter accompanying the aging of BSOA**

The aging of BSOA may change the CCN activation diameter and influence the prediction of $F_{CCN,BSOA}$. However, the change and the influence are considered to be small as explained below. If the range of $\kappa_i$ of 0.17 to 0.35 (i.e., the $\kappa_i$ range of (mean – SD) to (mean + SD) under 0.42 % SS condition in Deng et al. (2018)), and the approximate maximum $\varepsilon_{BSOA}$ of 0.6 (Fig. S18) are applied, the increase of $\kappa_{BSOA}$ by 0.09 (i.e., the difference between $\kappa_{BSOA}$ and $\kappa_{ROA}$ for particles with $d_{dry}$ of 100 nm) leads to a decrease in the CCN activation diameter by 3–7 nm. The resulting decrease leads to increase of the predicted $F_{CCN,BSOA}$ by 0.15–3.8 %, which is regarded as small.

[Figure]

**Figure S1:** Summary of the two-factor result from the PMF analysis: (a) organic mass spectra of LOOA and MOOA colored according to ion groups (fpeak = 0 and SEED = 1), and the atomic ratios of O to C, H to C, and N to C, as well as OM to OC ratio for each factor; (b) Q/Qexpected as a function of the number of factors, where Q is the sum of the weighted squared residuals and Qexpected is the expected Q value; (c) Q/Qexpected as a function of the fpeak values with SEED = 1 and the number of factor = 2; (d) Q/Qexpected as a function of the SEED values with fpeak = 0 and the number of factor = 2; (e) distributions of the scaled residual for each m/z (fpeak = 0 and SEED = 1); time series of (f) the measured organic mass concentrations and those reconstructed (= LOOA + MOOA) (fpeak = 0 and SEED = 1), and (g) residual OA (= measured − reconstructed) (fpeak = 0 and SEED = 1).

[Figure]

**Figure S2:** Mean mass-size distributions of organics (OA), sulfate ($SO_4$), ammonium ($NH_4$), nitrate ($NO_3$), and chloride (Chl) in (a) linear and (b) logarithmic scales over the entire study period.

[Figure]

**Figure S3:** d*M*/d*PToF* (*M* and *PToF* here refer to mass concentration and PToF time, respectively) versus PToF time for the observed OA and OA measured by placing a HEPA filter in the inlet tubing outside the instrument room (zero OA), averaged for the entire study period. The region shaded in pink indicates the PToF time region chosen as a baseline for evaluation of the effective PToF diameter range. (Note that this is not the DC marker region.) The dark solid curve is the absolute value of the ratio of the observed OA at respective PToF time to the mean of the observed OA in the baseline region ($R_{obs/base}$). The blue dash-dotted line indicates the $R_{obs/base}$ value of three. The vacuum aerodynamic diameter that corresponds to the PToF time is presented on the top axis.

[Figure]

**Figure S4:** $\kappa_{org}$ of 100 nm particles derived based on the chemical composition in the vacuum aerodynamic diameter ($d_{va}$) range of 69–138 nm versus that of 98–197 nm. Data points with $\kappa_{org}$ smaller than −0.50 or $\kappa_{org}$ higher than 0.60 are not presented.

[Figure]

**Figure S5:** $\kappa_t$ versus $\varepsilon_{org}$ for particles with $d_{dry}$ of 100, 200, 300, and 360 nm for the entire study period. In each panel, marks and a solid line represent individual data and the corresponding linear regression line, respectively. The correlation coefficient ($r$) of each is also presented.

[Figure]

**Figure S6:** Mean mass-size distributions of (a) residual, (b) LOOA, and (c) MOOA over the entire study period. The residual is the difference between the measured and reconstructed (i.e., LOOA + MOOA) mass concentrations of OA in each size bin. The error bars indicate the standard deviation. The ratios of the sum of the absolute value of the residual to the measured mass concentration of OA are superimposed in panel (a) (right axis).

[Figure]

**Figure S7:** Time series of (a) air temperature and relative humidity (RH), (b) precipitation and solar radiation, (c) mass concentration of BC, and mixing ratios of (c) CO, (d) $O_3$, (e) $CO_2$, and (f) NO, $NO_2$, and $NO_x$.

[Figure]

**Figure S8:** Diurnal variations of (a) air temperature and relative humidity (RH), (b) solar radiation and the mixing ratios of $O_3$, and (c) mixing ratios of NO, $NO_2$, $NO_x$, and CO over the entire study period.

[Figure]

**Figure S9:** Five-day backward air mass trajectories from 500 m agl (above ground level) over Wakayama Forest Research Station at one-day intervals. The arrival time of the air masses at the study site was 1500 JST. The solid circle denotes the location of the study site. Solid trajectories are for days when more than 22 of the 24 hourly trajectories are from terrestrial regions. Dashed trajectories are for days when more

than 21 of the 24 hourly trajectories are from the North Pacific. Dotted trajectories are for days when 10 of the 24 hourly trajectories are from the North Pacific. The map is based on GSHHG 2.3.4; the shoreline polygon data at crude resolution is used. We consider an air mass is from the North Pacific if the trajectory never passes over terrestrial area that appears on the map before it reaches the Kii Peninsula. The trajectories were produced using NOAA's HYSPLIT atmospheric transport and dispersion modeling system (Draxler and Hess, 1998).

[Figure]

**Figure S10.** Diurnal variation of the mass concentrations of OA ($m_{org}$), BC ($m_{BC}$), and non-BSOA-OA ($m_{non-BSOA,bulk}$; Text S9) during the entire study period. The left-slash pattern represents the background period. As an example, the vertical bars represent the estimates of the total enhancement of OA ($m_{org,ENH}$; blue) and the enhancement contributed by BSOA ($m_{BSOA,bulk}$; gray) for the period 1400–1600 JST (right slash pattern).

[Figure]

**Figure S11:** Time series of the probability distribution functions of the hygroscopic growth factors ($g_f$-PDF) of aerosol particles with $d_{dry}$ of (a) 30, (b) 70, (c) 100, (d) 300, and (e) 360 nm.

[Figure]

**Figure S12:** Time series of the mean growth factors ($g_{f,m}$) of aerosol particles with different dry diameters. The mean ± SD values over the entire study period for each diameter are also presented.

[Figure]

**Figure S13:** Diurnal variation of the size-resolved volume fractions of total inorganic salts ($\varepsilon_{\text{inorgsalt}}$) during the entire study period.

[Figure]

**Figure S14:** The $\kappa_{\text{org}}$ versus $v_{\text{LOOA}}/(v_{\text{LOOA}}+v_{\text{MOOA}})$ for particles with $d_{\text{dry}}$ of 300 and 360 nm. The time resolution of the data is 2 h. In each panel, marks and a solid line represent individual data and the corresponding linear regression line, respectively. The regression equation and correlation coefficient of each are also presented. Only data with $\varepsilon_{\text{org}}$ greater than 0.40 are used.

[Figure]

**Figure S15:** The $\kappa_{org}$ reconstructed using $\kappa_{LOOA}$, $\kappa_{MOOA}$, and size-resolved $\nu_{LOOA}/(\nu_{LOOA}+\nu_{MOOA})$ (reconstructed $\kappa_{org}$) versus the $\kappa_{org}$ derived from measured $\kappa_t$ and aerosol chemical composition (measured $\kappa_{org}$; Sect. 3.2). In panel (a), markers represent size-resolved diurnal variation data at 2 h resolution, solid lines are linear regression lines for particles with respective diameters, and the dashed line is the linear regression line for all 100–360 nm particles. In panel (b), markers represent the size-resolved mean $\kappa_{org}$ during 1200–2000 JST, and the solid line is the linear regression line for the size-resolved mean $\kappa_{org}$. Respective regression equations and coefficients of determination ($r^2$) are also presented. Only $\kappa_{org}$ data with $\varepsilon_{org}$ greater than 0.40 are used for the comparison.

[Figure]

**Figure S16:** Diurnal variation of the volume concentrations of LOOA and MOOA and the mass concentration of sulfate for particles with $d_{dry}$ of 100, 200, 300, and 360 nm over the entire study period. The scaled sulfate represents the diurnal variation of ROA that was contributed by LOOA (left panels) or MOOA (right panels). The scaling factor for the scaled sulfate in each panel is the mean volume concentration of OA during 0600–0800 JST, divided by the mean mass concentration of sulfate in the same period. The volume concentrations of LOOA and MOOA were derived from the respective mass concentrations (Text S8). The densities of LOOA and MOOA were calculated using their O:C and H:C ratios following Kuwata et al. (2012) and were 1.24 and 1.54 g cm$^{-3}$, respectively.

[Figure]

**Figure S17:** Estimate of the contributions of BSOA to the CCN number concentration from the viewpoint of its size-resolved contribution to the aerosol water uptake. The solid line indicates the mean aerosol number-size distribution during the entire study period. Shaded areas in green represent the fraction of CCN contributed by BSOA and in red, that contributed by other components assuming a CCN activation diameter of 70 nm (Text S11).

[Figure]

**Figure S18:** The diurnal variation of the volume fractions of BSOA ($\varepsilon_{BSOA}$) for particles with $d_{dry}$ of 100, 200, 300, and 360 nm over the entire study period (Text S11).

[Figure]

**Figure S19:** Box and whiskers plot of the diurnal variation of the O:C ratios of bulk OA (only data with $m_{org} > 0.3$ µg m$^{-3}$ are included) for the entire study period. The horizontal lines in the boxes indicate the median values, boundaries of the boxes indicate the 25th- and 75th-percentiles, and the whiskers indicate the highest and lowest values. The cross symbols in the boxes indicate the mean values.

**Table S1: Mode diameters** [a] **of PSL size standards measured by DMAs in the HTDMA (DMA1 and DMA2) and the SMPS (DMA3) (mean ± SD, nm)**

| Manufacturer warranty | DMA1 | | DMA2 | | DMA3 | |
|---|---|---|---|---|---|---|
| | Before [c] | After [d] | Before [c] | After [d] | Before [c] | After [d] |
| **55 (± 1)** [b] | 56.6 ± 0.4 | 56.2 ± 0.4 | 56.8 ± 0.4 | - | 59.9 ± 0.2 | - |
| **100 (± 3)** [b] | 98.0 ± 0.2 | 98.4 ± 0.1 | 97.0 ± 0.1 | 96.2 ± 0.2 | 101.2 ± 0.3 | 101.1 ± 0.1 |
| **309 (± 9)** [b] | 297.7 ± 0.8 | 298.0 ± 0.4 | 290.4 ± 1.2 | - | 303.6 ± 0.3 | - |
| **498 (± 9)** [b] | - | - | 478.4 ± 5.8 | - | 499.6 ± 4.4 | - |

[a] The mean ± SD of the mode diameters from fittings are presented (unit: nm).
[b] Mean diameter (± the expanded uncertainty; $k = 2$).
[c] Before the atmospheric observations.
[d] After the atmospheric observations.

**Table S2: The $g_{f,m}$ of ammonium sulfate (AS) particles measured under dry condition ($g_{f,m,dryAS}$) and at 85 % RH ($g_{f,m,wetAS}$), and calculated $g_f$ of AS particles at 85 % RH ($g_{f,AS}$)**

| $d_{dry}$ (nm) | 30 | 50 | 70 | 100 | 200 | 300 | 360 |
|---|---|---|---|---|---|---|---|
| $g_{f,m,dryAS}$ | 0.959 | 0.976 | 0.984 | 0.985 | 0.988 | 0.982 | 0.981 |
| $g_{f,m,wetAS}$[a] | 1.52 | 1.54 | 1.54 | 1.54 | 1.55 | 1.57 | 1.59 |

| $g_{f,AS}$[a] | 1.49 | 1.52 | 1.54 | 1.55 | 1.57 | 1.57 | 1.57 |
|---|---|---|---|---|---|---|---|
| **Difference (%)**[b] | 2.0 | 1.3 | 0 | –0.65 | –1.3 | 0 | 1.3 |

[a] Corrected for the difference of sizing between DMA1 and DMA2.
[b] $((g_{f,m,wetAS} - g_{f,AS}) / g_{f,AS}) \times 100$.

**Table S3: The $\kappa$ values of inorganic salts ($\kappa_i$) at 85 % RH derived using the surface tension of the solution and of pure water**

|  | $\kappa_i$, with surface tension of solution | | | | $\kappa_i$, with surface tension of pure water | | | |
|---|---|---|---|---|---|---|---|---|
| $d_{dry}$ (nm) | 100 | 200 | 300 | 360 | 100 | 200 | 300 | 360 |
| AN | 0.553 | 0.555 | 0.555 | 0.556 | 0.553 | 0.555 | 0.556 | 0.556 |
| AS | 0.533 | 0.527 | 0.525 | 0.524 | 0.531 | 0.526 | 0.524 | 0.524 |
| LET | 0.550 | 0.545 | 0.543 | 0.543 | 0.549 | 0.544 | 0.543 | 0.543 |
| AHS | 0.612 | 0.607 | 0.605 | 0.605 | 0.612 | 0.607 | 0.605 | 0.605 |
| SA | 0.972 | 0.959 | 0.955 | 0.953 | 0.971 | 0.959 | 0.955 | 0.953 |

**Table S4: Data in Fig. 3 of the main manuscript** ("DataInFigure3ofTheManuscript.xlsx").

**Table S5: Data in Fig. 4 of the main manuscript** ("DataInFigure4ofTheManuscript.xlsx").

**Table S6: Data in Fig. 5 of the main manuscript** ("DataInFigure5ofTheManuscript.xlsx").

**Table S7: Comparisons of $\kappa_{org}$ and $v_{LOOA}/(v_{LOOA}+v_{MOOA})$ between particles with different $d_{dry}$**

| $d_{dry}$ of particles to compare (nm) | 1200–2000 JST | | | | 2000–1200 JST | | | |
|---|---|---|---|---|---|---|---|---|
|  | $\kappa_{org}$ | | $v_{LOOA}/(v_{LOOA}+v_{MOOA})$ | | $\kappa_{org}$ | | $v_{LOOA}/(v_{LOOA}+v_{MOOA})$ | |
|  | Diff[a] | p-value[c] | Diff[b] | p-value[c] | Diff[a] | p-value[c] | Diff[b] | p-value[c] |
| **200 vs 100** | 0.06 | <0.01 | –0.06 | 0.02 | 0.04 | <0.01 | –0.01 | 0.65 |
| **300 vs 200** | <0.01 | 0.71 | –0.08 | <0.01 | <0.01 | 0.31 | –0.06 | <0.01 |
| **360 vs 300** | 0.02 | 0.15 | –0.04 | <0.01 | –0.02 | 0.02 | –0.05 | <0.01 |
| **360 vs 200** | 0.02 | 0.07 | –0.11 | <0.01 | –0.02 | 0.07 | –0.11 | <0.01 |

[a] The mean of (the $\kappa_{org}$ of particles with relatively large $d_{dry}$ – the $\kappa_{org}$ of particles with relatively small $d_{dry}$).
[b] The mean of (the $v_{LOOA}/(v_{LOOA}+v_{MOOA})$ of particles with relatively large $d_{dry}$ – the $v_{LOOA}/(v_{LOOA}+v_{MOOA})$ of particles with relatively small $d_{dry}$).
[c] From 10 % two-sided t-test for the significance of the difference of Diff from zero. Low values indicate significant differences.

**Table S8: Diurnal variation of $\kappa_i$ and $\kappa_{org}$ at 2 h resolution, and their mean and SD for the entire period**

|  | $\kappa_i$ | | | | | | | $\kappa_{org}$ | | | |
|---|---|---|---|---|---|---|---|---|---|---|---|
| $d_{dry}$ (nm) | 30 | 50 | 70 | 100 | 200 | 300 | 360 | 100 | 200 | 300 | 360 |
| **0000–0200 JST** | 0.16 | 0.20 | 0.22 | 0.24 | 0.34 | 0.37 | 0.36 | 0.16 | 0.23 | 0.25 | 0.21 |
| **0200–0400 JST** | 0.22 | 0.21 | 0.22 | 0.24 | 0.34 | 0.37 | 0.35 | 0.16 | 0.20 | 0.21 | 0.18 |

| | | | | | | | | | | | |
|---|---|---|---|---|---|---|---|---|---|---|---|
| **0400–0600 JST** | 0.18 | 0.21 | 0.22 | 0.25 | 0.34 | 0.38 | 0.39 | 0.16 | 0.21 | 0.24 | 0.23 |
| **0600–0800 JST** | 0.21 | 0.22 | 0.21 | 0.25 | 0.35 | 0.38 | 0.36 | 0.18 | 0.22 | 0.24 | 0.19 |
| **0800–1000 JST** | 0.15 | 0.18 | 0.21 | 0.22 | 0.32 | 0.33 | 0.37 | 0.16 | 0.19 | 0.16 | 0.18 |
| **1000–1200 JST** | 0.13 | 0.16 | 0.16 | 0.19 | 0.29 | 0.33 | 0.34 | 0.12 | 0.17 | 0.15 | 0.12 |
| **1200–1400 JST** | 0.090 | 0.14 | 0.14 | 0.16 | 0.26 | 0.32 | 0.34 | 0.11 | 0.16 | 0.19 | 0.18 |
| **1400–1600 JST** | 0.083 | 0.13 | 0.13 | 0.16 | 0.24 | 0.28 | 0.33 | 0.10 | 0.15 | 0.16 | 0.20 |
| **1600–1800 JST** | 0.10 | 0.14 | 0.14 | 0.17 | 0.25 | 0.28 | 0.32 | 0.089 | 0.18 | 0.16 | 0.19 |
| **1800–2000 JST** | 0.13 | 0.15 | 0.16 | 0.19 | 0.27 | 0.31 | 0.32 | 0.12 | 0.18 | 0.18 | 0.19 |
| **2000–2200 JST** | 0.14 | 0.18 | 0.18 | 0.21 | 0.29 | 0.33 | 0.34 | 0.14 | 0.19 | 0.20 | 0.17 |
| **2200–0000 JST** | 0.12 | 0.19 | 0.20 | 0.23 | 0.31 | 0.35 | 0.36 | 0.16 | 0.19 | 0.20 | 0.17 |
| **Mean for entire period** | 0.12 | 0.18 | 0.18 | 0.21 | 0.30 | 0.34 | 0.35 | 0.13 | 0.18 | 0.19 | 0.19 |
| **SD for entire period** | 0.079 | 0.090 | 0.089 | 0.094 | 0.10 | 0.087 | 0.086 | 0.11 | 0.085 | 0.084 | 0.11 |

**Table S9: Data in Fig. 6 of the main manuscript.**

| Hour of the day | $F_{\text{CCN,OA}}$ (%) | $F_{\text{CCN,BSOA}}$ (%; fresh BSOA) | $F_{\text{CCN,BSOA}}$ (%; aged BSOA) |
|---|---|---|---|
| 0000–0200 JST | 44.5 | 5.85 | 10.6 |
| 0200–0400 JST | 44.9 | 2.66 | 4.88 |
| 0400–0600 JST | 40.1 | 0.238 | 0.338 |
| 0600–0800 JST | 42.8 | $-6.41 \times 10^{-3}$ | $-0.0373$ |
| 0800–1000 JST | 44.5 | 5.62 | 10.3 |
| 1000–1200 JST | 42.7 | 12.7 | 21.3 |
| 1200–1400 JST | 53.0 | 26.2 | 39.8 |
| 1400–1600 JST | 52.7 | 28.4 | 42.6 |
| 1600–1800 JST | 47.3 | 24.9 | 38.5 |
| 1800–2000 JST | 45.7 | 19.8 | 31.9 |
| 2000–2200 JST | 49.4 | 16.6 | 27.5 |
| 2200–0000 JST | 45.3 | 4.99 | 9.10 |

**Table S10: Different assumptions of $\kappa_{org}$ for the prediction of $F_{CCN,OA}$**

| $d_{dry}$ (nm) | TimeSize $\kappa_{org}$[a] | | | | SizeReso $\kappa_{org}$[b] | | | | TimeReso $\kappa_{org}$[c] | | | | Single $\kappa_{org}$[d] | | | |
|---|---|---|---|---|---|---|---|---|---|---|---|---|---|---|---|---|
| | 100 | 200 | 300 | 360 | 100 | 200 | 300 | 360 | 100 | 200 | 300 | 360 | 100 | 200 | 300 | 360 |
| 0000–0200 JST | 0.16 | 0.23 | 0.25 | 0.21 | | | | | | 0.21 | | | | | | |
| 0200–0400 JST | 0.16 | 0.20 | 0.21 | 0.18 | | | | | | 0.19 | | | | | | |
| 0400–0600 JST | 0.16 | 0.21 | 0.24 | 0.23 | | | | | | 0.21 | | | | | | |
| 0600–0800 JST | 0.18 | 0.22 | 0.24 | 0.19 | | | | | | 0.21 | | | | | | |
| 0800–1000 JST | 0.16 | 0.19 | 0.16 | 0.18 | | | | | | 0.17 | | | | | | |
| 1000–1200 JST | 0.12 | 0.17 | 0.15 | 0.12 | 0.14 | 0.19 | 0.19 | 0.19 | | 0.14 | | | | 0.18 | | |
| 1200–1400 JST | 0.11 | 0.16 | 0.19 | 0.18 | | | | | | 0.16 | | | | | | |
| 1400–1600 JST | 0.10 | 0.15 | 0.16 | 0.20 | | | | | | 0.15 | | | | | | |
| 1600–1800 JST | 0.089 | 0.18 | 0.16 | 0.19 | | | | | | 0.15 | | | | | | |
| 1800–2000 JST | 0.12 | 0.18 | 0.18 | 0.19 | | | | | | 0.17 | | | | | | |
| 2000–2200 JST | 0.14 | 0.19 | 0.20 | 0.17 | | | | | | 0.17 | | | | | | |
| 2200–0000 JST | 0.16 | 0.19 | 0.20 | 0.17 | | | | | | 0.18 | | | | | | |

[a] Time- and size-resolved $\kappa_{org}$.
[b] Time-averaged, size-resolved $\kappa_{org}$.
[c] Time-resolved, size-averaged $\kappa_{org}$.
[d] Time- and size-averaged $\kappa_{org}$.

**Table S11: Different assumptions of $\kappa_{BSOA}$ for the prediction of $F_{CCN,BSOA}$**

| $d_{dry}$ (nm) | Size-resolved $\kappa_{BSOA}$ | Size-averaged $\kappa_{BSOA}$ | Aged, size-resolved $\kappa_{BSOA}$ |
|---|---|---|---|
| 100 | 0.089 | | 0.18 |
| 200 | 0.11 | 0.11 | 0.18 |
| 300 | 0.12 | | 0.18 |
| 360 | 0.12 | | 0.19 |

**Table S12: Data in Fig. 7 of the main manuscript.**

| Hour of day | Ratios of different $F_{CCN,OA}$ | | | | Ratios of different $F_{CCN,BSOA}$ | | |
|---|---|---|---|---|---|---|---|
| | TimeSize $\kappa_{org}$[a] | SizeReso $\kappa_{org}$[b] | TimeReso $\kappa_{org}$[c] | Single $\kappa_{org}$[d] | Size-resolved $\kappa_{BSOA}$ | Size-averaged $\kappa_{BSOA}$ | Aged, Size-resolved $\kappa_{BSOA}$ |
| 0000–0200 JST | 1 | 0.913 | 1.09 | 0.993 | 1 | 1.19 | 1.81 |
| 0200–0400 JST | 1 | 0.932 | 1.04 | 1.01 | 1 | 1.17 | 1.84 |
| 0400–0600 JST | 1 | 0.920 | 1.10 | 1.00 | 1 | 0.899 | 1.42 |
| 0600–0800 JST | 1 | 0.876 | 1.04 | 0.951 | 1 | 3.61 | 5.82 |
| 0800–1000 JST | 1 | 0.964 | 1.02 | 1.04 | 1 | 1.19 | 1.82 |
| 1000–1200 JST | 1 | 1.10 | 1.05 | 1.18 | 1 | 1.14 | 1.68 |
| 1200–1400 JST | 1 | 1.08 | 1.10 | 1.14 | 1 | 1.12 | 1.52 |
| 1400–1600 JST | 1 | 1.13 | 1.13 | 1.19 | 1 | 1.13 | 1.50 |
| 1600–1800 JST | 1 | 1.18 | 1.19 | 1.26 | 1 | 1.14 | 1.55 |
| 1800–2000 JST | 1 | 1.06 | 1.13 | 1.15 | 1 | 1.15 | 1.61 |
| 2000–2200 JST | 1 | 0.983 | 1.06 | 1.06 | 1 | 1.16 | 1.65 |
| 2200–0000 JST | 1 | 0.950 | 1.04 | 1.03 | 1 | 1.18 | 1.82 |

[a] Time- and size-resolved $\kappa_{org}$.
[b] Time-averaged, size-resolved $\kappa_{org}$.
[c] Time-resolved, size-averaged $\kappa_{org}$.
[d] Time- and size-averaged $\kappa_{org}$.

---

## Author Response (AR2)

*The authors answered some of my questions, but I am not satisfied with the answers regarding the method and the conclusions that how the authors calculated the contributions to CCN concentration from BSOA from an ambient dataset. The logic behind it as I see is the involving of BSOA into organic aerosols will change the hygroscopicity parameter κ, then influence the critical diameter of particles that are able to activate, for instance, not 70 nm anymore, which thus change the potential CCN concentration. However, the calculation method from the authors are all based on fixing the size of CCN activation. Then these two things are actually in conflict with each other. Without any CCN measurements with BSOA and without BSOA in your aerosol population, it is hardly to derive the contribution of BSOA to CCN concentration. Your current claims are wrong from your calculations. I strongly suggest the authors change the whole section regarding this part and make proper conclusions with respect to CCN contributions from BSOA.*

We appreciate the reviewer's comment. Considering the comment, we calculated $d_{\text{act}}$ values at 0.4 % SS for respective 2 h time bins of day using the reconstructed hygroscopicity parameter values of aerosol particles ($\kappa_{\text{t,reconst,org}}$ and $\kappa_{\text{t,reconst,BSOA}}$ in Text S11), following the method using $\kappa$-Köhler theory in Deng et al. (2018). We then used the $d_{\text{act}}$ values for the estimate of the fractional contribution of OA/BSOA to CCN number concentrations. Related changes in the manuscript and its SI are as follows.

**Texts S12 and S13:** They have been deleted because the use of the fixed $d_{\text{act}}$ value of 70 nm does not need to be justified anymore.

**Figs. 6 and 7 and Tables S9 and S12:** Values based on new $d_{\text{act}}$ have been applied. Note that all the changes of these values are within −2.22%–1.56% except those associated with BSOA during 0400–0800 JST when the volume fraction of BSOA was low (Fig. S18).

**First paragraph of Sect. 4.4:** The sentence "In two previous observations at the same site, the average CCN activation diameters of aerosols under 0.41 and 0.42 % SS were 71 and 68 nm, respectively (Kawana et al., 2017; Deng et al., 2018). Based on these facts, all the particles with $d_{\text{dry}}$ greater than 70 nm were assumed to be CCN active." has been changed to "The CCN activation diameters ($d_{\text{act}}$) for respective 2 h time bins of day at 0.4 % SS were calculated from reconstructed hygroscopicity parameter values of aerosol particles based on $\kappa$-Köhler theory (Text S11)." The

sentence "The diurnal variation of the CCN activation diameter was not considered for the estimate of $F_{CCN,OA}$ and $F_{CCN,BSOA}$ (Text S12)." has been omitted. The phrase "(Text S11)" has been omitted (line 16), and the expression "CCN activation diameter (70 nm)" has been changed to "$d_{act}$" (line 20).

**Last paragraph of Sect. 4.4:** The sentences "Furthermore, because fresh BSOA probably become aged after atmospheric transport, the influence of the aging of the estimated fresh BSOA (assuming $\kappa_{BSOA}$ was as large as that of $\kappa_{ROA}$ (Table S11)) on $F_{CCN,BSOA}$ was also evaluated. Here, the estimation of $F_{CCN,BSOA}$ in the aged condition ignored the possible change in the CCN activation diameter accompanying the aging processes (Text S13). The possible change in the aerosol size distribution accompanying the aging process was also not considered here." have been changed to "Furthermore, because fresh BSOA probably become aged after atmospheric transport, the influence of the aging of the estimated fresh BSOA (assuming $\kappa_{BSOA}$ for the calculation of $d_{act}$ and $F_{CCN,BSOA}$ was as large as that of $\kappa_{ROA}$ (Table S11)) on $F_{CCN,BSOA}$ was also evaluated. Here, the estimation of $F_{CCN,BSOA}$ in the aged condition ignored the possible change in the aerosol size distribution accompanying the aging process."

**Text S11:** The sentence "For the estimate, the observed aerosol particles were assumed to be internally mixed, and all the particles with $d_{dry}$ larger than 70 nm were assumed to be CCN active." has been changed to "For the estimate, the observed aerosol particles were assumed to be internally mixed, and the CCN activation diameters ($d_{act}$) under the condition of 0.4 % SS were derived using the reconstructed hygroscopicity parameters ($\kappa_{t,reconst,org}$ and $\kappa_{t,reconst,BSOA}$ for the analysis of the contributions of OA and BSOA, respectively). For the calculation of $d_{act}$, $\kappa$-Köhler theory was applied in the manner of Deng et al. (2018)." Because $d_{act}$ at 0.4 % SS is used now, the hygroscopicity parameter values of inorganic salts presented in Text S11 have also been modified.

**Other related changes:** All of them are associated with the changes of the values in Figs. 6 and 7 and are presented in the track-change mode version of the manuscript. They are mainly in Sects. 4.4 and 5, and the abstract.

Besides, the expression "(volume concentration less than $0.01 \times 10^{-6}$ cm$^3$ m$^{-3}$; Fig. S18)" in the caption of Fig. 7 has been modified to "(volume concentration less than 0.01 ×

$10^{-6}$ cm$^3$ m$^{-3}$)", because Fig. S18 does not present volume concentrations but present volume fractions.

We would like to note that the analysis of the contributions of OA and BSOA to CCN concentrations in Sect. 4.4 is not a "sensitivity" analysis to quantify the increase in CCN number concentrations by the presence of OA, as compared to the hypothetical case that OA was absent. In this type of sensitivity analysis, "fractional" contributions of OA and inorganics cannot be quantified because the sensitivity of CCN number concentrations on the presence/absence of OA and inorganics are not additive. In the present study, we quantified fractional 
[revised manuscript text omitted]
_{\text{org}}$ (TimeSizeReso$\kappa_{\text{org}}$), time-averaged and size-resolved $\kappa_{\text{org}}$ (SizeReso$\kappa_{\text{org}}$), size-averaged and time-resolved $\kappa_{\text{org}}$ (TimeReso$\kappa_{\text{org}}$), and time- and size-averaged $\kappa_{\text{org}}$ (Single$\kappa_{\text{org}}$) to that derived using the time- and size- resolved $\kappa_{\text{org}}$ (TimeSizeReso$\kappa_{\text{org}}$). (b) Diurnal variation of the ratios of the $F_{\text{CCN,BSOA}}$ derived using size-resolved $\kappa_{\text{BSOA}}$ (SizeReso$\kappa_{\text{BSOA}}$), size-averaged $\kappa_{\text{BSOA}}$ (SizeAver$\kappa_{\text{BSOA}}$), and aged size-resolved $\kappa_{\text{BSOA}}$ (AgedSizeReso$\kappa_{\text{BSOA}}$) to that derived using the size-resolved $\kappa_{\text{BSOA}}$ (SizeReso$\kappa_{\text{BSOA}}$). In panel (b), the condition of aged size-resolved $\kappa_{\text{BSOA}}$ assumes that the value of $\kappa_{\text{BSOA}}$ equals that of $\kappa_{\text{ROA}}$, and the data during 0400–0800 JST, when the concentration of BSOA was low (volume concentration less than $0.01 \times 10^{-6}$ cm$^3$ m$^{-3}$), are not presented (data are presented in Table S12).

Because obvious diurnal variations and size-dependence of $\kappa_{\text{org}}$ were found and because $\kappa_{\text{BSOA}}$ was also estimated to be size-dependent (Sect. 4.2), the sensitivities of the estimated $F_{\text{CCN,OA}}$ and $F_{\text{CCN,BSOA}}$ on the variations of $\kappa_{\text{org}}$ and $\kappa_{\text{
[revised manuscript text omitted]

**Text S1. Performance check of the DMAs**

Before and after the atmospheric observation, the accuracy of the sizing by three DMAs was assessed using standard size PSL particles (JSR SIZE STANDARD PARTICLES: SC-0055-D, SC-0100-D, and SC-032-S; Thermo Scientific[TM]: 3500A). The mode diameters from fittings for the measurement data (Kawana et al., 2014) were compared with the manufacturer warranty (Table S1), which is interpreted as prescribed ranges of mean diameter ± the expanded uncertainty ($k = 2$). The mode diameters after the atmospheric observation agreed with those before the observation within 0.84 %. The results obtained before the atmospheric observation are as follows. For DMA1, whereas the measured mode diameter of SC-0100-D was within the prescribed range, the measured mode diameters of SC-0055-D and SC-032-S were 1.0 % larger than the upper end of the prescribed range and 0.76 % lower than the lower end of the prescribed range, respectively. For DMA2, the measured mode diameter of SC-0055-D was 1.5 % larger than the upper end of the prescribed range, and the measured mode diameters of SC-0100-D, SC-032-S, and 3500A were 0.85, 3.2, and 2.2 % lower than the lower end of the prescribed ranges, respectively. For DMA3, the measured mode diameter of SC-0055-D was 6.9 % larger than the upper end of the prescribed diameter range, and the measured mode diameters for all three of the other PSL standards were within the prescribed ranges.

**Text S2. Performance check of HTDMA using ammonium sulfate particles**

Before the atmospheric observation, an aqueous solution of ammonium sulfate (AS) (99.999 % purity, Sigma-Aldrich) was nebulized and the generated aerosols were dried and introduced to the HTDMA to assess the difference in the sizing between the two DMAs under dry condition and to validate the control of RH in the HTDMA. The diameter setting for the measurements was the same as that of ambient particles (Sect. 2). The mean growth factors ($g_{f,m}$) of the AS particles under both dry and wet conditions were

retrieved using the same method as that for ambient particles (Sect. 3.1). The $g_{f,m}$ of the dry AS aerosol particles with diameters of 30–360 nm were 1.2–4.1 % deviated from unity (Table S2). The deviations were used to correct the difference of sizing between DMA1 and DMA2 for the $g_f$ of AS and ambient aerosol particles measured at 85 % RH. The respective $g_{f,m}$ of AS particles at 85 % RH with $d_{dry}$ of 30, 50, 70, 100, 200, 300, and 360 nm were 1.52, 1.54, 1.54, 1.54, 1.55, 1.57, and 1.59, which agree within 2.0 % with the calculated values ($g_{f,AS}$; Table S2) based on the Extended AIM Aerosol Thermodynamics Model II (E-AIM II, http://www.aim.env.uea.ac.uk/aim/model2/model2a.php; Clegg et al., 1998; Wexler and Clegg, 2002). The derivation of the hygroscopic growth factor of AS particles using the E-AIM II model is presented in Text S6.

**Text S3. PMF analysis of organic mass spectra**

The bulk mass spectra of organics observed in V-mode were subjected to PMF analysis (PMF Evaluation Tool v3.04A). For the analysis, high resolution fragment ions with signal to noise ratio (SNR) smaller than 0.2 were omitted, fragments with SNR in the range of 0.2–2 were down-weighted by a factor of three, and fragments related to $CO_2$ (i.e., $CO_2$, CO, $H_2O$, HO, and O) were down-weighted so that fragment $CO_2$ only contributed once. The obtained two-factor solution, with a more-oxygenated OA component (MOOA) and a less-oxygenated OA component (LOOA), with seed = 1 and fpeak = 0, was adopted for the explanation of OA composition. The two-factor PMF results are summarized in Fig. S1.

**Text S4. Data screening methods**

All the data except for the meteorological data obtained during the atmospheric observation were subjected to the screening to exclude data under possible influence from intermittent local anthropogenic emissions. For the SMPS data with 5 min resolution, and the BC and gaseous species data with 30 min resolution, if

the data value at one point was more than 30 % deviated from both the former data point and the next data point in the time series, the data at that time point was deleted. The chemical composition data derived from AMS measurements with 30 min resolution were deleted whenever the BC data were deleted. The hygroscopic growth data derived from the HTDMA measurements were deleted whenever the SMPS data were deleted. Furthermore, if the total count of particles measured using the CPC in the HTDMA in the diameter range of 0.80–2.2 times of $d_{dry}$ (or 0.80–2.0 times of $d_{dry}$ for particles with $d_{dry}$ of 360 nm) during a single scan was less than eight, the HTDMA data was not used, either. Here, the $d_{dry}$ were corrected for the difference of sizing between DMA1 and DMA2.

**Text S5. Derivation of size-resolved volume fractions of the chemical components**

The size-resolved volume fractions of inorganic salts ($\varepsilon_i$), organics ($\varepsilon_{org}$), and BC ($\varepsilon_{BC}$) were calculated as follows. First, BC was assumed to be internally mixed with non-refractory aerosol components and to have the same mass-size distribution as OA. The aerosol particles were assumed to be spherical and without voids. Using the PToF mode data from the AMS, the mass concentrations of aerosol components in the vacuum aerodynamic diameter ($d_{va}$) ranges that are ~1.0 (0.98–0.99) to 2.0 times of $d_{dry}$ were obtained: the ranges of $d_{va}$ for particles with $d_{dry}$ of 100, 200, 300, and 360 nm were 98–197, 197–395, 295–589, and 353–707 nm, respectively. Second, the mole amounts of sulfate, nitrate, and ammonium in 1 $m^3$ of air were derived. Third, the amount of ammonium nitrate (AN), ammonium sulfate (AS), letovicite (LET), ammonium hydrogen sulfate (AHS), and sulfuric acid (SA) per mole in 1 $m^3$ of air was determined, assuming that nitrate was fully neutralized with ammonium, and that sulfate could present in the form of AS, LET, AHS, and/or SA according to the amount of remaining ammonium. The detected non-refractory chloride was not considered because of its low concentration compared with the concentrations of the other non-refractory components. The contribution of sea salt and minerals to the sub-micrometer aerosol

particles was likely small (Han et al., 2014; Deng et al., 2018) and was also not considered. Fourth, the volumes of BC, OA, and LET were derived using their respective densities and the volumes of AN, AS, AHS, and SA were derived using their respective molar volumes. The density of BC was assumed to be 1.77 g cm$^{-3}$ (Park et al., 2004). The density of organics ($\rho_{org}$) was estimated to be $1.32 \pm 0.09$ g cm$^{-3}$ using the O:C and H:C ratios of organics derived from AMS measurements (Kuwata et al., 2012), and the mean value of 1.32 g cm$^{-3}$ was adopted for this study. The density of LET was assumed to be 1.83 g cm$^{-3}$ (Padró et al., 2010). The molar volumes of AN, AS, AHS, and pure liquid SA, which are the same as the ones used in the E-AIM II model, were adopted in this study. Finally, the volume fractions of each species were obtained.

**Text S6. Derivation of $g_{f,AS}$ and $\kappa_i$ using the E-AIM II model**

The hygroscopic growth factor of pure ammonium sulfate particles ($g_{f,AS}$) and the hygroscopicity parameter of each inorganic salt ($\kappa_i$) at 85 % RH were derived based on the output of the E-AIM II model and the $\kappa$–Köhler equation (Petters and Kreidenweis, 2007) as follows.

The water activity ($a_w$) range from 0.8000 to 0.8499 at a resolution of 0.0001 was applied to the E-AIM II model for unit mole of each inorganic salt at the temperature of 294 K. For the calculation, the partition of HNO$_3$, NH$_3$, and H$_2$SO$_4$ into the vapor phase was prevented and the formation of solid AS was also prevented. The hygroscopic growth factor ($g_f$) corresponding to each $a_w$ was derived from the ratio of total wet volume ($V_{wet}$) to the dry molar volume of the pure salt. The $\kappa_i$ corresponding to each $a_w$ was derived from Equation 2 of Petters and Kreidenweis (2007). The exponential part of the $\kappa$–Köhler equation was derived on the assumption that the partial molar volume of water equals the molar volume of pure water. Here, $a_w$ was calculated based on the relationship that the RH above the particle surface equals the product of the $a_w$ and the exponential part of the $\kappa$–Köhler equation representing the Kelvin effect. For particles

with respective $d_{dry}$, an $a_w$ value at which the RH above the particle surface was nearest to 85 % ($a_{w,85}$) was obtained by applying a wet diameter represented by the product of $d_{dry}$ and $g_f$. The hygroscopicity parameter value at $a_{w,85}$ was defined as $\kappa_i$ (Table S3). For AS, the $g_f$ at $a_{w,85}$ was defined as $g_{f,AS}$ (Table S2). For the derivation of $\kappa_i$, the surface tension of the solution was applied, whereas for the derivation of $\kappa_t$ (Sect. 3.1) the surface tension of pure water was applied. The uncertainty of the surface tension should not introduce large uncertainty in the derived $\kappa_{org}$ because the difference of $\kappa_i$ obtained using the surface tension of pure water and that obtained using the surface tension of the solution was small (within 0.38 %, Table S3).

**Text S7. Determination of the range of $d_{va}$ for the calculation of $\kappa_{org}$**

The PToF mode AMS data over a lower $d_{va}$ range was subjected to low signal intensity (Fig. S2) when it was adopted for the derivation of size-resolved $\kappa_{org}$. To determine the applicable range of $d_{va}$ for this study, the mean mass-size distribution data of OA during the entire study period was compared to that of a baseline region (Fig. S3). The baseline region here is the transition region between the regions dominated by gaseous species and particle signals of OA, and corresponds to the PToF time region of 0.00135458–0.00150458 s. To eliminate data under the strong influence of the signals from gaseous species, the $d_{va}$ of 98 nm (where the mass ratio of the mean observed OA to mean baseline OA became greater than three) was adopted as the lower limit of the $d_{va}$ range for the deviation of $\kappa_{org}$. In Fig. S3, the mean mass-size distribution of OA for filtered air (collected by connecting a HEPA filter (TSI) to the inlet tubing outside the instrument room) is also presented for comparison. Only in the $d_{va}$ range of 90–716 nm, the mean values of the observed OA mass in each $d_{va}$ bin were greater than that of OA for the filtered air. This also indicates the strong influence of the signals from gaseous species at $d_{va}$ smaller than ~90 nm. Because the OA data of filtered air was noisy, it was not used for the determination of the applicable $d_{va}$ range.

**Text S8. Derivation of size-resolved PMF factors**

To explain the diurnal variation and size-dependence of $\kappa_{org}$ from the compositional characteristics of OA, size-resolved PMF factors (LOOA and MOOA) were derived through multivariable linear regression analysis as follows.

First, the fragment profiles of LOOA and MOOA from bulk OA mass spectra were converted to the profiles in unit m/z resolution by summing up the intensity of the fragment ion signals at the same unit m/z. Then, for each time period in 2 h time resolution and for each PToF size bin $i$, the contributions from LOOA ($a_i$) and MOOA ($b_i$) were derived using the Solver function in Microsoft Excel by minimizing the value of the following formula:

$$\sum_{j=12}^{115} [x_{ij} - (a_i f_{1j} + b_i f_{2j})]^2$$

Where $x_{ij}$ is the measured signal intensity in size bin $i$ at m/z $= j$, and $f_{1j}$ and $f_{2j}$ are the respective normalized signal intensity of LOOA and MOOA at m/z $= j$. The unit of $x_{ij}$, $a_i$, and $b_i$ was µg m$^{-3}$, whereas $f_{1j}$ and $f_{2j}$ were dimensionless.

The size-resolved LOOA and MOOA were derived for the $d_{va}$ range from $\leq 10$ nm to around 900 nm (Fig. S6). However, only the $d_{va}$ range above 98 nm was adopted for the analysis. This is because the uncertainty of the contributions from LOOA and MOOA in the lower $d_{va}$ range was presumably relatively large, given the low organic signal intensity (Fig. S3) and high residual to measured OA mass ratio (Fig. S6).

**Text S9. Assessment of the contributions of BSOA and anthropogenic OA to the enhancement of OA in the daytime**

The contributions of BSOA and anthropogenic OA to the enhancement of OA in the daytime (in relation to the background period) were assessed using BC as a tracer of OA that did not come from BSOA

formation (non-BSOA-OA). Here, the non-BSOA-OA was considered the sum of regional OA and other anthropogenic OA. The diurnal variation data on the mass concentration of BC was scaled to represent the diurnal variation of non-BSOA-OA. For the scaling, the observed OA during the background period (i.e., 0600–0800 JST, when the daily minima of $m_{\mathrm{org}}$ appeared) was assumed to be composed only of non-BSOA-OA. The scaling factor was calculated to be 36.5. The mass concentrations of non-BSOA-OA ($m_{\mathrm{non\text{-}BSOA,bulk}}$) and BSOA ($m_{\mathrm{BSOA,bulk}}$) were then estimated using the following equations.

$$m_{\mathrm{non\text{-}BSOA,bulk}} = m_{\mathrm{BC}} \times 36.5 \qquad\qquad\qquad (S1)$$

$$m_{\mathrm{BSOA,bulk}} = m_{\mathrm{org}} - m_{\mathrm{non\text{-}BSOA,bulk}} \qquad\qquad\qquad (S2)$$

Note that $m_{\mathrm{BSOA,bulk}}$ and $m_{\mathrm{non\text{-}BSOA,bulk}}$ are different from the mass concentrations of the BSOA and ROA defined from the size-resolved LOOA/MOOA data (Sect. 4.2.2 and Text S10). The $m_{\mathrm{BSOA,bulk}}$ may be negatively biased because the charring of OA during the PSAP measurement may have resulted in a positive bias of $m_{\mathrm{BC}}$. The increase of the OA mass concentration in the daytime ($m_{\mathrm{org,ENH}}$) was estimated by subtracting the $m_{\mathrm{org}}$ during the background period from that during the period of interest. The ratio of $m_{\mathrm{BSOA,bulk}}$ to $m_{\mathrm{org,ENH}}$ was in the range 0.6–0.9 during 1200–1600 JST, and it was in the range 0.1–0.5 during 1600–2030 JST (Fig. S10). The result indicates that BSOA was the main contributor to the enhancement of OA at least during 1200–1600 JST. In a later period, a larger contribution from anthropogenic OA is not ruled out.

**Text S10. Derivation of $v_{\mathrm{BSOA}}$ and $v_{\mathrm{ROA}}$ from $v_{\mathrm{LOOA}}$ and $v_{\mathrm{MOOA}}$**

Because the observed OA can be assumed to be contributed either by LOOA and MOOA, or by BSOA and ROA, the sum of LOOA and MOOA should equal the sum of BSOA and ROA. The size-resolved $v_{\mathrm{BSOA}}$ and $v_{\mathrm{ROA}}$ can be derived from the size-resolved $v_{\mathrm{LOOA}}$ and $v_{\mathrm{MOOA}}$. From the analysis in Sect. 4.2.2, 0.97 (0.52), 0.88 (0.53), 0.80 (0.50), and 0.79 (0.45) of the volume of BSOA (ROA) for particles with $d_{\mathrm{dry}}$

of 100, 200, 300, and 360 nm were assigned to LOOA, and the balances were assigned to MOOA. The relations are expressed by the equations:

$$v_{BSOA} + v_{ROA} = v_{LOOA} + v_{MOOA} \qquad (S3)$$

$$a \times v_{BSOA} + b \times v_{ROA} = v_{LOOA} \qquad (S4)$$

Where $v_{BSOA}$, $v_{ROA}$, $v_{LOOA}$, and $v_{MOOA}$ are the size-resolved volume concentrations of BSOA, ROA, LOOA, and MOOA, respectively, and $a$ and $b$ represent the size-resolved volume fractions of BSOA and ROA, respectively, that were assigned to LOOA. The volume concentrations of BSOA and ROA were estimated using the equations:

$$v_{BSOA} = [(1-b)/(a-b)] \times v_{LOOA} - [b/(a-b)] \times v_{MOOA} \qquad (S5)$$

$$v_{ROA} = [(a-1)/(a-b)] \times v_{MOOA} + [a/(a-b)] \times v_{LOOA} \qquad (S6)$$

Equations (S5) and (S6) were used to estimate the contributions of BSOA to aerosol water uptake and to the CCN number concentration (Sect. 4.4 and Text S11).

**Text S11. Estimation of the contributions of OA and BSOA to CCN concentrations**

The contributions of OA and BSOA to the CCN number concentration were estimated from their size-resolved fractional contributions to aerosol water uptake and from the measured aerosol number-size distributions. The analysis was performed for diurnal variation data with 2 h resolution. A schematic of the estimate is presented in Fig. S17. For the estimate, the observed aerosol particles were assumed to be internally mixed, and the CCN activation diameters ($d_{act}$) under the condition of 0.4 % SS were derived using the reconstructed hygroscopicity parameters ($\kappa_{t,reconst,org}$ and $\kappa_{t,reconst,BSOA}$ for the analysis of the contributions of OA and BSOA, respectively). For the calculation of $d_{act}$, $\kappa$-Köhler theory was applied in the manner of Deng et al. (2018). The contributions of OA (BSOA) to the water uptake of particles with $d_{dry}$ of 100, 200, 300, and 360 nm were applied for the diameter ranges of $d_{act}$–150, 150–250, 250–330,

330–430 nm, respectively. For the diameter ranges larger than 430 nm, the CCN number concentration contributed by OA (BSOA) was not considered because of the low aerosol number concentrations. In each diameter range, the fractional contribution of OA (BSOA) to $dN_{CN}/dlogd_{dry}$ equals the fractional contribution of OA (BSOA) to the total aerosol water uptake, which was represented as the product of the volume fraction of OA (BSOA) and $\kappa_{org}$ ($\kappa_{BSOA}$) divided by $\kappa_i$ [i.e., $\varepsilon_{org}\kappa_{org}/\kappa_i$ ($\varepsilon_{BSOA}\kappa_{BSOA}/\kappa_i$)]. The total fractional contribution of OA (BSOA) to the total CCN number concentration, $F_{CCN,OA}$ ($F_{CCN,BSOA}$), equals the integration of the product of the water uptake fraction and $dN_{CN}/dlogd_{dry}$ above the assumed CCN activation diameter (70 nm), divided by the total CCN number concentration.

For the above analysis, the water uptake of each aerosol component was represented by the product of the volume fraction of the aerosol component ($\varepsilon_i$) and its hygroscopicity ($\kappa_i$), that is, $\varepsilon_i\kappa_i$. The mean $\varepsilon_i$ in each 2 h time section of the day was derived as follows. First, the mean values of the volume concentrations of each inorganic species, organics, and organic fractions (AN, AS, LET, AHS, SA, BC, OA, LOOA, and MOOA; $\bar{v}_i$) were calculated from the 2 h resolution data. Second, the mean volume concentrations of LOOA ($\bar{v}_{LOOA}$) and MOOA ($\bar{v}_{MOOA}$) were scaled so that their sum equals the mean volume concentration of OA ($\bar{v}_{OA}$). Third, the mean volume concentrations of BSOA ($\bar{v}_{BSOA}$) and ROA ($\bar{v}_{ROA}$) were estimated using Eqs. (S5) and (S6). Then, $\varepsilon_i$ was calculated directly from those $\bar{v}_i$. The $\kappa$ of the inorganic salts under the condition of 0.4 % SS, and at the temperature of the HTDMA measurement in this study, were used to consider the difference of $\kappa$ between sub- and super-saturated water vapor conditions. Here, the $\kappa$ for AN, AS, LET, AHS, and SA were calculated to be 0.72, 0.59, 0.62, 0.61, and 0.65, respectively, following the method in Deng et al. (2018). The difference of the $\kappa$ of organics under sub- and super-saturated conditions was not considered. The $\kappa$ values of OA and BSOA used for the calculation are presented in Tables S10 and S11, respectively. The contributions of OA and BSOA to the water uptake were calculated as $\varepsilon_{org}\kappa_{org}/\kappa_{i,reconst,org}$ and $\varepsilon_{BSOA}\kappa_{BSOA}/\kappa_{i,reconst,BSOA}$, respectively, where,

$$\kappa_{i,\text{reconst,org}} = \varepsilon_{AN}\kappa_{AN} + \varepsilon_{AS}\kappa_{AS} + \varepsilon_{LET}\kappa_{LET} + \varepsilon_{AHS}\kappa_{AHS} + \varepsilon_{SA}\kappa_{SA} + \varepsilon_{org}\kappa_{org} \qquad (S7)$$

$$\kappa_{i,\text{reconst,BSOA}} = \varepsilon_{AN}\kappa_{AN} + \varepsilon_{AS}\kappa_{AS} + \varepsilon_{LET}\kappa_{LET} + \varepsilon_{AHS}\kappa_{AHS} + \varepsilon_{SA}\kappa_{SA} + \varepsilon_{BSOA}\kappa_{BSOA} + \varepsilon_{ROA}\kappa_{ROA}$$

$$(S8)$$

[Figure]

**Figure S1:** Summary of the two-factor result from the PMF analysis: (a) organic mass spectra of LOOA and MOOA colored according to ion groups (fpeak = 0 and SEED = 1), and the atomic ratios of O to C, H to C, and N to C, as well as OM to OC ratio for each factor; (b) Q/Qexpected as a function of the number of factors, where Q is the sum of the weighted squared residuals and Qexpected is the expected Q value; (c) Q/Qexpected as a function of the fpeak values with SEED = 1 and the number of factor = 2; (d) Q/Qexpected as a function of the SEED values with fpeak = 0 and the number of factor = 2; (e) distributions of the scaled residual for each m/z (fpeak = 0 and SEED = 1); time series of (f) the measured organic mass concentrations and those reconstructed (= LOOA + MOOA) (fpeak = 0 and SEED = 1), and (g) residual OA (= measured − reconstructed) (fpeak = 0 and SEED = 1).

[Figure]

**Figure S2:** Mean mass-size distributions of organics (OA), sulfate ($SO_4$), ammonium ($NH_4$), nitrate ($NO_3$), and chloride (Chl) in (a) linear and (b) logarithmic scales over the entire study period.

[Figure]

**Figure S3:** d*M*/d*PToF* (*M* and *PToF* here refer to mass concentration and PToF time, respectively) versus PToF time for the observed OA and OA measured by placing a HEPA filter in the inlet tubing outside the instrument room (zero OA), averaged for the entire study period. The region shaded in pink indicates the PToF time region chosen as a baseline for evaluation of the effective PToF diameter range. (Note that this is not the DC marker region.) The dark solid curve is the absolute value of the ratio of the observed OA at respective PToF time to the mean of the observed OA in the baseline region ($R_{obs/base}$). The blue dash-dotted line indicates the $R_{obs/base}$ value of three. The vacuum aerodynamic diameter that corresponds to the PToF time is presented on the top axis.

[Figure]

**Figure S4:** $\kappa_{org}$ of 100 nm particles derived based on the chemical composition in the vacuum aerodynamic diameter ($d_{va}$) range of 69–138 nm versus that of 98–197 nm. Data points with $\kappa_{org}$ smaller than −0.50 or $\kappa_{org}$ higher than 0.60 are not presented.

[Figure]

**Figure S5:** $\kappa_t$ versus $\varepsilon_{org}$ for particles with $d_{dry}$ of 100, 200, 300, and 360 nm for the entire study period. In each panel, marks and a solid line represent individual data and the corresponding linear regression line, respectively. The correlation coefficient ($r$) of each is also presented.

[Figure]

**Figure S6:** Mean mass-size distributions of (a) residual, (b) LOOA, and (c) MOOA over the entire study period. The residual is the difference between the measured and reconstructed (i.e., LOOA + MOOA) mass concentrations of OA in each size bin. The error bars indicate the standard deviation. The ratios of the sum of the absolute value of the residual to the measured mass concentration of OA are superimposed in panel (a) (right axis).

[Figure]

**Figure S7:** Time series of (a) air temperature and relative humidity (RH), (b) precipitation and solar radiation, (c) mass concentration of BC, and mixing ratios of (c) CO, (d) $O_3$, (e) $CO_2$, and (f) NO, $NO_2$, and $NO_x$.

[Figure]

**Figure S8:** Diurnal variations of (a) air temperature and relative humidity (RH), (b) solar radiation and the mixing ratios of $O_3$, and (c) mixing ratios of NO, $NO_2$, $NO_x$, and CO over the entire study period.

[Figure]

**Figure S9:** Five-day backward air mass trajectories from 500 m agl (above ground level) over Wakayama Forest Research Station at one-day intervals. The arrival time of the air masses at the study site was 1500 JST. The solid circle denotes the location of the study site. Solid trajectories are for days when more than 22 of the 24 hourly trajectories are from terrestrial regions. Dashed trajectories are for days when more

than 21 of the 24 hourly trajectories are from the North Pacific. Dotted trajectories are for days when 10 of the 24 hourly trajectories are from the North Pacific. The map is based on GSHHG 2.3.4; the shoreline polygon data at crude resolution is used. We consider an air mass is from the North Pacific if the trajectory never passes over terrestrial area that appears on the map before it reaches the Kii Peninsula. The trajectories were produced using NOAA's HYSPLIT atmospheric transport and dispersion modeling system (Draxler and Hess, 1998).

[Figure]

**Figure S10.** Diurnal variation of the mass concentrations of OA ($m_{org}$), BC ($m_{BC}$), and non-BSOA-OA ($m_{non-BSOA,bulk}$; Text S9) during the entire study period. The left-slash pattern represents the background period. As an example, the vertical bars represent the estimates of the total enhancement of OA ($m_{org,ENH}$; blue) and the enhancement contributed by BSOA ($m_{BSOA,bulk}$; gray) for the period 1400–1600 JST (right slash pattern).

[Figure]

**Figure S11:** Time series of the probability distribution functions of the hygroscopic growth factors ($g_f$-PDF) of aerosol particles with $d_{dry}$ of (a) 30, (b) 70, (c) 100, (d) 300, and (e) 360 nm.

[Figure]

**Figure S12:** Time series of the mean growth factors ($g_{f,m}$) of aerosol particles with different dry diameters. The mean ± SD values over the entire study period for each diameter are also presented.

[Figure]

**Figure S13:** Diurnal variation of the size-resolved volume fractions of total inorganic salts ($\varepsilon_{\text{inorgsalt}}$) during the entire study period.

[Figure]

**Figure S14:** The $\kappa_{\text{org}}$ versus $v_{\text{LOOA}}/(v_{\text{LOOA}}+v_{\text{MOOA}})$ for particles with $d_{\text{dry}}$ of 300 and 360 nm. The time resolution of the data is 2 h. In each panel, marks and a solid line represent individual data and the corresponding linear regression line, respectively. The regression equation and correlation coefficient of each are also presented. Only data with $\varepsilon_{\text{org}}$ greater than 0.40 are used.

[Figure]

**Figure S15:** The $\kappa_{org}$ reconstructed using $\kappa_{LOOA}$, $\kappa_{MOOA}$, and size-resolved $v_{LOOA}/(v_{LOOA}+v_{MOOA})$ (reconstructed $\kappa_{org}$) versus the $\kappa_{org}$ derived from measured $\kappa_t$ and aerosol chemical composition (measured $\kappa_{org}$; Sect. 3.2). In panel (a), markers represent size-resolved diurnal variation data at 2 h resolution, solid lines are linear regression lines for particles with respective diameters, and the dashed line is the linear regression line for all 100–360 nm particles. In panel (b), markers represent the size-resolved mean $\kappa_{org}$ during 1200–2000 JST, and the solid line is the linear regression line for the size-resolved mean $\kappa_{org}$. Respective regression equations and coefficients of determination ($r^2$) are also presented. Only $\kappa_{org}$ data with $\varepsilon_{org}$ greater than 0.40 are used for the comparison.

[Figure]

**Figure S16:** Diurnal variation of the volume concentrations of LOOA and MOOA and the mass concentration of sulfate for particles with $d_{dry}$ of 100, 200, 300, and 360 nm over the entire study period. The scaled sulfate represents the diurnal variation of ROA that was contributed by LOOA (left panels) or MOOA (right panels). The scaling factor for the scaled sulfate in each panel is the mean volume concentration of OA during 0600–0800 JST, divided by the mean mass concentration of sulfate in the same period. The volume concentrations of LOOA and MOOA were derived from the respective mass concentrations (Text S8). The densities of LOOA and MOOA were calculated using their O:C and H:C ratios following Kuwata et al. (2012) and were 1.24 and 1.54 g cm$^{-3}$, respectively.

[Figure]

**Figure S17:** Estimate of the contributions of BSOA to the CCN number concentration from the viewpoint of its size-resolved contribution to the aerosol water uptake. The solid line indicates the mean aerosol number-size distribution during the entire study period. Shaded areas in green represent the fraction of CCN contributed by BSOA and in red, that contributed by other components assuming a CCN activation diameter of 70 nm (Text S11).

[Figure]

**Figure S18:** The diurnal variation of the volume fractions of BSOA ($\varepsilon_{BSOA}$) for particles with $d_{dry}$ of 100, 200, 300, and 360 nm over the entire study period (Text S11).

[Figure]

**Figure S19:** Box and whiskers plot of the diurnal variation of the O:C ratios of bulk OA (only data with $m_{org} > 0.3$ µg m$^{-3}$ are included) for the entire study period. The horizontal lines in the boxes indicate the median values, boundaries of the boxes indicate the 25[th]- and 75[th]-percentiles, and the whiskers indicate the highest and lowest values. The cross symbols in the boxes indicate the mean values.

**Table S1: Mode diameters [a] of PSL size standards measured by DMAs in the HTDMA (DMA1 and DMA2) and the SMPS (DMA3) (mean ± SD, nm)**

| Manufacturer warranty | DMA1 | | DMA2 | | DMA3 | |
|---|---|---|---|---|---|---|
| | Before [c] | After [d] | Before [c] | After [d] | Before [c] | After [d] |
| **55 (± 1) [b]** | 56.6 ± 0.4 | 56.2 ± 0.4 | 56.8 ± 0.4 | - | 59.9 ± 0.2 | - |
| **100 (± 3) [b]** | 98.0 ± 0.2 | 98.4 ± 0.1 | 97.0 ± 0.1 | 96.2 ± 0.2 | 101.2 ± 0.3 | 101.1 ± 0.1 |
| **309 (± 9) [b]** | 297.7 ± 0.8 | 298.0 ± 0.4 | 290.4 ± 1.2 | - | 303.6 ± 0.3 | - |
| **498 (± 9) [b]** | - | - | 478.4 ± 5.8 | - | 499.6 ± 4.4 | - |

[a] The mean ± SD of the mode diameters from fittings are presented (unit: nm).
[b] Mean diameter (± the expanded uncertainty; $k = 2$).
[c] Before the atmospheric observations.
[d] After the atmospheric observations.

**Table S2: The $g_{f,m}$ of ammonium sulfate (AS) particles measured under dry condition ($g_{f,m,dryAS}$) and at 85 % RH ($g_{f,m,wetAS}$), and calculated $g_f$ of AS particles at 85 % RH ($g_{f,AS}$)**

| $d_{dry}$ (nm) | 30 | 50 | 70 | 100 | 200 | 300 | 360 |
|---|---|---|---|---|---|---|---|
| $g_{f,m,dryAS}$ | 0.959 | 0.976 | 0.984 | 0.985 | 0.988 | 0.982 | 0.981 |
| $g_{f,m,wetAS}$[a] | 1.52 | 1.54 | 1.54 | 1.54 | 1.55 | 1.57 | 1.59 |

| | | | | | | | |
|---|---|---|---|---|---|---|---|
| $g_{f,AS}$[a] | 1.49 | 1.52 | 1.54 | 1.55 | 1.57 | 1.57 | 1.57 |
| **Difference (%)**[b] | 2.0 | 1.3 | 0 | −0.65 | −1.3 | 0 | 1.3 |

[a] Corrected for the difference of sizing between DMA1 and DMA2.
[b] $((g_{f,m,wetAS} - g_{f,AS}) / g_{f,AS}) \times 100$.

**Table S3: The $\kappa$ values of inorganic salts ($\kappa_i$) at 85 % RH derived using the surface tension of the solution and of pure water**

| | $\kappa_i$, with surface tension of solution | | | | $\kappa_i$, with surface tension of pure water | | | |
|---|---|---|---|---|---|---|---|---|
| $d_{dry}$ (nm) | **100** | **200** | **300** | **360** | **100** | **200** | **300** | **360** |
| **AN** | 0.553 | 0.555 | 0.555 | 0.556 | 0.553 | 0.555 | 0.556 | 0.556 |
| **AS** | 0.533 | 0.527 | 0.525 | 0.524 | 0.531 | 0.526 | 0.524 | 0.524 |
| **LET** | 0.550 | 0.545 | 0.543 | 0.543 | 0.549 | 0.544 | 0.543 | 0.543 |
| **AHS** | 0.612 | 0.607 | 0.605 | 0.605 | 0.612 | 0.607 | 0.605 | 0.605 |
| **SA** | 0.972 | 0.959 | 0.955 | 0.953 | 0.971 | 0.959 | 0.955 | 0.953 |

**Table S4: Data in Fig. 3 of the main manuscript** ("DataInFigure3ofTheManuscript.xlsx")**.**

**Table S5: Data in Fig. 4 of the main manuscript** ("DataInFigure4ofTheManuscript.xlsx")**.**

**Table S6: Data in Fig. 5 of the main manuscript** ("DataInFigure5ofTheManuscript.xlsx")**.**

**Table S7: Comparisons of $\kappa_{org}$ and $v_{LOOA}/(v_{LOOA}+v_{MOOA})$ between particles with different $d_{dry}$**

| $d_{dry}$ of particles to compare (nm) | 1200–2000 JST | | | | 2000–1200 JST | | | |
|---|---|---|---|---|---|---|---|---|
| | $\kappa_{org}$ | | $v_{LOOA}/(v_{LOOA}+v_{MOOA})$ | | $\kappa_{org}$ | | $v_{LOOA}/(v_{LOOA}+v_{MOOA})$ | |
| | Diff[a] | p-value[c] | Diff[b] | p-value[c] | Diff[a] | p-value[c] | Diff[b] | p-value[c] |
| **200 vs 100** | 0.06 | <0.01 | −0.06 | 0.02 | 0.04 | <0.01 | −0.01 | 0.65 |
| **300 vs 200** | <0.01 | 0.71 | −0.08 | <0.01 | <0.01 | 0.31 | −0.06 | <0.01 |
| **360 vs 300** | 0.02 | 0.15 | −0.04 | <0.01 | −0.02 | 0.02 | −0.05 | <0.01 |
| **360 vs 200** | 0.02 | 0.07 | −0.11 | <0.01 | −0.02 | 0.07 | −0.11 | <0.01 |

[a] The mean of (the $\kappa_{org}$ of particles with relatively large $d_{dry}$ – the $\kappa_{org}$ of particles with relatively small $d_{dry}$).
[b] The mean of (the $v_{LOOA}/(v_{LOOA}+v_{MOOA})$ of particles with relatively large $d_{dry}$ – the $v_{LOOA}/(v_{LOOA}+v_{MOOA})$ of particles with relatively small $d_{dry}$).
[c] From 10 % two-sided t-test for the significance of the difference of Diff from zero. Low values indicate significant differences.

**Table S8: Diurnal variation of $\kappa_t$ and $\kappa_{org}$ at 2 h resolution, and their mean and SD for the entire period**

| | $\kappa_t$ | | | | | | | $\kappa_{org}$ | | | |
|---|---|---|---|---|---|---|---|---|---|---|---|
| $d_{dry}$ (nm) | **30** | **50** | **70** | **100** | **200** | **300** | **360** | **100** | **200** | **300** | **360** |
| **0000–0200 JST** | 0.16 | 0.20 | 0.22 | 0.24 | 0.34 | 0.37 | 0.36 | 0.16 | 0.23 | 0.25 | 0.21 |
| **0200–0400 JST** | 0.22 | 0.21 | 0.22 | 0.24 | 0.34 | 0.37 | 0.35 | 0.16 | 0.20 | 0.21 | 0.18 |

| | | | | | | | | | | | |
|---|---|---|---|---|---|---|---|---|---|---|---|
| **0400–0600 JST** | 0.18 | 0.21 | 0.22 | 0.25 | 0.34 | 0.38 | 0.39 | 0.16 | 0.21 | 0.24 | 0.23 |
| **0600–0800 JST** | 0.21 | 0.22 | 0.21 | 0.25 | 0.35 | 0.38 | 0.36 | 0.18 | 0.22 | 0.24 | 0.19 |
| **0800–1000 JST** | 0.15 | 0.18 | 0.21 | 0.22 | 0.32 | 0.33 | 0.37 | 0.16 | 0.19 | 0.16 | 0.18 |
| **1000–1200 JST** | 0.13 | 0.16 | 0.16 | 0.19 | 0.29 | 0.33 | 0.34 | 0.12 | 0.17 | 0.15 | 0.12 |
| **1200–1400 JST** | 0.090 | 0.14 | 0.14 | 0.16 | 0.26 | 0.32 | 0.34 | 0.11 | 0.16 | 0.19 | 0.18 |
| **1400–1600 JST** | 0.083 | 0.13 | 0.13 | 0.16 | 0.24 | 0.28 | 0.33 | 0.10 | 0.15 | 0.16 | 0.20 |
| **1600–1800 JST** | 0.10 | 0.14 | 0.14 | 0.17 | 0.25 | 0.28 | 0.32 | 0.089 | 0.18 | 0.16 | 0.19 |
| **1800–2000 JST** | 0.13 | 0.15 | 0.16 | 0.19 | 0.27 | 0.31 | 0.32 | 0.12 | 0.18 | 0.18 | 0.19 |
| **2000–2200 JST** | 0.14 | 0.18 | 0.18 | 0.21 | 0.29 | 0.33 | 0.34 | 0.14 | 0.19 | 0.20 | 0.17 |
| **2200–0000 JST** | 0.12 | 0.19 | 0.20 | 0.23 | 0.31 | 0.35 | 0.36 | 0.16 | 0.19 | 0.20 | 0.17 |
| **Mean for entire period** | 0.12 | 0.18 | 0.18 | 0.21 | 0.30 | 0.34 | 0.35 | 0.13 | 0.18 | 0.19 | 0.19 |
| **SD for entire period** | 0.079 | 0.090 | 0.089 | 0.094 | 0.10 | 0.087 | 0.086 | 0.11 | 0.085 | 0.084 | 0.11 |

**Table S9: Data in Fig. 6 of the main manuscript.**

| Hour of the day | $F_{CCN,OA}$ (%) | $F_{CCN,BSOA}$ (%; fresh BSOA) | $F_{CCN,BSOA}$ (%; aged BSOA) |
|---|---|---|---|
| 0000–0200 JST | 44.7 | 5.81 | 10.6 |
| 0200–0400 JST | 44.9 | 2.67 | 4.89 |
| 0400–0600 JST | 40.4 | 0.249 | 0.356 |
| 0600–0800 JST | 43.2 | $-3.63 \times 10^{-3}$ | -0.0326 |
| 0800–1000 JST | 44.3 | 5.50 | 10.1 |
| 1000–1200 JST | 42.0 | 12.7 | 21.3 |
| 1200–1400 JST | 51.8 | 25.8 | 39.4 |
| 1400–1600 JST | 51.7 | 27.8 | 42.2 |
| 1600–1800 JST | 47.0 | 24.5 | 38.2 |
| 1800–2000 JST | 45.8 | 19.6 | 31.8 |
| 2000–2200 JST | 49.4 | 16.3 | 27.3 |
| 2200–0000 JST | 45.4 | 4.99 | 9.09 |

**Table S10: Different assumptions of $\kappa_{org}$ for the prediction of $F_{CCN,OA}$**

| $d_{dry}$ (nm) | TimeSize $\kappa_{org}$[a] | | | | SizeReso $\kappa_{org}$[b] | | | | TimeReso $\kappa_{org}$[c] | | | | Single $\kappa_{org}$[d] | | | |
|---|---|---|---|---|---|---|---|---|---|---|---|---|---|---|---|---|
| | 100 | 200 | 300 | 360 | 100 | 200 | 300 | 360 | 100 | 200 | 300 | 360 | 100 | 200 | 300 | 360 |
| 0000–0200 JST | 0.16 | 0.23 | 0.25 | 0.21 | | | | | | | 0.21 | | | | | |
| 0200–0400 JST | 0.16 | 0.20 | 0.21 | 0.18 | | | | | | | 0.19 | | | | | |
| 0400–0600 JST | 0.16 | 0.21 | 0.24 | 0.23 | | | | | | | 0.21 | | | | | |
| 0600–0800 JST | 0.18 | 0.22 | 0.24 | 0.19 | | | | | | | 0.21 | | | | | |
| 0800–1000 JST | 0.16 | 0.19 | 0.16 | 0.18 | | | | | | | 0.17 | | | | | |
| 1000–1200 JST | 0.12 | 0.17 | 0.15 | 0.12 | 0.14 | 0.19 | 0.19 | 0.19 | | | 0.14 | | | 0.18 | | |
| 1200–1400 JST | 0.11 | 0.16 | 0.19 | 0.18 | | | | | | | 0.16 | | | | | |
| 1400–1600 JST | 0.10 | 0.15 | 0.16 | 0.20 | | | | | | | 0.15 | | | | | |
| 1600–1800 JST | 0.089 | 0.18 | 0.16 | 0.19 | | | | | | | 0.15 | | | | | |
| 1800–2000 JST | 0.12 | 0.18 | 0.18 | 0.19 | | | | | | | 0.17 | | | | | |
| 2000–2200 JST | 0.14 | 0.19 | 0.20 | 0.17 | | | | | | | 0.17 | | | | | |
| 2200–0000 JST | 0.16 | 0.19 | 0.20 | 0.17 | | | | | | | 0.18 | | | | | |

[a] Time- and size-resolved $\kappa_{org}$.
[b] Time-averaged, size-resolved $\kappa_{org}$.
[c] Time-resolved, size-averaged $\kappa_{org}$.
[d] Time- and size-averaged $\kappa_{org}$.

**Table S11: Different assumptions of $\kappa_{BSOA}$ for the prediction of $F_{CCN,BSOA}$**

| $d_{dry}$ (nm) | Size-resolved $\kappa_{BSOA}$ | Size-averaged $\kappa_{BSOA}$ | Aged, size-resolved $\kappa_{BSOA}$ |
|---|---|---|---|
| 100 | 0.089 | | 0.18 |
| 200 | 0.11 | 0.11 | 0.18 |
| 300 | 0.12 | | 0.18 |
| 360 | 0.12 | | 0.19 |

**Table S12: Data in Fig. 7 of the main manuscript.**

| Hour of day | Ratios of different $F_{CCN,OA}$ | | | | Ratios of different $F_{CCN,BSOA}$ | | |
|---|---|---|---|---|---|---|---|
| | TimeSize $\kappa_{org}$[a] | SizeReso $\kappa_{org}$[b] | TimeReso $\kappa_{org}$[c] | Single $\kappa_{org}$[d] | Size-resolved $\kappa_{BSOA}$ | Size-averaged $\kappa_{BSOA}$ | Aged, Size-resolved $\kappa_{BSOA}$ |
| **0000–0200 JST** | 1 | 0.909 | 1.10 | 0.995 | 1 | 1.19 | 1.82 |
| **0200–0400 JST** | 1 | 0.926 | 1.04 | 1.01 | 1 | 1.17 | 1.83 |
| **0400–0600 JST** | 1 | 0.916 | 1.11 | 1.01 | 1 | 0.904 | 1.43 |
| **0600–0800 JST** | 1 | 0.868 | 1.05 | 0.950 | 1 | 5.58 | 9.00 |
| **0800–1000 JST** | 1 | 0.960 | 1.02 | 1.05 | 1 | 1.19 | 1.84 |
| **1000–1200 JST** | 1 | 1.10 | 1.05 | 1.19 | 1 | 1.14 | 1.68 |
| **1200–1400 JST** | 1 | 1.09 | 1.11 | 1.16 | 1 | 1.12 | 1.53 |
| **1400–1600 JST** | 1 | 1.14 | 1.14 | 1.21 | 1 | 1.12 | 1.52 |
| **1600–1800 JST** | 1 | 1.18 | 1.19 | 1.26 | 1 | 1.13 | 1.56 |
| **1800–2000 JST** | 1 | 1.06 | 1.13 | 1.15 | 1 | 1.15 | 1.62 |
| **2000–2200 JST** | 1 | 0.983 | 1.06 | 1.07 | 1 | 1.16 | 1.67 |
| **2200–0000 JST** | 1 | 0.948 | 1.04 | 1.04 | 1 | 1.18 | 1.82 |

[a] Time- and size-resolved $\kappa_{org}$.
[b] Time-averaged, size-resolved $\kappa_{org}$.
[c] Time-resolved, size-averaged $\kappa_{org}$.
[d] Time- and size-averaged $\kappa_{org}$.

| Page 29: [1] Formatted | dyg | 2019/3/14 11:00:00 |
|---|---|---|

Normal

| Page 29: [2] Formatted | dyg | 2019/3/14 11:00:00 |
|---|---|---|

Normal

| Page 29: [3] Formatted Table | dyg | 2019/3/14 11:00:00 |
|---|---|---|

Formatted Table

| Page 29: [4] Formatted | dyg | 2019/3/14 11:00:00 |
|---|---|---|

Normal, No widow/orphan control

| Page 29: [5] Formatted | dyg | 2019/3/14 11:00:00 |
|---|---|---|

Normal

| Page 29: [6] Formatted | dyg | 2019/3/14 11:00:00 |
|---|---|---|

Normal, No widow/orphan control

| Page 29: [7] Formatted | dyg | 2019/3/14 11:00:00 |
|---|---|---|

Normal

| Page 29: [8] Formatted | dyg | 2019/3/14 11:00:00 |
|---|---|---|

Normal, No widow/orphan control

| Page 29: [9] Formatted | dyg | 2019/3/14 11:00:00 |
|---|---|---|

Normal

| Page 29: [10] Formatted | dyg | 2019/3/14 11:00:00 |
|---|---|---|

Normal, No widow/orphan control

| Page 29: [11] Formatted | dyg | 2019/3/14 11:00:00 |
|---|---|---|

Font: Calibri

| Page 29: [12] Deleted | dyg | 2019/3/14 11:00:00 |
|---|---|---|

| Page 29: [12] Deleted | dyg | 2019/3/14 11:00:00 |
|---|---|---|

| Page 29: [13] Formatted | dyg | 2019/3/14 11:00:00 |
|---|---|---|

Normal

| Page 29: [14] Formatted | dyg | 2019/3/14 11:00:00 |

Normal, No widow/orphan control

| Page 29: [15] Formatted | dyg | 2019/3/14 11:00:00 |

Normal

| Page 29: [16] Formatted | dyg | 2019/3/14 11:00:00 |

Normal, No widow/orphan control

| Page 29: [17] Formatted | dyg | 2019/3/14 11:00:00 |

Normal

| Page 29: [18] Formatted | dyg | 2019/3/14 11:00:00 |

Normal, No widow/orphan control

| Page 29: [19] Formatted | dyg | 2019/3/14 11:00:00 |

Normal

| Page 29: [20] Formatted | dyg | 2019/3/14 11:00:00 |

Normal, No widow/orphan control

| Page 29: [21] Formatted | dyg | 2019/3/14 11:00:00 |

Normal

| Page 29: [22] Formatted | dyg | 2019/3/14 11:00:00 |

Normal, No widow/orphan control

| Page 29: [23] Formatted | dyg | 2019/3/14 11:00:00 |

Normal

| Page 29: [24] Formatted | dyg | 2019/3/14 11:00:00 |

Normal, No widow/orphan control

| Page 29: [25] Formatted | dyg | 2019/3/14 11:00:00 |

Normal

| Page 29: [26] Formatted | dyg | 2019/3/14 11:00:00 |

Normal, No widow/orphan control

| Page 29: [27] Formatted | dyg | 2019/3/14 11:00:00 |
|---|---|---|

Normal

| Page 29: [28] Formatted | dyg | 2019/3/14 11:00:00 |
|---|---|---|

Normal, No widow/orphan control

| Page 2: [29] Formatted | dyg | 2019/3/14 11:00:00 |
|---|---|---|

Font color: Auto

| Page 2: [30] Formatted | dyg | 2019/3/14 11:00:00 |
|---|---|---|

Font color: Auto

| Page 2: [31] Formatted | dyg | 2019/3/14 11:00:00 |
|---|---|---|

Font color: Auto

| Page 2: [32] Formatted Table | dyg | 2019/3/14 11:00:00 |
|---|---|---|

Formatted Table

| Page 2: [33] Formatted | dyg | 2019/3/14 11:00:00 |
|---|---|---|

Font color: Auto

| Page 2: [34] Formatted | dyg | 2019/3/14 11:00:00 |
|---|---|---|

Font color: Auto

| Page 2: [35] Formatted | dyg | 2019/3/14 11:00:00 |
|---|---|---|

Font color: Auto

| Page 2: [36] Formatted | dyg | 2019/3/14 11:00:00 |
|---|---|---|

Font color: Auto

| Page 2: [37] Formatted | dyg | 2019/3/14 11:00:00 |
|---|---|---|

Font color: Auto

| Page 2: [38] Formatted | dyg | 2019/3/14 11:00:00 |
|---|---|---|

Font color: Auto

| Page 2: [39] Formatted | dyg | 2019/3/14 11:00:00 |
|---|---|---|

Font color: Auto

| Page 2: [40] Formatted | dyg | 2019/3/14 11:00:00 |
|---|---|---|

Font color: Auto

| Page 2: [41] Formatted | dyg | 2019/3/14 11:00:00 |
|---|---|---|

Font color: Auto

| Page 2: [42] Formatted | dyg | 2019/3/14 11:00:00 |
|---|---|---|

Font color: Auto

| Page 2: [43] Formatted | dyg | 2019/3/14 11:00:00 |
|---|---|---|

Font color: Auto

| Page 2: [44] Formatted | dyg | 2019/3/14 11:00:00 |
|---|---|---|

Font color: Auto

| Page 2: [45] Formatted | dyg | 2019/3/14 11:00:00 |
|---|---|---|

Font color: Auto

| Page 2: [46] Formatted | dyg | 2019/3/14 11:00:00 |
|---|---|---|

Font color: Auto

| Page 2: [47] Formatted | dyg | 2019/3/14 11:00:00 |
|---|---|---|

Font color: Auto

| Page 2: [48] Formatted | dyg | 2019/3/14 11:00:00 |
|---|---|---|

Font color: Auto

| Page 2: [49] Formatted | dyg | 2019/3/14 11:00:00 |
|---|---|---|

Font color: Auto

| Page 2: [50] Formatted | dyg | 2019/3/14 11:00:00 |
|---|---|---|

Font color: Auto

| Page 2: [51] Formatted | dyg | 2019/3/14 11:00:00 |
|---|---|---|

Font color: Auto

| Page 2: [52] Formatted | dyg | 2019/3/14 11:00:00 |
|---|---|---|

Font color: Auto

| Page 2: [53] Formatted | dyg | 2019/3/14 11:00:00 |
|---|---|---|

Font color: Auto

| Page 2: [54] Formatted | dyg | 2019/3/14 11:00:00 |
|---|---|---|

Font color: Auto

| Page 2: [55] Formatted | dyg | 2019/3/14 11:00:00 |
|---|---|---|

Font color: Auto

| Page 2: [56] Formatted | dyg | 2019/3/14 11:00:00 |
|---|---|---|

Font color: Auto

| Page 2: [57] Formatted | dyg | 2019/3/14 11:00:00 |
|---|---|---|

Font color: Auto

| Page 2: [58] Formatted | dyg | 2019/3/14 11:00:00 |
|---|---|---|

Font color: Auto

| Page 2: [59] Formatted | dyg | 2019/3/14 11:00:00 |
|---|---|---|

Font color: Auto

| Page 2: [60] Formatted | dyg | 2019/3/14 11:00:00 |
|---|---|---|

Font color: Auto

| Page 2: [61] Formatted | dyg | 2019/3/14 11:00:00 |
|---|---|---|

Font color: Auto

| Page 2: [62] Formatted | dyg | 2019/3/14 11:00:00 |
|---|---|---|

Font color: Auto

| Page 2: [63] Formatted | dyg | 2019/3/14 11:00:00 |
|---|---|---|

Font color: Auto

| Page 2: [64] Formatted | dyg | 2019/3/14 11:00:00 |
|---|---|---|

Font color: Auto

| Page 2: [65] Formatted | dyg | 2019/3/14 11:00:00 |
|---|---|---|

Font color: Auto

| Page 2: [66] Formatted | dyg | 2019/3/14 11:00:00 |
|---|---|---|

Font color: Auto

Page 2: [67] Formatted                    dyg                    2019/3/14 11:00:00

Font color: Auto

Page 2: [67] Formatted                    dyg                    2019/3/14 11:00:00

Font color: Auto